# SPA-BENCH: A COMPREHENSIVE BENCHMARK FOR SMARTPHONE AGENT EVALUATION

**Jingxuan Chen[1*], Derek Yuen[1*], Bin Xie[2], Yuhao Yang[1], Gongwei Chen[2], Zhihao Wu[1],**

**Yixing Li[2], Xurui Zhou[2], Weiwen Liu[1], Shuai Wang[1], Kaiwen Zhou[1], Rui Shao[2†],**

**Liqiang Nie[2], Yasheng Wang[1], Jianye Hao[1,3], Jun Wang[4], Kun Shao[1†]**

[1]Huawei Noah's Ark Lab, [2]Harbin Institute of Technology, Shenzhen, [3]Tianjin University,
[4]AI Centre, University College London

## ABSTRACT

Smartphone agents are increasingly important for helping users control devices efficiently, with (Multimodal) Large Language Model (MLLM)-based approaches emerging as key contenders. Fairly comparing these agents is essential but challenging, requiring a varied task scope, the integration of agents with different implementations, and a generalisable evaluation pipeline to assess their strengths and weaknesses. In this paper, we present SPA-BENCH, a comprehensive **S**mart**P**hone **A**gent **Bench**mark designed to evaluate (M)LLM-based agents in an interactive environment that simulates real-world conditions. SPA-BENCH offers three key contributions: (1) A diverse set of tasks covering system and third-party apps in both English and Chinese, focusing on features commonly used in daily routines; (2) A plug-and-play framework enabling real-time agent interaction with Android devices, integrating over ten agents with the flexibility to add more; (3) A novel evaluation pipeline that automatically assesses agent performance across multiple dimensions, encompassing seven metrics related to task completion and resource consumption. Our extensive experiments across tasks and agents reveal challenges like interpreting mobile user interfaces, action grounding, memory retention, and execution costs. We propose future research directions to ease these difficulties, moving closer to real-world smartphone agent applications. SPA-BENCH is available at `https://ai-agents-2030.github.io/SPA-Bench/`.

## 1 INTRODUCTION

The growing capabilities of Large Language Models (LLMs) and Multimodal Large Language Models (MLLMs) have broadened the application of AI agents across various domains (Gur et al., 2023; Gou et al., 2023; Cai et al., 2023; Li et al., 2023a; Wang et al., 2023; Wu et al., 2023a). One promising area is smartphone control, where agents assist users in tasks like booking hotels or setting alarms. These agents can be broadly categorised into two main types: (1) agent-as-a-model (Lai et al., 2024), where fine-tuned or pre-trained (M)LLMs are customised for agentic tasks (Zhan & Zhang, 2023; Hong et al., 2024; Bai et al., 2024; Lu et al., 2024; Christianos et al., 2024; Wang et al., 2024d), and (2) agentic workflow (Shang et al., 2024), which typically relies on off-the-shelf models and modular designs to support agentic functionality (Yang et al., 2023b; Wen et al., 2024; Wang et al., 2024b;a; Rawles et al., 2024a). In both cases, these models act as the "brains" for decision-making. The information these agents use to interact with smartphones can vary, with common methods involving direct screen observation (Wang et al., 2024b;a; Zhan & Zhang, 2023; Hong et al., 2024; Bai et al., 2024; Lu et al., 2024), accessing non-visible data via Android View Hierarchy or Extensible Markup

---

* Equal contribution.
† Corresponding authors: shaorui@hit.edu.cn, shaokun2@huawei.com.

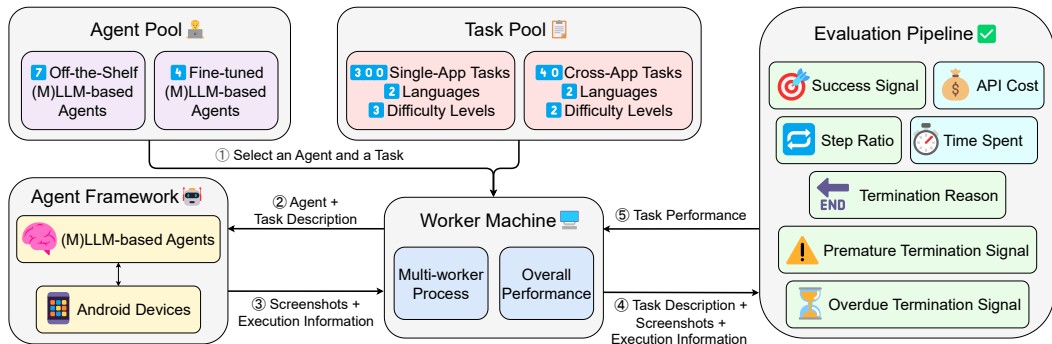

Figure 1: An overview of SPA-BENCH. The worker machine iterates through the task and agent pools, assigning tasks to agents within the framework for execution, and then passes the execution results to the evaluation pipeline for measuring task completion and resource consumption performance.

Language (XML) (Wen et al., 2024), or a combination of both (Yang et al., 2023b; Rawles et al., 2024a).

As the number of (M)LLM-based smartphone agents grows, fair performance comparisons become crucial for identifying their strengths and shortcomings, leading to an increasing need for benchmarking (Chan et al., 2023; Liu et al., 2023a;b; Wu et al., 2023b). Regarding smartphone agent benchmarks, existing studies use three main approaches to evaluate agents: actions-based (Xing et al., 2024), states-based (Rawles et al., 2024a; Zhang et al., 2024; Lee et al., 2024; Wang et al., 2024c), or a hybrid of both (Wang et al., 2024c). Each method faces specific difficulties: action-based evaluation may involve multiple correct sequences, while state-based methods struggle to determine the appropriate post-action state. A hybrid approach could mitigate these limitations, but the challenge lies in effectively utilising both action and state information.

Despite these efforts, current research (Rawles et al., 2024a; Xing et al., 2024; Zhang et al., 2024; Lee et al., 2024; Wang et al., 2024c) still has several key limitations: (1) The focus remains primarily on system and Google suite applications (apps) in English, which are often free from distractions like ads and pop-ups that could introduce complexity and randomness; (2) The number of evaluated agents is typically fewer than five, with some studies including only similar variants of the same agent; (3) Automated success detection methods frequently require human intervention (e.g., handcrafted validation logic for each task) or rely on data that may be inaccessible in certain cases (e.g., Android View Hierarchy data, which is unavailable in WebView apps (Xing et al., 2024)).

In this paper, we introduce SPA-BENCH, a **S**mart**P**hone **A**gent **Bench**mark designed to evaluate more than 10 smartphone control agents in daily tasks. As illustrated in Figure 1, SPA-BENCH comprises 340 tasks, including 150 single-app tasks and 20 cross-app tasks, in both English and Chinese apps, as well as third-party ones. It integrates 11 agents into a unified framework based on their original implementations. This framework is linked to an automated evaluation pipeline that measures agent performance and can be applied to additional tasks beyond this benchmark, without requiring human inputs. Our experiments show that agents following the agentic workflow outperform those in the agent-as-a-model category, although the former one remain impractical for real-world deployment due to time and cost constraints. We also provide a detailed discussion on the challenges and future directions for smartphone agents, covering topics such as building perceptive mobile interfaces, reasoning mechanisms, and user-friendly applications.

In summary, our comprehensive benchmark makes several key contributions: (1) **a diverse task collection** of 340 tasks with increasing difficulty, accompanied by human trajectory annotations. It covers both English and Chinese apps, including single-app and cross-app scenarios, and featuring 58 third-party apps (Section 3); (2) **a plug-and-play agent framework** supporting 11 agents, which allows for easy integration of new agents with minimal adaptation and offers features like automatic Android setup and multi-device emulator support (Section 4); (3) **an automated and scalable evaluation pipeline** assesses agent performance using task completion and resource consumption metrics. It employs success detection methods that achieve average F1 scores of 90.5% for single-app

Table 1: Comparison of SPA-BENCH and other smartphone agent benchmarks. Agents that differ only in their base models are not counted as separate agents.

| Benchmark | Third-party app? | Cross-app? | Chinese app? | Difficulty level? | Number of tasks | Number of agents | Number of metrics | Free of hand-crafted validation? | Information for success detection |
|---|---|---|---|---|---|---|---|---|---|
| AndroidArena (Xing et al., 2024) | ✗ | ✓ | ✗ | ✗ | 221 | 1 | 4 | ✗ | Action only |
| AndroidWorld (Rawles et al., 2024a) | ✓ | ✓ | ✗ | ✓ | 116 | 3 | 1 | ✗ | State only |
| LlamaTouch (Zhang et al., 2024) | ✓ | ✗ | ✗ | ✓ | 495 | 4 | 1 | ✗ | State only |
| B-MoCA (Lee et al., 2024) | ✗ | ✗ | ✗ | ✗ | 60 | 3 | 1 | ✗ | State only |
| MobileAgentBench (Wang et al., 2024c) | ✗ | ✗ | ✗ | ✓ | 100 | 5 | 6 | ✗ | Action and State |
| **SPA-BENCH** | ✓ | ✓ | ✓ | ✓ | 340 | 11 | 7 | ✓ | Action and State |

tasks and 84.5% for cross-app tasks compared to human evaluators (Section 5); and (4) **extensive experiments** across agents and tasks, providing a detailed analysis of current smartphone agent capabilities and limitations, while also offering directions for future research (Section 6).

## 2 RELATED WORK

**Smartphone Agent.** Smartphone agents aim to automate tasks on mobile apps in a human-like way. Early agents, like Siri and Google Assistant, relied on system-level APIs and customisation, limiting their generality. Recently, (M)LLM-based agents have emerged, using the user interface (UI) to achieve a more general approach. These agents, with (M)LLMs as their "brains", also require "hands" (actions) and "eyes" (observations) to interact with smartphones. They are based on either off-the-shelf or fine-tuned models and perform human-like actions (e.g., tapping, typing, and swiping). According to how they observe the UI, recent works are categorised into text-based, vision-based, and combined approaches. Text-based methods (Wen et al., 2024; Rawles et al., 2024a) rely on UI document data (e.g., XML) or convert visual information into text, vision-based methods (Wang et al., 2024b;a; Zhan & Zhang, 2023; Hong et al., 2024; Bai et al., 2024; Lu et al., 2024) use screenshots to capture the complete visual context, while combined approaches (Yang et al., 2023b; Rawles et al., 2024a) integrate both text and vision inputs for greater informativeness. **SPA-BENCH** evaluates all three types of agents to provide a comprehensive comparison of their capabilities.

**Smartphone Agent Evaluation.** Effective evaluation of smartphone agents is crucial for identifying limitations and guiding improvements. Success rate, which measures task completion, is the most commonly used metric, with some studies also considering efficiency. Success detection methods are generally classified into two types: human detection (Yang et al., 2023b; Wang et al., 2024b;a), which is accurate but resource-intensive, and automated detection, which is less costly but varies in accuracy. Current automated methods primarily rely on hand-crafted validation logic, making them unscalable without human intervention. They are restricted to evaluating tasks involving apps that are limited to English-only and simpler apps (e.g., system, Google Suite, and open-source apps), with minimal coverage of other third-party ones. These automated methods can be further divided into action-based, state-based, and hybrid approaches. Action-based methods (Xing et al., 2024) compare agents' actions to human demonstrations but struggle with the non-unique nature of correct action sequences. State-based methods (Rawles et al., 2024a; Zhang et al., 2024; Lee et al., 2024) assess whether essential states are reached but may miss minor actions. Hybrid approaches (Wang et al., 2024c) combine state and action data for more accurate success detection. **SPA-BENCH** introduces two hybrid approaches for evaluating single-app and cross-app tasks. Compared to other automated methods, our approaches support a wider range of apps and tasks. They do not rely on hand-crafted validation logic, making them adaptable without human intervention. Table 1 presents a comparison between our work and other automated evaluation-based smartphone agent benchmarks, highlighting our comprehensive evaluation of various agents in diverse tasks across multiple dimensions.

## 3 SPA-BENCH TASK

### 3.1 OVERVIEW

SPA-BENCH builds a collection of smartphone agent tasks across both English and Chinese apps, featuring 39 English and 29 Chinese apps divided into eight categories based on core features (see Appendix B.1). The collection includes 150 single-app tasks and 20 cross-app tasks for each language. These tasks focus on core app functions that reflect everyday use, providing a realistic assessment of

smartphone agents' performance. The inclusion of diverse Chinese and third-party apps increases complexity, primarily due to the difficulties agents encounter in understanding Chinese and navigating more intricate UIs. A complete list of tasks is provided in Appendix B.2.

The single-app tasks are grouped into sets, with progressively increasing complexity across three difficulty levels. In each set, Level 1 tasks serve as foundational and straightforward activities, while Level 2 and Level 3 tasks introduce more complex requirements, such as handling intricate UI elements or animations. Generally, Level 1 tasks require fewer than five actions, while Level 2 tasks typically involve fewer than ten, and Level 3 tasks fewer than fifteen. While each set follows similar instructions, the tasks are designed to use distinct entities (e.g., creating folders with different names) to prevent any influence from earlier tasks. Examples of single-app tasks are shown in Figure 2. For cross-app tasks, we refer to the recent work GUI Odyssey (Lu et al., 2024),

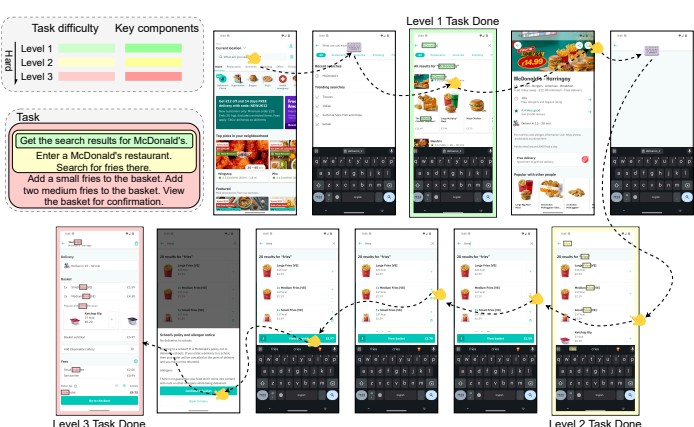

Figure 2: A sample set of tasks within the Deliveroo app, annotated by human. In this example, simpler tasks form the foundation for more complex ones, resulting in shared trajectories in the initial stages. The final screenshots for tasks of all three difficulty levels are highlighted in corresponding colours. Each final screenshot highlights the key components used in coarse detection (explained further in Section 5), with the zoomed-in versions available in Appendix B.3.

which defines six task types: General Tool, Information Management, Web Shopping, Media Entertainment, Social Sharing, and Multi-Apps. Unlike single-app tasks, cross-app difficulty levels are determined by the number of apps involved in a task. Level 1 tasks require interactions between two apps, while Level 2 tasks necessitate switching between three. As the number of apps increases, complexity arises not only from additional steps but also from inter-app dependencies and coordination. Our cross-app tasks include three Level 1 tasks for each of the first five types, and five Level 2 tasks for the Multi-Apps type. Appendix B.4 provides examples.

## 3.2 TASK CONSTRUCTION

Our tasks were primarily constructed by human annotators. For single-app tasks, we selected commonly used apps and supplemented them with apps from related works (Yang et al., 2023b; Wang et al., 2024b). Based on each app's core features, tasks were created following an annotation guideline specifying: (1) A clear **task description** that reflects the task's goal and difficulty level. For descriptions inspired by prior works, we standardised them and assigned appropriate difficulty levels. (2) A **human-executed trajectory** presented as a series of screenshots. Between any two adjacent screenshots, only one action (e.g., tap, type) is allowed. The total number of actions in the human execution serves as the "golden steps" in our experiments. To ensure a reproducible and unbiased baseline, we instruct human annotators to avoid irrelevant actions and refrain from using shortcuts that are inherently dynamic, influenced by factors such as recommendation algorithms or user-specific history. (3) **Key components of the final state**, which are pieces of text that must appear in the final screenshot if the task is successfully completed. We focus only on the final state because there may be multiple correct paths to complete the task, but they typically converge to the same final state (Wang et al., 2024c). These key components are designed for future use, as detailed in Section 5.2. For cross-app tasks, annotations include only task descriptions and human-executed trajectories due to the flexibility of final states. Most cross-app English tasks were drawn from GUI Odyssey (Lu et al., 2024), and we reformatted descriptions and recollected trajectories where necessary.

To ensure task quality, a validation process followed task annotation. Annotators cross-checked all tasks for clarity, trajectory accuracy, and key component quality. The tasks were also tested

across different Android devices, Android versions, and app versions to verify feasibility. The same validation was repeated before experiments.

In total, SPA-BENCH includes 300 single-app and 40 cross-app tasks, evenly split between English and Chinese. Each task may consist of multiple subtasks (e.g., adding, modifying, deleting, searching). The distribution of steps performed by humans for these tasks, categorised by task type, is illustrated in Appendix B.5.

## 4 AGENT FRAMEWORK

### 4.1 A UNIFIED PLUG-AND-PLAY FRAMEWORK

Our framework facilitates the execution of autonomous smartphone agents and tasks. As shown in Figure 3, the worker machine manages communication, providing task descriptions and receiving outcomes (trajectories and logs). It hosts multiple worker processes, each connecting an Android emulator and an agent. Each agent interacts with the Android device by performing actions based on observations, such as taking screenshots and generating actions like taps or swipes. The snapshot state is restored at the start of each experimental cycle.

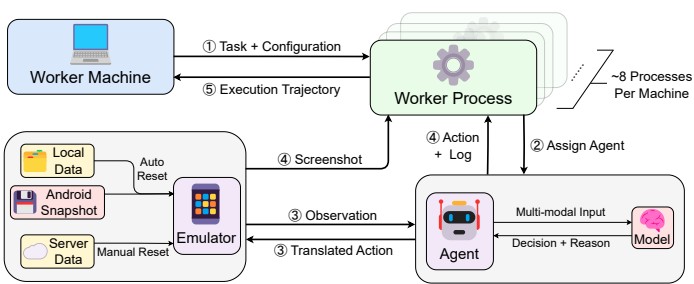

Figure 3: An overview of the agent framework using a multiprocessing architecture. Each worker process connects an agent to an Android emulator, and they interact multiple times throughout the task (i.e., step 3 is repeated) until completion. The emulators are reset after the agent has executed all assigned tasks.

The framework is highly scalable. Unlike existing research (Rawles et al., 2024a; Xing et al., 2024; Zhang et al., 2024; Lee et al., 2024; Wang et al., 2024c), which integrates a limited number of agents tightly into the framework, ours allows easy addition of new agents with minimal integration, ensuring each agent operates independently within an isolated environment. Details about the agents integrated into our framework are provided in Appendix C.

### 4.2 SNAPSHOT-BASED EMULATOR FOR CONSISTENT TESTING

The framework integrates Android emulators as a scalable alternative to physical devices, replicating most Android functions for parallel testing and rapid experiment deployment. For instance, a 24-core CPU with 64GB RAM can support up to eight emulators or worker processes simultaneously, depending on the agents' resource needs.

To ensure consistency, emulators can be quickly loaded from snapshots, which capture and restore system states (e.g., installed apps, login credentials, and local settings). This eliminates repetitive setup processes by preserving pre-configured settings (e.g., a pre-existing contact for messaging tasks). However, since some app data is stored externally, manual intervention is required after each experiment cycle, such as unsubscribing from channels post-task completion.

## 5 AUTOMATED EVALUATION PIPELINE

### 5.1 METRICS

We define seven key metrics for comprehensive evaluation:

**Completion-related Metrics**. (1) **Success signal** – a binary indicator of task success. For single-app and cross-app tasks, we develop two different hybrid approaches that leverage both action and state information, allowing for multiple valid execution paths. These approaches eliminate the need for human evaluators and handcrafted evaluation logic (details are provided in Section 5.2). (2) **Step**

**ratio** – measures execution efficiency by comparing agent steps with human steps (the "golden steps" from Section 3.2). This is considered only when the task is successful (i.e., success signal is "true"). A higher ratio indicates more unnecessary actions and lower efficiency. (3) **Termination reason** – explains why the task was terminated, including self-reported completion (i.e., an agent proactively terminates a task based on its belief that the task has been completed successfully), reaching the configured maximum step limit, or execution errors (e.g., invalid actions).(4) **Premature termination signal** – a binary indicator applicable only when the termination reason is self-reported completion. It is set to "true" when the success signal is "false", indicating the agent mistakenly believed the task was completed. This premature stopping reduces success rates by causing the agent to assume success before finishing the task. (5) **Overdue termination signal** – a binary indicator applicable only when the termination reason is reaching the maximum step limit. It is set to "true" when the success signal is "true", meaning the agent mistakenly thought the task was incomplete. This results in unnecessary steps, reducing efficiency as the agent takes extra actions before concluding the task.

**Consumption-related Metrics**. (6) **Time spent** – the time taken for task execution, recorded in seconds. (7) **API cost** – the monetary cost incurred by API usage, measured in US dollars. However, these two metrics apply only to agents using proprietary MLLMs, as for locally hosted fine-tuned models, the time taken heavily depends on computational resources, and there are no monetary costs from external API calls.

## 5.2 SUCCESS DETECTION

**Single-App Success Detection.**
We employ a coarse-to-fine success detection pipeline that uses key component matching followed by MLLM evaluation. As shown in Figure 4, for each agent-task pair, the pipeline first applies coarse detection using the annotated key components introduced in Section 3.2, filtering out trajectories irrelevant to the task. If passed, fine detection follows, using an MLLM evaluator for final success determination. This approach is

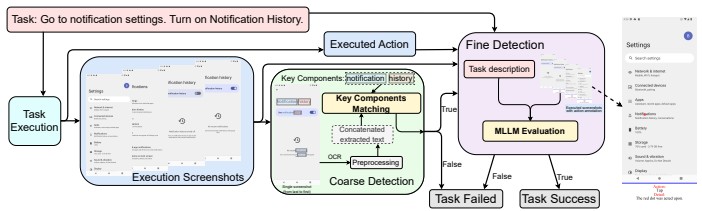

Figure 4: An example of our single-app success detection pipeline. It features coarse detection through key component matching on execution screenshots and pre-annotated key components, followed by fine detection using MLLM evaluation given action information.

motivated by the need to balance scalability and cost efficiency, addressing the limitations of relying on extensive human labour or expensive MLLM-based evaluations in large-scale benchmarks. We compared our single-app success detection approach with human evaluations and found it achieves an F1 score of 0.926 for English tasks and 0.884 for Chinese tasks. Further details on the single-app success detection and its performance can be found in Appendix D.

**Cross-App Success Detection.**
Unlike single-app success detection which processes the entire task at once, our cross-app approach splits tasks into subtasks and evaluates them sequentially. This is because cross-app tasks are usually longer than single-app tasks and require switching between multiple apps, increasing the complexity of success detection. As illustrated in Figure 5, a MLLM first generates subtasks based on the involved apps, followed by a human review. During evaluation, another MLLM splits the trajectory into multiple segments based solely on

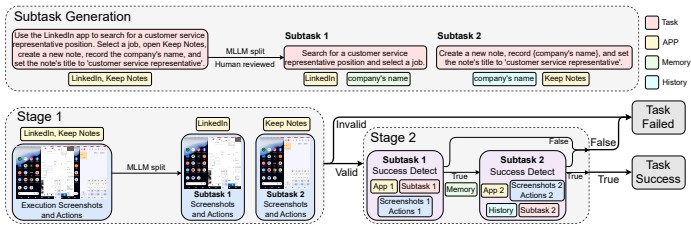

Figure 5: An example of our cross-app success detection pipeline that is based on subtasks instead of the entire task. The first stage involves splitting the full trajectory into segments, while the second stage checks the subtasks sequentially.

each app in the ordered list. If the segmentation is valid, each subtask is then evaluated sequentially

until either the final subtask is checked or an earlier subtask fails. Our cross-app success detection method closely aligns with human evaluations, achieving an F1 score of 0.845. More details on the cross-app success detection and its performance are provided in Appendix E.

# 6 EXPERIMENTS

In this paper, the success rate results were derived using the automated success detection methods outlined in Section 5.2, with GPT-4o serving as the MLLM. To account for agents with multiple variants, detailed configurations for each agent are provided in Appendix F.1. Furthermore, case studies illustrating various agent behaviors are presented in Appendix H.

## 6.1 OVERVIEW OF SUCCESS RATE

Table 2 shows the overall success rates. Notably, M3A consistently achieved the highest success rates across all task sets. We found that agents generally performed English tasks better than Chinese tasks, agents with the agentic workflow outperformed those categorised as agent-as-a-model, and cross-app tasks were more challenging than single-app tasks for agents.

**Comparison in Single-App Tasks.** For single-app English tasks, M3A, T3A, and MobileAgentV2 were the best-performing ones, with success rates ranging from 0.640 to 0.433. These agents are equipped with reflection modules that help prevent them from stalling. AppAgent and AutoDroid performed less well, though they would likely had performed better with access to external knowledge documents, as in their original implementations. For single-app Chinese tasks, MobileAgentV2 outperformed T3A, while its performance was more comparable to M3A. A potential factor could be the overly complex accessibility (a11y) tree layout used by T3A. MobileAgentV2, relying

Table 2: Success rates across all tasks and agents in this benchmark, categorised by task type. The first seven agents fall under the category of agentic workflow, while the last four belong to agent-as-a-model. AutoDroid was tested only on single-app tasks as its agent framework, Droidbot (Li et al., 2017), supports only these tasks.

| Agent | Single-App | | | Cross-App | | |
|---|---|---|---|---|---|---|
| | Overall | English | Chinese | Overall | English | Chinese |
| *Agentic Workflow (GPT-4o)* | | | | | | |
| AppAgent | 0.294 | 0.340 | 0.247 | 0 | 0 | 0 |
| AutoDroid | 0.257 | 0.327 | 0.187 | - | - | - |
| MobileAgent | 0.314 | 0.387 | 0.240 | 0.075 | 0.050 | **0.100** |
| MobileAgentV2 | 0.437 | 0.433 | 0.440 | 0.100 | 0.100 | **0.100** |
| M3A | **0.544** | **0.640** | **0.447** | **0.150** | **0.200** | **0.100** |
| T3A | 0.434 | 0.487 | 0.380 | 0.100 | 0.100 | **0.100** |
| SeeAct | 0.360 | 0.393 | 0.327 | 0.075 | 0.100 | 0.050 |
| *Agent-as-a-Model* | | | | | | |
| Auto-UI | 0.010 | 0.013 | 0.007 | 0 | 0 | 0 |
| CogAgent | 0.027 | 0.027 | 0.027 | 0 | 0 | 0 |
| DigiRL | 0.010 | 0.020 | 0 | 0 | 0 | 0 |
| OdysseyAgent | 0.037 | 0.053 | 0.020 | 0 | 0 | 0 |

on OCR and raw screenshots, averaged 12,400 prompt tokens per step in Chinese single-app tasks, compared to T3A's 22,000 tokens using only a11y trees, indicating larger or more intricate structures in Chinese apps, potentially contributing to the agent's degraded performance. A similar trend was observed in English single-app tasks, with lower token usage across both agents: 11,200 for MobileAgentV2 and 19,700 for T3A. In general, a decrease in success rates for Chinese tasks was observed due to the limited capabilities of (M)LLMs in Chinese, compounded by the increased complexity of Chinese apps. These apps often feature more intricate layouts, frequent animations, and distracting elements such as ads and pop-ups.

**Impact of Core Models and Input Modalities.** There was a significant gap in success rates between agents using proprietary models like GPT-4o and those based on fine-tuned models. Agents following the agentic workflow significantly outperformed those in the agent-as-a-model category, the latter of which often struggled to complete any tasks. This contrasts with the high action matching scores reported in prior studies (Zhan & Zhang, 2023; Hong et al., 2024; Bai et al., 2024; Lu et al., 2024), indicating that fine-tuned agents are often optimised for generating textual actions based on fixed UI scenarios. While such optimisations may achieve high accuracy in offline environments, they often fail in dynamic, real-world settings. For example, a tap action is deemed successful if its coordinates fall within 14% of the screen distance to the ground truth (Rawles et al., 2024b), but this tolerance

Table 3: Task performance on single-app English tasks. SRC and MSR refer to Self-Reported Completion and Maximum Steps Reached, respectively. The execution time and token costs of the last four agents are omitted because they use locally hosted open-source models.

| Agent | Success Rate | Mean Step Ratio on Success | Termination Reason | | | Termination Inaccuracy | | Mean Exec Time per Step (sec) | Mean Token Cost per Step (USD) |
|---|---|---|---|---|---|---|---|---|---|
| | | | SRC Rate | MSR Rate | Error Rate | Premature Rate | Overdue Rate | | |
| *Agentic Workflow (GPT-4o)* | | | | | | | | | |
| AppAgent | 0.340 | 1.33 | 0.327 | 0.507 | 0.166 | 0.347 | 0.197 | 26.5 | 0.014 |
| AutoDroid | 0.327 | 1.10 | 0.593 | 0.340 | 0.067 | 0.494 | 0.078 | 34.0 | **0.008** |
| MobileAgent | 0.387 | 1.24 | 0.367 | 0.633 | **0** | 0.109 | 0.095 | 27.1 | 0.053 |
| MobileAgentV2 | 0.433 | 1.05 | 0.580 | 0.420 | **0** | 0.333 | 0.111 | 56.1 | 0.067 |
| M3A | **0.640** | **0.92** | **0.847** | **0.153** | **0** | 0.244 | **0** | 19.3 | 0.092 |
| T3A | 0.487 | 1.04 | 0.707 | 0.293 | **0** | 0.368 | 0.136 | **9.6** | 0.116 |
| SeeAct | 0.393 | 1.60 | 0.200 | 0.773 | 0.027 | **0.100** | 0.276 | 41.2 | 0.046 |
| *Agent-as-a-Model* | | | | | | | | | |
| Auto-UI | 0.013 | 1.50 | 0.060 | 0.940 | 0 | 1.000 | 0.015 | - | - |
| CogAgent | 0.020 | 1.67 | 0.147 | 0.820 | 0.033 | 1.000 | 0.024 | - | - |
| DigiRL | 0.020 | 1.52 | 0.227 | 0.607 | 0.166 | 0.971 | 0.022 | - | - |
| OdysseyAgent | 0.053 | 2.00 | 0 | 1.000 | 0 | - | 0.013 | - | - |

can cause inaccuracies with actionable elements in practice. Furthermore, reliance on predefined scenarios limits the agents' ability to generalise to unseen UI contexts or to recover from detoured states caused by mistaken actions. On the other hand, agents utilising agentic workflow are typically equipped with input from visual modules, such as mark-up documents and set-of-marks (Yang et al., 2023a). These layout documents are sometimes incomplete, failing to capture all available UI elements on the interface. In other cases, they are unnecessarily complex for models to handle, as seen in the case of T3A mentioned above. This highlights a critical gap in grounding capabilities, which are essential for end-to-end task completion but remain challenging especially for fine-tuned models (Zheng et al., 2024).

**Complexity and Memory Retention in Cross-App Task.** For cross-app tasks, most agents, except M3A, completed no more than 4 tasks in total across both English and Chinese apps. Although M3A performed better, completing 6 out of 40 tasks, overall performance was still low, reflecting the complexity of cross-app tasks. These tasks require more steps, reasoning, and the ability to switch between apps while retaining memory of previous actions. In some cases, agents might nearly complete the task but fail in the end due to minor mistakes or missed requirements, especially in long sequences or multi-context scenarios. Even OdysseyAgent (Lu et al., 2024), specifically designed for cross-app tasks, faced difficulty completing them end-to-end. It sometimes handled subtasks within a single app well but faltered when transitioning between apps, illustrating the challenge of maintaining context and reasoning across environments. These findings suggest that current agents, including the best-performing ones, struggle with multi-step cross-app tasks, often losing context or forgetting prior actions. This highlights the need for better memory mechanisms, enhanced inter-app reasoning, and advanced handling of complex, multi-context environments (Shinn et al., 2023; Li et al., 2023b; Pan et al., 2024). These capabilities are essential for tasks users expect autonomous agents to manage.

## 6.2 COMPLETION- AND CONSUMPTION-RELATED METRICS

When comparing completion- and consumption-related metrics across agents, we observed consistent trends across single-app and cross-app tasks in both English and Chinese. Since the single-app English results are the most comprehensive, this section focuses primarily on those results, with additional details available in Appendix F.2. Table 3 shows full task performance for single-app English scenarios.

**Step Efficiency and Success Rate.** As discussed in Section 6.1, agents with the agentic workflow substantially outperformed those belong to agent-as-a-model. Higher success rates correlate with lower mean step ratios, indicating more efficient task completion with fewer unnecessary actions or errors. Conversely, agents facing difficult tasks tend to make more errors, even if they ultimately succeed. M3A exhibited a notably low mean step ratio of 0.92, indicating it used fewer steps than a human. This efficiency is partly achieved through combined actions specifically defined by the agent itself, where a single action encompasses multiple operations, such as typing in the search box

and pressing "enter" in one step. Agents may also exploit strategic shortcuts, such as clicking on a recommended item instead of using the search bar. Thus, both approaches allow agents to reduce the steps needed to complete a task.

**Task Termination and Success Rate.** Regarding task termination, a higher success rate generally aligns with a higher Self-Reported Completion (SRC) rate and a lower Maximum Steps Reached (MSR) rate. Agents terminated tasks either when they believed the task was complete or when they reached the step limit or encounter errors. However, agents did not always accurately determine task completion, leading to discrepancies between success rates and SRC rates. This can be further analysed by examining the premature termination rate (PTR) and overdue termination rate (OTR). As mentioned in Section 5.1, PTR can affect the success rate, while OTR can influence task efficiency. Notably, a pattern emerges where agents with a lower PTR tend to have a higher OTR. This compromise likely arises from the agent's internal decision thresholds. For instance, SeeAct exhibited the lowest PTR (0.100) but the highest OTR (0.276). This demonstrates a trade-off in the sensitivity of the agent's internal success detector, balancing the risk of premature termination with the tendency to extend task completion unnecessarily. An ideal success detector should minimise both premature and overdue terminations to optimise both task accuracy and efficiency.

**Enhancing Robustness through Error Handling Mechanisms.** Error-handling mechanisms are crucial for improving success rates and ensuring reliable performance. Agents lacking these mechanisms were more prone to failure, which led to early termination when execution errors occurred. Common issues include parsing errors arising from the agents' inability to correctly interpret model outputs as valid actions. For example, the output may be missing specific phrases, such as "Thought: ", that are required by the agent's parsing module. Some agents also encountered failures when necessary inputs (e.g., XML files) could not be accessed. These failures highlight the need for better error detection and recovery strategies, allowing agents to correct mistakes and improve their overall success rates.

**Limitations in Cost and Efficiency for Real-World Use.** While agents categorised as agent-as-a-model do not incur token costs and their execution time varies with device power, their low success rates make them impractical for deployment. Among agents with the agentic workflow, AutoDroid is the most cost-effective, using only \$0.008 per step due to its text-based input. However, it has a long execution time (34 seconds per step) and a success rate of only 0.327. M3A and T3A, though faster (under 20 seconds per step) and more successful, have higher token costs at around \$0.10 per step due to the complexity of inputs generated by UI elements. MobileAgentV2, while more affordable at \$0.067 per step, suffers from a complex visual perception pipeline, resulting in the longest execution time (56.1 seconds per step). These results highlight the trade-off between efficiency and effectiveness. Agents like T3A, despite achieving relatively high success rates and faster execution times, still fall short of human-level usability due to their monetary cost. Such limitations stem from two major factors. One is the delay between UI information collection and action execution, which can cause inaccuracies especially when dynamic content appears. The other is the agents' slower speeds and higher costs compared to human users. Users are unlikely to rely on an autonomous agent to complete a task if they have to wait for extended periods or pay several dollars, especially when they could complete it in a few steps themselves.

**Performance Variation Across Difficulty Levels, Open-Ended Task Experiments, and Case Studies.** We compared agent performance across difficulty levels, showing that easier tasks are executed more successfully, as demonstrated in Appendix F.3. To further explore the scalability of our success detection approaches, we conducted initial experiments on "open-ended" single-app English tasks, detailed in Appendix G. In Appendix H, we present three case studies that illustrate representative scenarios of agent task execution.

## 6.3 KEY INSIGHTS

To enhance the performance of autonomous smartphone agents, future research may need to address several core dimensions, including UI understanding and action grounding, dataset diversity, memory retention, reflection and error-handling mechanisms, internal task termination recognition, and execution efficiency.

First, integrating more advanced visual perception modules is essential for enhancing agents' understanding of complex UI layouts and precise action grounding across various scenarios. Although agents using a11y trees and OCR have shown relatively good performance in English tasks, their

effectiveness is still limited in Chinese tasks, which often feature more visually complex and dynamic content. Currently, some agents struggle to ground actions in these dynamic environments, often failing to recognise actionable elements or map generated actions to the correct coordinates. Future designs should focus on building more robust visual models that can accurately interpret these environments and perform end-to-end task completion in interactive settings.

Diversifying fine-tuning datasets is also essential for making agents more generalisable. Datasets should include various task instruction formats, languages, and both single-app and cross-app scenarios to better simulate real-world conditions. This would ensure that agents are prepared to handle a broader range of interactions, particularly in multilingual environments where language and UI complexity vary.

Memory retention mechanisms can be improved as well, especially for handling long, multi-step tasks that span multiple apps. Current agents often lose context during complex tasks or app transitions, which leads to incomplete task execution. Memory-augmented networks or episodic memory architectures could enable agents to retain context across transitions, which is particularly valuable in cross-app scenarios where agents usually struggle. These scenarios closely resemble real-world tasks that require continuity and context recall over extended sequences.

Reflection and error-handling capabilities are another critical area for improvement. Many agents fail to learn from mistakes, repeatedly making the same errors without self-correction. Implementing robust reflection modules, similar to those found in M3A, would allow agents to better assess their past actions and adjust their strategies dynamically. Additionally, error-handling mechanisms, such as error identification, recovery loops, self-correction, and fallback strategies, are vital for maintaining performance in unpredictable, dynamic environments. Agents need to be able to detect and resolve issues such as invalid model outputs, unactionable UI elements, or parsing errors, rather than terminating prematurely or getting stuck in unproductive actions.

In task termination, agents must carefully balance premature and overdue termination. Some agents still struggle to accurately determine when a task is truly complete. For example, while SeeAct showed a low premature termination rate, it also exhibited a high overdue termination rate. This indicates that although SeeAct avoided ending tasks prematurely, it often failed to recognise when tasks were completed, leading to inefficiencies. A well-designed internal success detector can minimise both types of termination inaccuracies, thereby improving task accuracy and efficiency.

Finally, execution time and cost need to be optimised for real-world deployment. Agents such as MobileAgentV2, which rely on multiple modules, need to reduce overhead and streamline execution to minimise task completion time. MLLM-based agents, in contrast to T3A, may also focus on reducing input context size to lower token costs while preserving critical information for task completion. A hybrid model approach that combines the speed and efficiency of lightweight models with the robustness of more complex ones could provide a promising solution for balancing performance and

## 7 CONCLUSION

In this paper, we introduced SPA-BENCH, a comprehensive benchmark for evaluating smartphone agents across diverse tasks. The evaluation covers English and Chinese apps, single-app and cross-app scenarios, and varying difficulty levels. Our experiments reveal that even the best-performing agents can complete less than 70% of tasks successfully, and there are significant performance gaps between agents following the agentic workflow and those in the agent-as-a-model category, particularly in action grounding and generalisation within complex Chinese apps. While some agents excel in simpler tasks, their long execution times and high costs limit their practicality for real-world use. Our findings highlight the need for better memory mechanisms, robust error handling, accurate self-evaluator, improved integration of reasoning with UI understanding, and optimising execution time and cost for real-world deployment. Additionally, agents based on fine-tuned models should be adapted to diverse scenarios and focus on long-sequence decision-making rather than isolated actions. By developing SPA-BENCH as a fair and scalable benchmark, we aim to accelerate the development of more efficient, practical, and user-friendly smartphone agents.

ACKNOWLEDGEMENTS

This work was supported by the National Natural Science Foundation of China (Grant Nos. 62422605, 92370132).

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

APPENDIX

## A  LIMITATION AND FUTURE WORK

Given that constructing tasks is both time-consuming and resource-intensive, SPA-BENCH currently includes 300 single-app tasks and 40 cross-app tasks, evenly split between English and Chinese. We plan to expand the scope of our task collection and increase the diversity of task presentation (e.g., as explored in the initial examples in Appendix G, by adding vague task descriptions and mimicking various human tones). Since some apps are difficult to operate using emulators, we also aim to design tasks that can be more easily experimented with. Additionally, we will execute experiments multiple times to ensure robustness.

Regarding our agent framework, we intend to expand the scope of our research by supporting a broader range of agents. This will include, but is not limited to, locally deployable agents, cloud-based agents, and agents capable of operating other smart devices, such as Android tablets and iOS devices.

In terms of our evaluation method, particularly for single-app success detection, we plan to introduce a more accurate approach and extend support for cross-app success detection. Furthermore, we will define a more fine-grained metric to assess how agents complete tasks, moving beyond a simple binary success signal.

## B  TASK COLLECTION

### B.1  TASK APPS

The distribution and categories of apps for the 300 single-app tasks are presented in Figure 6.

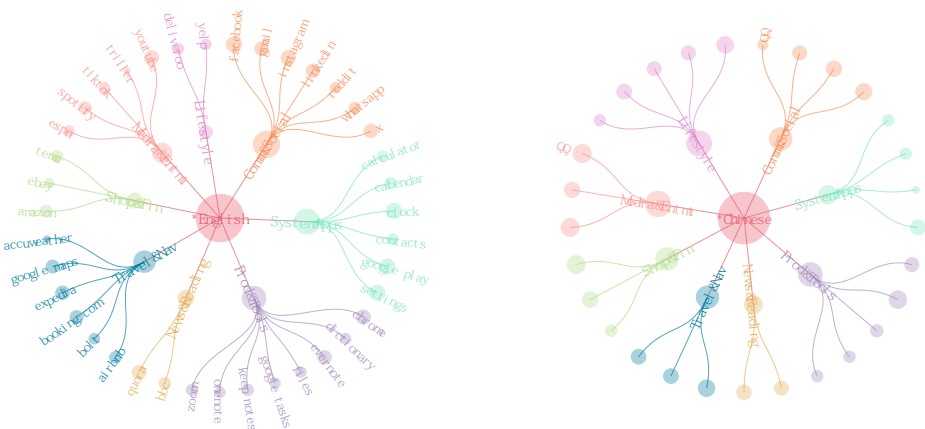

Figure 6: Distribution of apps and their categories. Left: English apps. Right: Chinese apps. The circle size is proportional to the number of tasks.

### B.2  LIST OF TASKS

The 340 tasks, encompassing single-app English, single-app Chinese, cross-app English, and cross-app Chinese categories, are detailed in Tables 4, 5, 6, 7 respectively.

### B.2.1  SINGLE-APP ENGLISH TASKS

Table 4: Single-app English tasks.

| App | Diff Level | Golden Step | Key Components | Task Description |
|---|---|---|---|---|
| Airbnb | 1 | 4 | 1, guest | Get the search results for stay for 1 adult anywhere any week. |
| Airbnb | 2 | 9 | 1, guest, wembley, stadium | Get the search results for stay tonight near 'wembley stadium' for 1 adult. |
| Airbnb | 3 | 13 | 1, guest, wembley, stadium | Get the search results for stay tonight near 'wembley stadium' for 1 adult. Add one result to wishlist. Confirm that this item is in the wishlist. |
| Amazon | 1 | 3 | sunglasses | Get the search results for 'sunglasses'. |
| Amazon | 2 | 8 | sunglasses, checkout | Get the search results for 'sunglasses'. Filter with 'kids'. Add one result to cart. Confirm that this item is in the cart. |
| Amazon | 3 | 11 | goggles, checkout | Get the search results for 'goggles'. Filter with 'adult'. Add one result to cart. Confirm that this item is in the cart. Compare with similar items. Add one of the similar items to cart. |
| BBC | 1 | 3 | save | Navigate to 'Innovation' section. Select 'Technology' tab. Open any news article. |
| BBC | 2 | 10 | save | Go to app settings. Change the Text size to 'Smaller'. Navigate to 'Innovation' section. Select 'Technology' tab. Open any news article. |
| BBC | 3 | 15 | saved, items | Go to app settings. Change the Text size to 'Larger'. Navigate to 'Business' block. Select 'Technology of Business' tab. Open any news article. Save this article. Go to Saved Items to confirm the article was added. |
| Bolt | 1 | 3 | eiffel, tower, route | Select Eiffel Tower as my destination. |
| Bolt | 2 | 6 | eiffel, tower | Select Louvre museum Paris as my pick-up location. Select Eiffel Tower as my destination. |
| Bolt | 3 | 10 | arc, de, triomphe, bolt, cash | Select Louvre museum Paris as my pick-up location. Select Eiffel Tower as my destination. Add 'Arc de Triomphe' as the final destination and Eiffel Tower as stopping point. |
| Booking | 1 | 5 | berlin | Get the search results for stays in Berlin. Select any date, rooms and guests. |
| Booking | 2 | 11 | man, cdg | Navigate to Flights section. Select any date. Choose a flight from Manchester Airport to CDG Paris. Get the search results for a round trip. |
| Booking | 3 | 15 | london, shanghai | Navigate to Flights section. Select one way flight. Choose the 1st of any month as the flight date. Get the search results from Shanghai to London. |
| Booking | 1 | 3 | settings | Navigate to app settings. |
| Booking | 2 | 7 | settings, celsius, metric | Navigate to app settings. Change Temperature to 'Degrees in Celsius'. Change Units to 'Metric (km, m)'. |
| Booking | 3 | 12 | notifications | Navigate to app settings. Change Currency to 'Pound Sterling'. Disable all notifications. |
| Calculator | 1 | 4 | 2 | Get the result for '1+1'. |
| Calculator | 2 | 10 | 2, 3 | Get the result for 'log(20)+ln(e)'. |
| Calculator | 3 | 14 | 5, 040 | Get the result for 'log(20)+ln(e)'. Clear the results. Get the result for factorial 7. |
| Calendar | 1 | 5 | halloween, 31 | Check the upcoming 31 October. Click on the event for that day. |
| Calendar | 2 | 9 | 16, haircut | Set up an all-day event titled 'Haircut' on the 16th of any month. |
| Calendar | 3 | 14 | 17, dental, check, 7, 9, pm | Set up an event titled 'Dental Check' on the 17th of any month. Set the time to from 7pm to 9pm. |
| Chrome | 1 | 3 | taylor, swift | Get the search results for Taylor Swift. |
| Chrome | 2 | 10 | taylor, swift, wiki, bookmark | Get the search results for Taylor Swift. Go to her Wikipedia page. Add it to bookmarks. Check the Bookmarks for confirmation. |
| Chrome | 3 | 16 | taylor, swift, wiki, reading | Get the search results for Taylor Swift. Go to her Wikipedia page. Add it to bookmarks. Move this bookmark to Reading List. Check the Reading List for confirmation. |
| Clock | 1 | 4 | 8 | Set an alarm for 8am. |
| Clock | 2 | 9 | 9 | Set an alarm for 9am on weekdays. |
| Clock | 3 | 15 | 11 | Set an alarm for 10am on weekdays. Disable vibration for this alarm. Set another alarm for 11am on weekends. |
| Clock | 1 | 3 | clock, london | Add current time at London (UK) to clock. |
| Clock | 2 | 6 | clock, home, hong, kong | Set Home time zone to 'Hong Kong'. |
| Clock | 3 | 12 | settings, analog | Add current time at Melbourne (Australia) to clock. Change style to Analog for clock. Change style to Analog for screen saver. |
| Contacts | 1 | 7 | agent, contact | Create a contact named 'Agent'. The phone number is +44 1234 567 890. |
| Contacts | 2 | 11 | agent, two, contact, gmail | Create a contact named 'Agent Two'. The phone number is +44 1234 567 890. The email is benchmark@gmail.com |
| Contacts | 3 | 15 | three, contact, work, gmail, huawei | Modify the last name of one of the contacts to 'Three'. Update the label for the contact's phone number to Work. Set the company to 'Huawei'. Add an email agent.benchmark.2024@gmail.com. Label the email as Work. |
| Deliveroo | 1 | 2 | mcdonald | Get the search results for McDonald's. |
| Deliveroo | 2 | 5 | fries | Get the search results for McDonald's. Enter a McDonald's restaurant. Search for fries there. |
| Deliveroo | 3 | 10 | order, fries | Get the search results for McDonald's. Enter a McDonald's restaurant. Search for fries there. Add a small fries to the basket. Add two medium fries to the basket. View the basket for confirmation. |
| Merriam-Webster | 1 | 3 | definition, dictionary, thesaurus | Look up the definition of the word 'agent'. |
| Merriam-Webster | 2 | 6 | definition, dictionary, thesaurus | Look up the definition of the word 'agent'. Switch to Thesaurus tab to find its synonyms. Click on one of its synonyms. Switch back to Dictionary tab. |
| Merriam-Webster | 3 | 9 | saved, words | Look up the definition of the word 'agent'. Switch to Thesaurus tab to find its synonyms. Click on one of its synonyms. Switch back to Dictionary tab. Save this synonym. Confirm that synonym is in the saved words. |
| ESPN | 1 | 3 | klay, thompson | Get the search results for 'Klay Thompson'. |
| ESPN | 2 | 5 | klay, thompson, like | Get the search results for 'Klay Thompson'. See all the articles. Open one of the articles. |

| App | | | Keywords | Description |
|---|---|---|---|---|
| ESPN | 3 | 11 | thompson | Get the search results for 'Klay Thompson'. See all the articles. Open one of the articles. Return to the player's search results. Select the player. Turn on player news notification. Follow the player. |
| Evernote | 1 | 3 | agent, cookbook | Create a new notebook 'Agent Cookbook'. |
| Evernote | 2 | 7 | agent, first, note | Create a new notebook 'Agent'. Create a new note in the notebook with title 'First note'. Return to the 'Agent' notebook to confirm the note. |
| Evernote | 3 | 13 | agent2, first, note, hello, world, test | Create a new notebook 'Agent2'. Create a new note in the notebook. Write content 'Hello World!' and title 'First note'. Create a new tag 'test'. Apply the tag 'test' to the note. Save the note. Return to the 'Agent2' notebook. |
| Evernote | 1 | 3 | literature, review | Create a new task 'Literature Review'. |
| Evernote | 2 | 6 | paper, writing, to-morrow | Create a new task 'Paper Writing'.Set the due date to tomorrow. Navigate to the Tasks tab for confirmation. |
| Evernote | 3 | 12 | recurring, main, git, repo | Create a new task 'Maintain Git Repo'.Set it to repeat daily. Navigate to the Tasks tab. Apply the recurring tasks filter. Confirm that task exists. |
| Expedia | 1 | 4 | rome, 2 | Check stays in Rome. The dates do not matter. Get the search results for 1 room and 2 people. |
| Expedia | 2 | 8 | paris, 25, 28, 2 | Check stays in Paris. Choose from 25th to 28th any month. Get the search results for 1 room for 2 people. |
| Expedia | 3 | 12 | hong, kong, 25, 28, 2 | Check stays in Hong Kong. Choose from 25th to 28th any month. Get the search results for 1 room for 2 people. Filter hotels with parking. |
| Expedia | 1 | 7 | paris, 25, 28 | Check things to do in Paris. Get the search results for 25th to 28th of any month. |
| Expedia | 2 | 11 | rome, 26, 29 | Check things to do in Rome. Get the search results for 26th to 29th of any month. Save it to my trips. |
| Expedia | 3 | 13 | paris, 25, 28, save | Check things to do in Paris. Get the search results for 25th to 28th of any month. Save it to my trips. Confirm that by checking the saved Paris trip. |
| Facebook | 1 | 4 | hello, world | Create a new post saying 'Hello World!'. Post it. |
| Facebook | 2 | 8 | morning | Create a new Public post saying 'Morning!'. Change to black background. Post it. |
| Facebook | 3 | 11 | bonne, nuit, eiffel, tower, paris | Create a new Public post saying 'Bonne Nuit'. Add the location as Eiffel Tower. Post it. |
| Facebook | 1 | 2 | settings | Navigate to settings. |
| Facebook | 2 | 6 | birthday, notifications | Navigate to settings. Disallow notifications for Birthdays. |
| Facebook | 3 | 11 | notifications, email, sms | Navigate to settings. Disallow notifications for Marketplace from Email and SMS. Disallow notifications for Memories from Email and SMS. |
| Files | 1 | 3 | dcim | Go to the 'DCIM' folder in the internal storage. |
| Files | 2 | 7 | dcim, agent, created | Go to the 'DCIM' folder in the internal storage. Create a subfolder named 'Agent created'. |
| Files | 3 | 18 | agent, created | Go to the 'DCIM' folder in the internal storage. Create a subfolder named 'Agent created 2'. Create another subfolder named 'Agent created 3'. Then move the folder 'Agent created 2' into the 'Documents' folder in the internal storage. |
| Gmail | 1 | 5 | paper | Draft an email to agent.benchmark.2024@gmail.com asking them about their new paper. |
| Gmail | 2 | 9 | paper | Send an email to agent.benchmark.2024@gmail.com asking them about their new paper. Navigate to the Sent tab. Check the email details for confirmation after sending. |
| Gmail | 3 | 11 | paper, scheduled | Draft an email to agent.benchmark.2024@gmail.com asking them about their new paper. Schedule it to be sent tomorrow morning. Navigate to the Scheduled tab. Check the email details for confirmation for confirmation after scheduling. |
| Gmail | 1 | 2 | settings | Navigate to settings. |
| Gmail | 2 | 6 | gmail, notification | Navigate to settings. Check current setting for notifications. Turn off notification for Attachments. |
| Gmail | 3 | 11 | inbox | Navigate to settings. Check current setting for notifications. Turn off notification for Miscellaneous. Disable 'notification dot'. Return to Inbox. |
| Google Maps | 1 | 3 | hotel | Get the search results for nearby hotel rooms. |
| Google Maps | 2 | 7 | hotel, 4 | Get the search results for nearby hotel rooms. Filter the results to show only those that can accommodate 4 adults. |
| Google Maps | 3 | 10 | hotel, 4 | Get the search results for nearby hotel rooms. Filter the results to show only those that can accommodate 4 adults. Further filter the results with ratings higher than 4. |
| Google Maps | 1 | 3 | gas, station | Get the search results for nearby gas stations. |
| Google Maps | 2 | 6 | your, location | Get the search results for a nearby gas station that is open now. Get a driving route to it. |
| Google Maps | 3 | 12 | your, location, mc-donald | Get the search results for a nearby gas station that is open now. Get a driving route with the gas station as the first stop. Set McDonald's as the final destination. |
| Google Play | 1 | 3 | whatsapp | Get the search results for WhatsApp. |
| Google Play | 2 | 7 | review | Get the search results for Facebook. Leave a 5-star review on its app store page. |
| Google Play | 3 | 14 | whatsapp, review, recent | Get the search results for WhatsApp. Leave a 5-star review on its app store page. Sort the reviews by most recent. |
| Google Play | 1 | 3 | settings | Check the details of General settings. |
| Google Play | 2 | 6 | settings | Check the details of General settings. Switch to dark theme. |
| Google Play | 3 | 10 | notification, settings | Check the details of General settings. Turn off all notifications. Confirm that all notification settings for this device are off. |
| Google Tasks | 1 | 3 | work, tasks | Create a new list 'Work'. |

| App | | | | |
|---|---|---|---|---|
| Google Tasks | 2 | 6 | tasks, buy, groceries, weekend | Create a new list 'Weekend'. Add new task 'Buy groceries'. |
| Google Tasks | 3 | 10 | tasks, 12, visa, travel | Create a new list 'Travel'. Add new task 'Visa'. Set date to the 12th of any month. |
| Instagram | 1 | 4 | messi, posts | Get the search results for 'Messi'. |
| Instagram | 2 | 6 | cristiano, following, message | Get the search results for 'Cristiano Ronaldo'. Follow one account. |
| Instagram | 3 | 10 | minions, notifications, all | Get the search results for 'Minions'. Follow one account. Set to get all notifications when they goes live. Turn on notifications for their posts. |
| Instagram | 1 | 3 | edit, profile | Navigate to the page to edit my profile. |
| Instagram | 2 | 10 | it | Navigate to the page to edit my profile. Add bio 'Hello World!'. Change pronouns to 'it'. |
| Instagram | 3 | 17 | account, privacy, private | Navigate to the page to edit my profile. Add link 'https://github.com'. Change gender to Custom 'Them'. Switch to private account. |
| Keep Notes | 1 | 2 | note, hello, 1 | Create a new note. Write 'Hello this is a note1' in the content. |
| Keep Notes | 2 | 5 | note, agent, hello, 2 | Create a new note. Write 'Hello this is a note2' in the content. Write 'Written by Agent2' as the note title. |
| Keep Notes | 3 | 11 | agent, python, java | Create a new checklist. Add two items 'Learn Python' and 'Learn Java'. Write 'Goal of agent' as the checklist title. Label this checklist as 'Agent'. |
| LinkedIn | 1 | 4 | following, openai | Get the search results for 'OpenAI'. Follow their page. |
| LinkedIn | 2 | 7 | join, huawei, groups | Get the search results for 'Huawei'. Follow their page. Filter the search results to Groups. Join one of the Huawei groups. |
| LinkedIn | 3 | 12 | huawei, reposted, hkrc | Get the search results for 'Huawei HKRC'. Follow their page. Leave a 'Cheers!' comment on one of its posts. Like the post. Repost the post instantly. View the repost to confirm. |
| LinkedIn | 1 | 4 | engineer, jobs | Get the search results for 'Engineer' job. |
| LinkedIn | 2 | 7 | engineer, jobs, spain | Get the search results for 'Engineer' job in Spain. |
| LinkedIn | 3 | 10 | engineer, jobs, spain, saved | Get the search results for 'Engineer' jobs in Spain. Save one of the jobs. Confirm it is saved in My Jobs. |
| Microsoft OneNote | 1 | 4 | agent, benchmark | Create a new page with title 'Benchmark' and content 'Test Agent'. |
| Microsoft OneNote | 2 | 7 | benchmark2, appagent, mobile, agent | Create a new page with title 'Benchmark2' and content TODO 'AppAgent' and 'Mobile Agent'. |
| Microsoft OneNote | 3 | 11 | prompts, test, pages | Create a new notebook 'test'. Create a new section 'prompts' in 'test' notebook. Enter section 'prompts' for confirmation. |
| Quora | 1 | 3 | search, openai | Get the search results for 'OpenAI'. |
| Quora | 2 | 6 | search, openai | Get the search results for 'OpenAI'. Filter to show only questions. |
| Quora | 3 | 11 | worth, thinking | Get the search results for 'OpenAI'. Filter to show only questions. Select one question or answer from the results to see more details. Add a comment 'Worth thinking" to the answer. |
| Quora | 1 | 3 | following | Discover any Space. Follow that space. |
| Quora | 2 | 7 | questions, follow, answer | Discover any Space. Follow that space. Go to questions in the space. Filter unanswered questions. Follow one question. |
| Quora | 3 | 11 | following, ask | Discover any Technology Spaces. Follow that space. Also follow one of the suggested spaces. Turn off notification for the suggested space. Follow one of the contributors of the suggested space. |
| Reddit | 1 | 4 | worldnews, joined | Get the search results for 'r/worldnews'. Join the group. |
| Reddit | 2 | 8 | premierleague, liverpool | Get the search results for 'r/PremierLeague'. Filter posts for Liverpool. Join the group. Click on one of the posts. |
| Reddit | 3 | 11 | blackmythwukong | Get the search results for 'r/BlackMythWukong'. Join the group. Set community alerts to frequent. Click on one of the posts. |
| Settings | 1 | 3 | screen, timeout | Check the current screen timeout. |
| Settings | 2 | 5 | screen, timeout, 5, min | Check the current screen timeout. Set it to 5 minutes. |
| Settings | 3 | 10 | dark, theme | Check the current screen timeout. Set it to 10 minutes. Then turn the dark theme on. |
| Settings | 1 | 3 | notification, history | Go to notification settings. Turn on Notification History. |
| Settings | 2 | 6 | store, notifications | Go to notification settings. Turn off the notification from Google Play Store. |
| Settings | 3 | 11 | instagram, storage | Go to notification settings. Turn off the 'Alerts' and 'Likes' notification from Instagram. Clear the cache from storage. |
| Spotify | 1 | 3 | taylor, swift | Get the search results for the artist Taylor Swift. |
| Spotify | 2 | 6 | taylor, swift | Get the search results for the artist Taylor Swift. Enter her artist page. Shuffle play her playlist. |
| Spotify | 3 | 15 | agent, playlist, love, story, the, scientist | Get the search results for the song 'Love Story' by Taylor Swift. Add this song to the new playlist namely 'Agent Playlist'. Then add another song 'The Scientist' by Coldplay to the same playlist. Check the playlist for confirmation. |
| Temu | 1 | 3 | gaming, headset | Get the search results for gaming headset. |
| Temu | 2 | 7 | gaming, headset, checkout | Get the search results for gaming headset. Sort the result by the lowest price to highest. Add one to my shopping cart. Confirm that this item is in the cart. |
| Temu | 3 | 12 | checkout | Get the search results for gaming mouse. Filter items priced above 10. Add one to cart. Confirm that this item is in the cart. |
| TikTok | 1 | 3 | cat | Get the search results for videos about pet cats. |
| TikTok | 2 | 8 | cute, cat | Get the search results for videos about pet cats. Comment on a video with 'Such a cute cat.' |
| TikTok | 3 | 13 | cat | Get the search results for videos about pet cats. Comment on a video with 'Such a cute cat.' Swipe through another two videos and like them. |
| WhatsApp | 1 | 5 | hi, you, message | Send a message 'Hi' to myself. |
| WhatsApp | 2 | 9 | mark, bench, contact | Add new contact with the name 'Mark Bench' and (+44)7437321230. |
| WhatsApp | 3 | 13 | smart, agent, hi, message | Add new contact with the name 'Smart Agent' and (+44)7746953749. Send a message 'Hi' to 'Smart Agent'. |

| App | | | | |
|---|---|---|---|---|
| X | 1 | 3 | agent, post, 1 | Draft a post with the content 'Written by Agent1'. |
| X | 2 | 9 | agent, post, 2, reply | Create a post with the content 'Written by Agent2'. Tag '#animalcrossing'. Post it. Check it from the profile. |
| X | 3 | 12 | agent, post, 3, reply, amazing | Create a post with the content 'Written by Agent3'. Tag '#animalcrossing'. Post it. Check it from the profile. Then Like it. Reply to it with 'Amazing post'. |
| X | 1 | 5 | mayday, following | Search for the account @Mayday EN. Follow it. |
| X | 2 | 8 | nintendo, super, mario | Search for the account @Nintendo. Follow it. Search its post about 'Super Mario'. |
| X | 3 | 15 | animal, crossing, timmy, tommy, post | Search for the account @animalcrossing. Follow it. Search its post about 'Timmy and Tommy'. Repost one result. Check it from the profile for confirmation. |
| Yelp | 1 | 2 | restaurants | Get the search results for nearby restaurants. |
| Yelp | 2 | 6 | restaurants, chinese | Get the search results for nearby restaurants. Filter to include only Chinese restaurants that offer takeout. Sort them by distance. |
| Yelp | 3 | 10 | review | Get the search results for nearby restaurants. Filter to include only Chinese restaurants that offer takeout. Sort them by distance. Select one result. Filter for 5-star reviews. |
| YouTube | 1 | 4 | tesla, subscribed | Get the search results for the channel '@Tesla'. Subscribe to the channel. |
| YouTube | 2 | 8 | subscribed | Get the search results for the channel '@BMW'. Subscribe to the channel. Get the search results for the channel '@Mercedes'. Subscribe to the channel. |
| YouTube | 3 | 12 | all, subscriptions, microsoft, google | Get the search results for the channel '@Google'. Subscribe to the channel. Get the search results for the channel '@Microsoft'. Subscribe to the channel. Navigate to the Subscriptions tab. Show all subscriptions. Sort the subscriptions from A to Z. |
| YouTube | 1 | 4 | lebron | Get the search results for videos about LeBron James. |
| YouTube | 2 | 10 | lebron, views | Get the search results for videos about LeBron James. Filter videos under 4 minutes. |
| YouTube | 3 | 14 | comment | Get the search results for videos about LeBron James. Filter videos under 4 minutes. Select any one of the results. Leave a comment 'great performance!'. |
| Zoom | 1 | 5 | smartphone, agent, benchmark | Schedule a meeting titled 'Smartphone Agent Benchmark'. Use personal meeting ID. |
| Zoom | 2 | 9 | smartphone, agent, benchmark | Schedule a meeting titled 'Smartphone Agent Benchmark'. Use personal meeting ID. Change the timezone to Hawaii. Repeat the meeting every day. |
| Zoom | 3 | 14 | smartphone, agent, benchmark | Schedule a meeting titled 'Smartphone Agent Benchmark'. Use personal meeting ID. Change the timezone to Hawaii. Repeat the meeting every day. Disable waiting room. Turn on host and participant video. |

### B.2.2 SINGLE-APP CHINESE TASKS

Table 5: Single-app Chinese tasks.

| App | Diff Level | Golden Step | Key Components | Task Description |
|---|---|---|---|---|
| 支付宝 | 1 | 3 | 汇率换算 | 搜索汇率换算。 |
| 支付宝 | 2 | 9 | 港币, 欧元 | 进入汇率换算小程序，查看港币兑欧元汇率。 |
| 支付宝 | 3 | 13 | 港币, 欧元, 100 | 进入汇率换算小程序，查看港币兑欧元汇率。计算100港币 可以兑换多少欧元。 |
| 哔哩哔哩 | 1 | 3 | 游戏解说, 搜索 | 搜索关键词"游戏解说" |
| 哔哩哔哩 | 2 | 7 | 简介, 评论 | 搜索关键词"游戏解说"，对搜索结果按播放量排序，选择一个视频。 |
| 哔哩哔哩 | 3 | 12 | 哈哈, 发布 | 搜索关键词"游戏解说"，对搜索结果按播放量排序，选择一个视频，并对它点赞、收藏，编辑评论"哈哈"。 |
| 哔哩哔哩 | 1 | 3 | 公开课, 学科课程 | 查看公开课分区，展示学科课程栏目的内容。 |
| 哔哩哔哩 | 2 | 6 | 公开课, 学科课程, 收藏最多 | 查看公开课分区，展示学科课程栏目的内容，查看交叉学科相关的视频，并按照收藏数排序。 |
| 哔哩哔哩 | 3 | 10 | 好好好, 发布 | 查看公开课分区，展示学科课程栏目的内容，查看交叉学科相关的视频，并按照收藏数排序，在收藏量最高的视频下面发送评论"好好好"。（停留在评论发送页面） |
| 哔哩哔哩 | 1 | 1 | 消息, 聊天列表 | 浏览个人消息通知。 |
| 哔哩哔哩 | 2 | 5 | 动态 | 浏览个人消息通知，挑选一个聊天，查看好友动态。 |
| 哔哩哔哩 | 3 | 9 | 评论, 哈哈 | 浏览个人消息通知，挑选一个聊天，查看好友动态，点赞好友的一条动态，编辑评论"哈哈"。 |
| 菜鸟裹裹 | 1 | 2 | 地址 | 在我的页面中查看"收货地址" |
| 菜鸟裹裹 | 2 | 7 | 收货地址, 收件人, 张三 | 在我的页面中选择收货地址，选择添加收货地址，姓名输入张三，手机号输入123456789 |
| 菜鸟裹裹 | 3 | 13 | 我的地址, 张三, 九龙城区 | 在我的页面中选择收货地址，选择添加收货地址，姓名输入张三，手机号输入123456789，详细地址输入无，地区选择九龙的九龙城区，然后保存这个收货地址。 |
| 万年历 | 1 | 2 | 黄历, 运程 | 查看今天的黄历，然后查看今天的运程。 |
| 万年历 | 2 | 5 | 宜出行 | 查看今天的黄历，然后查看今天的运程。然后在"工具"页面中打开"择吉日"，然后查看"出行"的吉日。 |
| 万年历 | 3 | 9 | 宜出行, 星期日, 星期六 | 查看今天的黄历，然后查看今天的运程。然后在"工具"页面中打开"择吉日"，然后查看"出行"的吉日，然后将开始日期调整为下个月，并且设置"只看周末"。 |
| 时钟 | 1 | 4 | 重复, 一 | 新建闹钟，设置在每个星期一重复，然后停止操作 |
| 时钟 | 2 | 7 | 闹钟, 一个闹钟 | 新建闹钟，设置在每个星期一重复，修改标签（备注）或闹钟名为"一个闹钟"，然后停止操作 |
| 时钟 | 3 | 11 | 闹钟, 响铃 | 新建闹钟，设置在每个星期一重复，修改标签（备注）或闹钟名为"一个闹钟"，更换一个铃声音乐，保存闹钟 |

| 电信 | 1 | 3 | 查用量, 查费用, 查积分 | 进入查询办理，查看自己的积分、费用与用量（不分先后） |
| 电信 | 2 | 6 | 支付方式, 确认交易 | 进入查询办理，查看自己的积分、费用与用量（不分先后），随后查看自己的话费账单并充值10元额度（停在选择支付方式界面） |
| 电信 | 3 | 9 | 中国电信, 立即支付 | 进入查询办理，查看自己的积分、费用与用量（不分先后），随后查看自己的话费账单并充值10元额度（停在选择支付方式界面），选择微信支付选项并停在立即支付界面前，不要付钱 |
| 电信 | 1 | 2 | 我的, 账户信息 | 进入用户中心的个人信息页面。 |
| 电信 | 2 | 5 | 5G开启, 梦想加速 | 进入用户中心的个人信息页面，设置电信签名为5G开启，梦想加速。 |
| 电信 | 3 | 9 | 操作成功 | 进入用户中心的个人信息页面（如果需要登录则登录账号），设置电信签名为5G开启，梦想加速，最后从本地相册选择一张图片设置为个人头像。 |
| 抖音 | 1 | 3 | 张国伟, 关注 | 搜索博主张国伟 |
| 抖音 | 2 | 6 | 张国伟 | 搜索博主张国伟，查看博主主页并查看其中一条视频，点赞该视频 |
| 抖音 | 3 | 13 | 张国伟, 已关注, 私信 | 搜索博主张国伟，查看博主主页并查看其中一条视频，点赞该视频任意一条评论，并关注主播 |
| 抖音 | 1 | 3 | 已关注 | 进入关注界面，查看关注博主的主页 |
| 抖音 | 2 | 8 | 评论, 期待下一条视频 | 进入关注界面，查看关注博主的主页，观看关注的博主发布的视频并发表评论"期待下一条视频" |
| 抖音 | 3 | 11 | 收藏, 视频 | 进入关注界面，查看关注博主的主页，观看关注的博主发布的视频并发表评论"期待下一条视频"，收藏该视频并查看我的收藏夹 |
| 饿了么 | 1 | 2 | 美食外卖, 快餐便当 | 进入美食外卖，选择快餐便当 |
| 饿了么 | 2 | 5 | 点餐 | 进入美食外卖，选择快餐便当，按照好评优先排序，选择一家店铺查看详情 |
| 饿了么 | 3 | 10 | 加入购物车, 详情 | 进入美食外卖，选择快餐便当，按照好评优先排序，选择一家店铺查看详情，浏览评价，查看商家信息或品牌故事，返回点餐，选择任意餐食查看详情 |
| 饿了么 | 1 | 2 | 搜索 | 进入甜品饮品板块，进入搜索界面 |
| 饿了么 | 2 | 6 | 生椰拿铁, 温度, 数量 | 进入甜品饮品板块，进入搜索界面，搜索瑞幸咖啡，选择推荐的店铺查看详情，选择生椰拿铁规格 |
| 饿了么 | 3 | 9 | 确认订单 | 进入甜品饮品板块，进入搜索界面，搜索瑞幸咖啡，选择推荐的店铺查看详情，选择生椰拿铁规格，选择冰，不额外加糖其余默认，加入购物车并去结算，不要提交订单 |
| 高德地图 | 1 | 3 | 美食 | 搜索附近的餐厅。 |
| 高德地图 | 2 | 6 | 导航, 路线 | 搜索附近的餐厅，按照好评优先排序，并点击列表里面的一家餐厅。 |
| 高德地图 | 3 | 10 | 公交, 推荐 | 搜索附近的餐厅，按照好评优先排序，并点击列表里面的一家餐厅，查看用户详情然后点击"路线"来规划乘公交从当前位置到达该餐厅的路线。 |
| 高德地图 | 1 | 3 | 超市, 位置距离, 推荐排序 | 查找附近的超市。 |
| 高德地图 | 2 | 6 | 我的位置, 驾车, 开始导航 | 查找附近的超市，点击其中一个超市，点击"路线"来规划路线。 |
| 高德地图 | 3 | 10 | 我的位置, 加油站, 驾车, 开始导航 | 查找附近的超市，点击其中一个超市，点击"路线"来规划路线，然后查看周围是否有加油站，选择一个合适的加油站作为经停点并确认最佳的驾车路线。 |
| 高德地图 | 1 | 3 | 著名景区, 位置距离, 推荐排序 | 进入附近界面，点击旅游，进入著名景区界面 |
| 高德地图 | 2 | 6 | 太阳岛风景区 | 进入附近界面，点击旅游，进入著名景区界面，切换到哈尔滨。将推荐排序换为好评优先，选择太阳岛风景区 |
| 高德地图 | 3 | 9 | 提交订单 | 进入附近界面，点击旅游，进入著名景区界面，切换到哈尔滨。将推荐排序换为好评优先，选择太阳岛风景区。点击实用信息查看相关消息，购买太阳岛寒地野生动物园的票，进入订单界面。 |
| 航旅纵横 | 1 | 6 | 北京, 深圳 | 搜索某个月份16号北京到深圳的机票 |
| 航旅纵横 | 2 | 11 | 首都大兴, 宝安 | 搜索某个月份16号北京到深圳的机票，筛选起飞时段为12:00-18:00并规定舱位为经济舱 |
| 航旅纵横 | 3 | 15 | 经济舱, 大兴 | 搜索某个月份16号北京到深圳的机票，筛选起飞时段为12:00-18:00并规定舱位为经济舱，从中选择一班飞机，并查看退改签详细信息 |
| 航旅纵横 | 1 | 5 | 深圳, 筛选, 酒店 | 进入酒店信息界面，选择某个月份16日-18日深圳的酒店预定 |
| 航旅纵横 | 2 | 10 | 星级, 好评优先, 筛选 | 进入酒店信息界面，选择某个月份16日-18日深圳的酒店预定，筛选品牌为"全部经济酒店"，推荐顺序为好评优先 |
| 航旅纵横 | 3 | 18 | 封面 | 进入酒店信息界面，选择某个月份16日-18日深圳的酒店预定，筛选品牌为"全部经济酒店"，推荐顺序为好评优先，选择位置在机场附近的一家酒店 |
| 好大夫 | 1 | 2 | 订单列表, 全部 | 查看个人的全部订单 |
| 好大夫 | 2 | 9 | 神经外科 | 查看个人的全部订单。回到首页在专家门诊中查找神经外科有关的医生 |
| 好大夫 | 3 | 14 | 预约 | 查看个人的全部订单。回到首页在专家门诊中查找神经外科有关的医生，选择一位医生申请服务并预约挂号，时间任意选择。 |
| 好大夫 | 1 | 3 | 失眠, 介绍 | 进入知识界面，选择失眠选项，查看介绍。 |
| 好大夫 | 2 | 7 | 北京, 推荐, 医院 | 进入知识界面，选择失眠选项，查看介绍。点击推荐医院，更改地区为北京全部。 |
| 好大夫 | 3 | 11 | 信息 | 进入知识界面，选择失眠选项，查看介绍。点击推荐医院，更改地区为北京全部。点击排行第一的医院并选择一位专家将其加入备选，并点击预约挂号查看预约信息。 |
| 华为浏览器 | 1 | 3 | 首页 | 搜索bilibili.com. |
| 华为浏览器 | 2 | 7 | 潜艇伟伟迷 | 搜索bilibili.com，在网站中搜索"潜艇伟伟迷"并进入UP主列表 |
| 华为浏览器 | 3 | 12 | 书签, 潜艇伟伟迷 | 搜索bilibili.com，在网站中搜索"潜艇伟伟迷"并进入UP主列表，添加该页面为书签，并确认在书签管理中存在该书签 |

| | | | | |
|---|---|---|---|---|
| 华为浏览器 | 1 | 3 | 淘宝 | 浏览电商网站taobao.com |
| 华为浏览器 | 2 | 6 | 商品, 华为, 搜索 | 浏览电商网站taobao.com，搜索华为mate60pro |
| 华为浏览器 | 3 | 10 | 全部, 评 | 浏览电商网站taobao.com，搜索华为mate60pro，选择按销量排序，点击任意商品并查看评价 |
| 京东 | 1 | 3 | 索尼, WH | 搜索索尼 WH-1000XM4 头戴式耳机。 |
| 京东 | 2 | 9 | 索尼, WH | 搜索索尼 WH-1000XM4 头戴式耳机，筛选出价格低于2000元的结果。 |
| 京东 | 3 | 14 | 索尼, WH, 结算 | 搜索索尼 WH-1000XM4 头戴式耳机，筛选出价格低于2000元的结果，查看商品详情页，加入购物车，查看购物车以确认。 |
| 京东 | 1 | 3 | 地毯 | 搜索一款地毯 |
| 京东 | 2 | 7 | 购物车, 详情 | 搜索一款地毯，筛选销量最多的结果，查看商品详情，如果尚未收藏则收藏商品。 |
| 京东 | 3 | 12 | 购物车, 结算 | 搜索一款地毯，筛选销量最多的结果，查看商品详情，如果尚未收藏则收藏商品，并查看优惠详情，然后选择商品，设置数量为2，加入购物车 |
| 美团 | 1 | 3 | 螺蛳粉 | 搜索一家附近的螺蛳粉店。 |
| 美团 | 2 | 7 | 螺蛳粉 | 搜索一家附近的螺蛳粉店，并查看店内评分、评论以及菜品。 |
| 美团 | 3 | 13 | 螺蛳粉, 订单 | 搜索一家附近的螺蛳粉店，并查看店内评分、评论以及菜品，然后下单一份螺蛳粉(停在支付前的订单界面)。 |
| 美团 | 1 | 3 | 健身 | 查找附近的一家健身馆。 |
| 美团 | 2 | 6 | 教练印象 | 查找附近的一家健身馆，查看一名高评价的健身教练介绍。 |
| 美团 | 3 | 9 | 订单, 支付 | 查找附近的一家健身馆，查看一名高评价的健身教练介绍并且购买票。（停留在订单界面） |
| 美团 | 1 | 2 | 浏览记录, 商户 | 查看浏览过的商品或商家 |
| 美团 | 2 | 8 | 评价 | 为浏览过的商品或商家并撰写评论"Hello world" |
| 美团 | 3 | 13 | 评价详情 | 为浏览过的商品或商家撰写评论"Hello world"，回到个人主页，查看自己刚刚发布的评论 |
| QQ音乐 | 1 | 3 | 周杰伦 | 搜索歌手周杰伦 |
| QQ音乐 | 2 | 6 | 周杰伦, 评论 | 搜索歌手周杰伦，打开他的一个专辑然后查看他的专辑评论。 |
| QQ音乐 | 3 | 9 | 周杰伦, 评论 | 搜索歌手周杰伦，查看他的专辑然后发表评论"hello world"到评论区。 |
| QQ音乐 | 1 | 3 | 个人资料 | 查看个人资料 |
| QQ音乐 | 2 | 6 | 留言 | 查看个人资料，然后查看自己发表过的评论 |
| QQ音乐 | 3 | 9 | 留言 | 查看个人资料，然后查看自己发表过的评论，并回复"好好好"给自己的评论 |
| QQ音乐 | 1 | 2 | 巅峰榜, 热歌榜 | 浏览音乐排行榜 |
| QQ音乐 | 2 | 5 | 热歌榜, 我喜欢 | 浏览音乐排行榜，找到排名前五的歌曲。将前三名添加至我喜欢。 |
| QQ音乐 | 3 | 9 | 微信好友, 分享 | 浏览音乐排行榜，找到排名前五的歌曲，将前五名添加至我喜欢并将第五名分享给微信好友。（停留在分享界面） |
| 去哪儿 | 1 | 4 | 国内, 海外, 入住, 离店, 东京 | 在首页选择民宿 客栈，选择海外，城市选择东京 |
| 去哪儿 | 2 | 7 | 海外, 入住, 离店, 6日, 19日 | 在首页选择民宿 客栈，选择海外，城市选择东京，入住时间选择为某个月份的6日，离店时间选择为某个月份的19日 |
| 去哪儿 | 3 | 12 | 东京, 搜索, 2人, 2床, 1居 | 在首页选择民宿 客栈，选择海外，城市选择东京，入住时间选择为某个月份的6日，离店时间选择为某个月份的19日，入住条件中总人数选择2人，床铺数选择2床，居室数选择1居 |
| 去哪儿 | 1 | 2 | 机票, 单程, 往返, 乘机人 | 在首页选择机票，选择往返界面 |
| 去哪儿 | 2 | 6 | 机票, 往返, 深圳, 南京 | 在首页选择机票，选择往返界面，出发城市选择深圳，抵达城市选择南京 |
| 去哪儿 | 3 | 12 | 深圳, 南京, 9天, 去程, 返程 | 在首页选择机票，选择往返界面，出发城市选择深圳，抵达城市选择南京，出发日期定为某个月份9号，返回日期定为某个月份17号，点击搜索。 |
| 设置 | 1 | 3 | 开发, 选项, 内存 | 点击"系统与更新"，进入开发者选项 |
| 设置 | 2 | 6 | 内存使用量, 小时 | 点击"系统与更新"，进入开发者选项，点击内存，将选项换成1天后查看各个应用的内存使用量 |
| 设置 | 3 | 9 | 内存使用量, 详细信息 | 点击"系统与更新"，进入开发者选项，点击内存，将选项换成1天后查看各个应用的内存使用量，然后进入其中两个应用的内存使用量页面分别查看。 |
| 淘宝 | 1 | 3 | 华为P50 | 搜索"华为P50"。 |
| 淘宝 | 2 | 6 | 评价 | 搜索"华为P50"，点击一件商品查看详情，下拉查看评价。 |
| 淘宝 | 3 | 10 | 关注成功 | 搜索"华为P50"，点击一件商品查看详情，下拉查看评价，收藏此商品，进入店铺，关注此店铺。 |
| 淘宝 | 1 | 3 | 抹茶旦旦 | 搜索《抹茶旦旦》周边。 |
| 淘宝 | 2 | 7 | 抹茶旦旦 | 搜索《抹茶旦旦》周边，查看并选择一款拼图，加入购物车。 |
| 淘宝 | 3 | 10 | 鬼灭之刃, 分享给好友 | 搜索《鬼灭之刃》动漫周边，查看并选择一款T恤，加入购物车，收藏商品，并且分享给好友（结束在分享页面前就可以）。 |
| 淘宝 | 1 | 3 | 男士洗发水 | 搜索男士洗发水。 |
| 淘宝 | 2 | 7 | 加入购物车 | 搜索男士洗发水，查看商品详情，然后加入购物车。 |
| 淘宝 | 3 | 10 | 已关注, 已收藏, 加入购物车, 关注成功 | 搜索男士香水，查看商品详情并收藏，然后加入购物车，并将香水店铺放入店铺关注列表。 |
| 腾讯文档 | 1 | 3 | 请输入标题, 请输入正文 | 新建一个空白文档 |
| 腾讯文档 | 2 | 6 | 标题, 你好 | 新建一个空白文档，设置标题为"标题"，设置正文为"你好" |
| 腾讯文档 | 3 | 10 | 今天, 标题 | 新建一个空白文档，设置标题为"标题"，设置正文为"你好"，查看文档大纲后返回至主页 |
| 腾讯会议 | 1 | 3 | 预定会议, 入会密码 | 预定一个常规会议，开启入会密码 |
| 腾讯会议 | 2 | 9 | 会议水印 | 预定一个常规会议，开启入会密码并将入会密码设置为"111111"后进入设置会议水印界面 |

| 腾讯会议 | 3 | 12 | 会议详情 | 预定一个常规会议，开启入会密码并将入会密码设置为"111111"，开启会议水印后完成设置。 |
|---|---|---|---|---|
| 今日头条 | 1 | 4 | 科技新闻 | 搜索关键词"科技新闻"。 |
| 今日头条 | 2 | 7 | 分享 | 搜索关键词"科技新闻"，查看前2条新闻。 |
| 今日头条 | 3 | 10 | 分享, 收藏 | 搜索关键词"科技新闻"，查看前2条新闻，点赞并收藏。 |
| 今日头条 | 1 | 2 | 设置, 编辑资料, 账号与安全 | 打开设置。 |
| 今日头条 | 2 | 5 | 历史, 编辑, 关注 | 打开设置，清理缓存文件并查看历史记录。 |
| 今日头条 | 3 | 10 | 消息, 私信, 评论 | 打开设置，清理缓存文件，查看历史记录并使用"一键清空"功能清空历史记录，然后浏览消息私信。 |
| 微信 | 1 | 2 | | 进入朋友圈的页面。 |
| 微信 | 2 | 9 | 今天真开心 | 发表一则朋友圈，从相册中任意选一张图，并配文"今天真开心"。 |
| 微信 | 3 | 15 | 今天天气真好, 希望大家点赞 | 发表一则朋友圈，从相册中任意选一张图，并配文"今天天气真好"。点赞这条朋友圈，并评论"希望大家点赞"。 |
| 微博 | 1 | 3 | 周末去哪儿玩, 综合 | 搜索"周末去哪儿玩" |
| 微博 | 2 | 6 | 深圳美食, 综合, 已关注 | 搜索"深圳美食"，关注两个发微博的用户。 |
| 微博 | 3 | 13 | 这个地方看起来不错 | 搜索"周末去哪儿玩"，关注两个发微博的用户并转发其中一条微博，附上评论"这个地方看起来不错"（并且停留在发送页面） |
| 微博 | 1 | 3 | 李子柒 | 搜索用户名为"李子柒"的用户 |
| 微博 | 2 | 7 | 李子柒, 全部微博 | 搜索用户名为"李子柒"的用户，关注该用户，浏览其最新发布的一条微博 |
| 微博 | 3 | 10 | 腥味猫罐, 发送成功 | 搜索用户名为"腥味猫罐"的用户，关注该用户，浏览其发布的一条微博，并在这条微博下进行评论："hello world"（并且发送出去） |
| 小红书 | 1 | 3 | 减肥食谱, 全部, 用户, 筛选 | 搜索一种名为"减肥食谱"的笔记。 |
| 小红书 | 2 | 6 | 减肥食谱, 全部, 用户, 筛选 | 搜索一种名为"减肥食谱"的笔记，按照热度排序。 |
| 小红书 | 3 | 11 | 发送, 很有用, 谢谢 | 搜索一种名为"减肥食谱"的笔记，按照热度排序，观看其中一个热度最高的笔记，点赞该笔记，收藏该笔记，然后编辑评论"很有用，谢谢"，停留在评论发送页面不要发送。 |
| 小红书 | 1 | 2 | 关注, 说点什么 | 在首页切换至视频类别，观看一条推荐的视频笔记。 |
| 小红书 | 2 | 5 | 评论 | 在首页切换至视频类别，观看一条推荐的视频笔记点赞该视频，关注视频发布者，并查看视频评论区。 |
| 小红书 | 3 | 10 | 已关注, 收藏成功 | 在首页观看一条推荐的视频笔记，点赞该视频，关注视频发布者，并查看视频评论区，随后查看该用户的其他笔记，并收藏其中两篇笔记。 |
| 协和医院 | 1 | 3 | 挂号, 日期, 门诊 | 进入预约挂号界面 |
| 协和医院 | 2 | 7 | 基本外科 | 进入预约挂号界面，打开"东单院区普通门诊"，查看基本外科医生列表 |
| 协和医院 | 3 | 10 | 医生主页, 咨询 | 进入预约挂号界面，打开"东单院区普通门诊"，查看基本外科医生列表，选择一个医生，查看医生介绍主页并点击常规咨询 |
| 协和医院 | 1 | 3 | 专科专病 | 在便民服务中点击专科专病 |
| 协和医院 | 2 | 6 | 胸部肿瘤组, 简介 | 在便民服务中点击专科专病，选择放射治疗科，点击胸部肿瘤组查看该病简介 |
| 协和医院 | 3 | 9 | 咨询 | 在便民服务中点击专科专病，选择放射治疗科，点击胸部肿瘤组查看该病简介后点击专家介绍，选择一个专家查看其主页并点击常规咨询选项 |
| 有道词典 | 1 | 4 | 高三, 图文 | 点击学习，搜索"高三" |
| 有道词典 | 2 | 7 | 评论, 发布 | 点击学习，搜索"高三"，进入图文分类点击一篇图文，查看评论 |
| 有道词典 | 3 | 10 | hello, world | 点击学习，搜索"高三"，进入图文分类点击一篇图文，查看评论后发表评论"hello world" |
| 知乎 | 1 | 3 | 编程猫, 实时 | 搜索"编程猫" |
| 知乎 | 2 | 7 | 编程猫, 关注 | 搜索"编程猫"，搜索用户，并进入结果中一个用户的主页，查看其最新发布的文章 |
| 知乎 | 3 | 13 | 编程猫, 评论 | 搜索"编程猫"，搜索用户，并进入结果中一个用户的主页，查看其最新发布的文章，点赞该文章，点击收藏，并在评论区发表评论"hello world" |
| 知乎 | 1 | 4 | 人工智能, 专栏 | 搜索"人工智能"专栏 |
| 知乎 | 2 | 10 | 评论 | 搜索"人工智能"专栏，查看一篇专栏中的文章并评论"hello world" |
| 知乎 | 3 | 13 | 更改 | 搜索"人工智能"专栏，查看一篇专栏中的文章并评论"hello world"，并将该文章收藏 |

### B.2.3   CROSS-APP ENGLISH TASKS

Table 6: Cross-app English tasks.

| Category | App | Diff Level | Golden Step | Task Description |
|---|---|---|---|---|
| General Tool | Google Play Store, Setting | 1 | 15 | Open Google Play Store, uninstall the Alibaba.com app, then go to Settings and verify if the app is still listed under app resources. |
| General Tool | Keep Notes, LinkedIn | 1 | 12 | Use the LinkedIn app to search for a customer service representative position. Select a job, open Keep Notes, create a new note, record the company's name, and set the note's title to 'customer service representative'. |
| General Tool | Clock, Setting | 1 | 12 | In the Settings app, enable 'Data Saver' mode. Open the Clock app and set an alarm for 6:00 AM. |
| Information Management | Facebook, Setting | 1 | 17 | Open Facebook, search for tropical pictures, save one picture to your phone, go to the Wallpaper section in the Settings app, and set the saved picture as your wallpaper. |

| Information Management | Calendar, Chrome | 1 | 16 | Using Chrome, search for the date of the next Winter Olympics opening ceremony and then set a reminder for that date in your Calendar. |
|---|---|---|---|---|
| Information Management | Spotify, Chrome | 1 | 13 | Open Chrome, search for the top Country songs of 2023, identify a song from the search results, then switch to Spotify and add that song to your playlist. |
| Media Entertainment | Google Play Store, Youtube | 1 | 12 | Watch a YouTube video about fitness tracking app recommendations, check the video's description for the suggested apps, then use Google Play Store to download one of the suggested apps. |
| Media Entertainment | Google Play Store, Chrome | 1 | 10 | Utilize Chrome to research different Recipe Organizer apps, and then proceed to Google Play Store, download one of your choice. |
| Media Entertainment | Clock, Youtube | 1 | 11 | Search for a relaxing soundscape video on YouTube, use the Clock app to set a timer for 3 hours, then go back to YouTube and play the video. |
| Multi Apps | Quora, eBay, Chrome | 2 | 20 | Utilize Chrome to search for a biography book, then use Quora to read reviews about the book, and finally add the book to watchlist on eBay. |
| Multi Apps | Clock, Chrome, Instagram | 2 | 20 | Organize a movie night by choosing a horror film using Chrome, sending an invitation to one of your friends via Instagram, and setting a reminder in the Clock app for 8:35 PM on Sunday. |
| Multi Apps | Triller, Setting, Google Play Store | 2 | 15 | First, install the Triller app from the Google Play Store. After the installation, open the Triller app, navigate to the Setting app to check current battery status, reopen the Triller app. |
| Multi Apps | Clock, WhatsApp, Zoom | 2 | 23 | Arrange a business meeting using Zoom, copy the sharing text, go to WhatsApp, send the copied text to a contact, set an alarm using the Clock app at the meeting time. |
| Multi Apps | AccuWeather, Evernote, Expedia | 2 | 25 | Utilize Expedia to search for Things to do in Beijing on 18-20th, choose one and record the sharing text using Evernote, open AccuWeather to check daily weather in Beijing. |
| Social Sharing | X, Facebook | 1 | 20 | Use the social media platform X to post a photo, copy the link to your post, then open Facebook and send the link to a friend |
| Social Sharing | BBC News, Gmail | 1 | 10 | Use the BBC News app to search for Artificial Intelligence news, read an article, share it via Gmail, send to agent.benchmark.2024@gmail.com. |
| Social Sharing | Spotify, Facebook | 1 | 19 | Listen to a Reggaeton album on Spotify, then share the album's name with a friend on Facebook. |
| Web Shopping | eBay, Facebook | 1 | 15 | Search for 'Circe by Madeline Miller' on Facebook, read one of the posts, head over to eBay, search the book, add it to watchlist. |
| Web Shopping | Amazon, Temu | 1 | 15 | Investigate the prices for Catan board game across Amazon and Temu, then proceed to add the cheaper option into your cart. |
| Web Shopping | Airbnb, Instagram | 1 | 19 | Use Instagram to search for an itinerary for Venice, Italy, and then proceed Airbnb, book accommodations at Venice, Italy. |

### B.2.4 CROSS-APP CHINESE TASKS

Table 7: Cross-app Chinese tasks.

| Category | App | Diff Level | Golden Step | Task Description |
|---|---|---|---|---|
| General Tool | 饿了么, 设置 | 1 | 10 | 打开饿了么，搜索"汉堡包"，然后进入设置APP，在应用中找到饿了么，关闭后台运行权限 |
| General Tool | 设置, 抖音 | 1 | 6 | 在设置APP中开启省流模式，然后打开抖音 |
| General Tool | 微信, 设置 | 1 | 12 | 进入设置，切换到深色模式，然后打开微信，将深色模式设置为"跟随系统" |
| Information Management | 华为浏览器, bilibili | 1 | 9 | 在华为浏览器中搜索"地球上最大的动物是"，然后在bilibili中搜索这种动物的视频并停留在搜索结果界面 |
| Information Management | 华为浏览器, QQ音乐 | 1 | 11 | 在华为浏览器中搜索 2024年的热门流行歌曲，选择一首歌曲后，切换到QQ音乐并将该歌曲添加到您的播放列表中 |
| Information Management | 小红书, 设置 | 1 | 18 | 打开小红书，搜索"冬日美景"，保存一张图片，然后在设置中将保存的图片更换为新的壁纸 |
| Media Entertainment | 华为浏览器, QQ音乐 | 1 | 12 | 在华为浏览器中搜索"歌曲七里香的作者是谁"，然后在QQ音乐中搜索这名歌手，进入歌手主页，播放任意一首歌并进入该歌曲的主页 |
| Media Entertainment | 抖音, 微博 | 1 | 16 | 利用抖音搜索"BLACKPINK"，观看任意一个视频，然后去微博搜索BLACKPINK账号并关注 |
| Media Entertainment | QQ音乐, bilibili | 1 | 10 | 打开QQ音乐，搜索周杰伦，查看他的主页，记录下一首歌曲，并在bilibili中搜索该歌曲相关的视频 |
| Multi Apps | 华为浏览器, bilibili, QQ | 2 | 14 | 在华为浏览器中搜索"贝塞斯达最成功的游戏是什么"，然后在bilibili搜索任意一个有关该游戏的视频，观看视频并分享到QQ空间 |
| Multi Apps | 淘宝, 京东, 腾讯文档 | 2 | 18 | 分别在淘宝和京东搜索"华为Mate60Pro"，然后在腾讯文档里新建一个"华为Mate60Pro"价格的文档，把淘宝和京东搜索到的价格记录下来 |
| Multi Apps | 高德地图, 美团, 微信 | 2 | 18 | 在美团搜索一家附近的餐厅，用高德地图查找驾车路线，把路线分享到微信朋友圈 |
| Multi Apps | 去哪儿, 航旅纵横, 微信 | 2 | 21 | 打开去哪儿APP搜索深圳酒店，切换到航旅纵横查看某天从北京飞往深圳的机票，并将其中一张机票分享给微信好友 |
| Multi Apps | 华为浏览器, 淘宝, 图库 | 2 | 16 | 在华为浏览器中搜索"英伟达最强专业计算卡"，在淘宝中搜索该计算卡并查看商品详情，保存预览图到图库，最后在图库查看这张图片 |
| Social Sharing | bilibili, QQ | 1 | 9 | 在bilibili中搜索"自制关卡 胆小菇之梦"，点击进入任意一个视频，分享该视频到qq空间 |
| Social Sharing | 小红书, QQ音乐 | 1 | 10 | 在QQ音乐上播放一首周杰伦的歌，然后将音乐分享到小红书，发布笔记 |

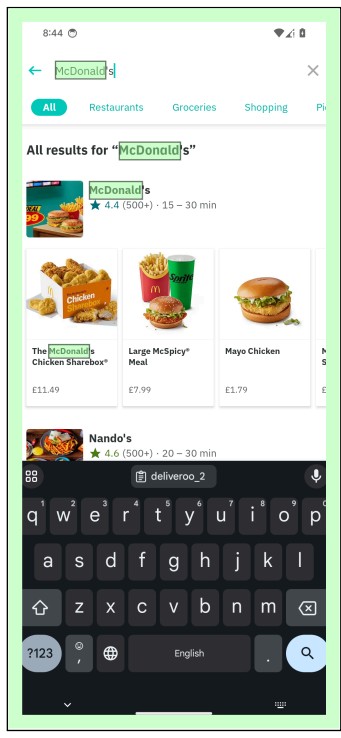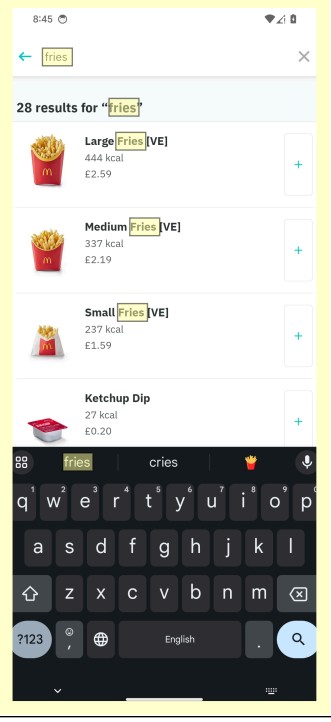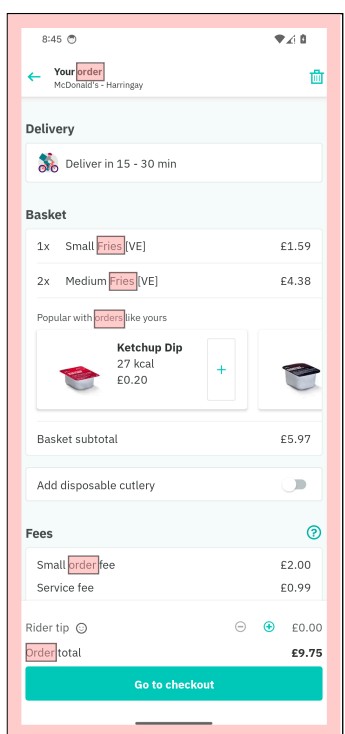

(a) Level 1: "mcdonald"        (b) Level 2: "fries"        (c) Level 3: "order" and "fries"

Figure 7: A visualised example of key components across three difficulty levels, with subcaptions indicating the key components for each level and highlighted key components in the corresponding screenshots.

| Social Sharing | 知乎, 微博 | 1 | 11 | 在知乎查看热榜，进入任意一个问题，然后将其转发到微博 |
| Web Shopping | 知乎, 京东 | 1 | 14 | 在知乎中搜索"1000元以下音箱推荐"，并在京东搜索其中提到的一款音箱，选择一个加入购物车 |
| Web Shopping | 小红书, 淘宝 | 1 | 14 | 在小红书上找到一款2024年推荐的运动相机，然后前往淘宝，将该商品加入购物车 |
| Web Shopping | 华为浏览器, 淘宝 | 1 | 14 | 在华为浏览器中搜索"最新款华为mate系列手机叫什么"，并在淘宝中搜索该型号的手机后将其加入购物车 |

### B.3 EXAMPLE OF KEY COMPONENTS

Figure 7 shows an example of key components.

### B.4 CROSS-APP EXAMPLE TASK DEMO

Figure 8 illustrates two examples of English cross-app tasks, each with a different difficulty level.

### B.5 STEPS OF TASKS

Refer to Figure 9 for a box plot illustrating the distribution of steps across tasks.

## C INTEGRATED AGENTS

The benchmark includes 11 state-of-the-art autonomous agents, shown in Table 8. These agents differ in core models, input modalities, action spaces, and additional training or prompting modules. They fall into two categories: those leveraging off-the-shelf MLLMs (e.g., GPT, Qwen), and those using

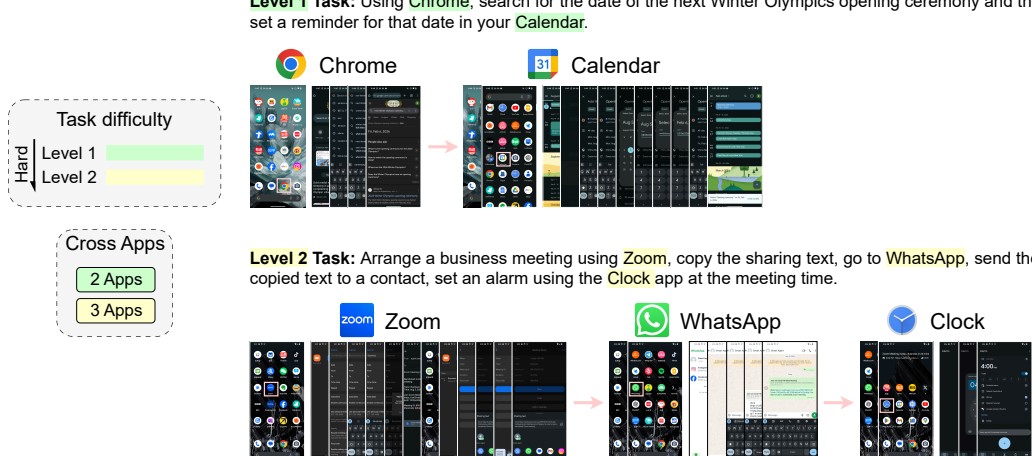

Figure 8: Example cross-app tasks with trajectories collected by human annotators.

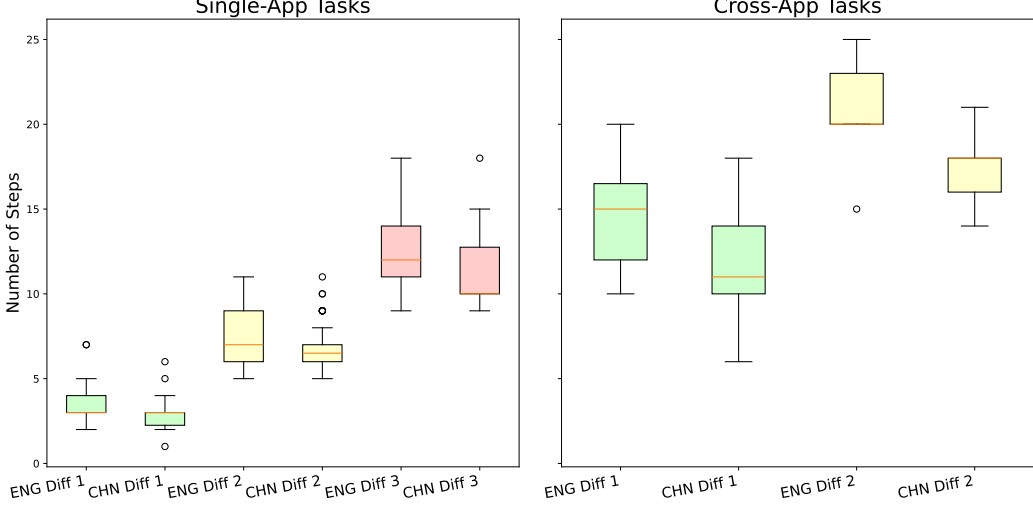

Figure 9: Distribution of steps taken by humans to execute tasks, categorised by difficulty level and task type.

fine-tuned models with parameter counts ranging from 1.3 billion to 18 billion. Fine-tuned models, trained primarily on the offline AITW (Rawles et al., 2024b) dataset, focus on action prediction, with DigiRL additionally employing online RL training. In our benchmarks, unlike their offline training settings, all agents are tested in real-world scenarios that require precise action grounding and long-sequence task execution.

## C.1 AGENT INPUT MODALITIES

Input modalities and action spaces define an agent's ability to interact with mobile user interfaces. Screenshot input is intuitive, capturing everything a human would see, but MLLMs often struggle to identify actionable UI elements and link them with screen coordinates (Zheng et al., 2024). To address this, some agents enhance input with XML files, accessibility trees, or information obtained through Optical Character Recognition (OCR). For instance, AppAgent (Yang et al., 2023b) and AutoDroid (Wen et al., 2024) use element IDs and coordinates, M3A (Rawles et al., 2024a) annotates screenshots with key UI elements, while MobileAgent (Wang et al., 2024b) first identifies interaction elements and then uses OCR or icon recognition to locate them.

Table 8: Comparison of agents integrated into SPA-BENCH framework across key dimensions.

| Agent | Core Model | UI Representation | Touch Point Localisation |
|---|---|---|---|
| AppAgent (Yang et al., 2023b) | GPT-4o | Screenshot + XML | Coordinates from XML |
| AutoDroid (Wen et al., 2024) | GPT-4o | HTML | Coordinates from HTML |
| MobileAgent (Wang et al., 2024b) | GPT-4o | Screenshot | OCR + Icon Recognition |
| MobileAgentV2 (Wang et al., 2024a) | GPT-4o | Screenshot | OCR + Icon Recognition |
| M3A (Rawles et al., 2024a) | GPT-4o | Screenshot + Accessibility Tree | Coordinates from Accessibility Tree |
| T3A (Rawles et al., 2024a) | GPT-4o | Accessibility Tree | Coordinates from Accessibility Tree |
| SeeAct (Rawles et al., 2024a; Zheng et al., 2024) | GPT-4o | Screenshot + Accessibility Tree | Coordinates from Accessibility Tree |
| Auto-UI (Zhan & Zhang, 2023) | Fine-tuned FLAN-Alpaca-Base (200M) + BLIP-2-T5-Instruct (1.1B) | Screenshot | Normalized coordinates from Model |
| CogAgent (Hong et al., 2024) | CogAgent-18B | Screenshot | Normalized coordinates from Model |
| DigiRL (Bai et al., 2024) | Fine-tuned FLAN-Alpaca-Base (200M) + BLIP-2-T5-Instruct (1.1B) | Screenshot | Normalized coordinates from Model |
| OdysseyAgent (Lu et al., 2024) | Fine-tuned Qwen-VL (9.6B) | Screenshot | Normalized coordinates from Model |

## C.2 ADOPTION OF AGENTS INTO FRAMEWORK

Integrating agents into the framework required several adaptations. We used their original open-source implementations, with the exception of SeeAct (Zheng et al., 2024), for which we adopted AndroidWorld's action grounding module. For agents using fine-tuned models (i.e., Auto-UI, DigiRL, OdysseyAgent, CogAgent), which lacked direct Android interaction capabilities, we used UIAutomator2 for end-to-end task execution.

## C.3 LOGS AND ERRORS

While task descriptions and screenshot trajectories remain the primary inputs/outputs, we also logged executed actions, performance metrics (steps, time, API costs), and errors. Errors were categorised as expected (e.g., invalid responses) or unexpected (e.g., network failures). Expected errors arise from the agent's limitations, such as failing to generate valid actions or when certain functionalities are restricted. Unexpected errors refer to unforeseeable issues like network failures, Android malfunctions, or CAPTCHA challenges. The framework automatically re-runs such tasks to avoid penalising agents for unexpected errors, ensuring a fair and accurate assessment of their capabilities and limitations.

## C.4 SCOPE OF USING ANDROID EMULATOR

Certain English tasks involving WhatsApp and OneNote, as well as most Chinese tasks, were executed exclusively on physical Android devices rather than emulators . This decision was due to strict app control measures, such as restrictions on logging in across multiple devices and compatibility issues with emulator system images. While physical Android devices can replace the emulator, doing so would eliminate the snapshot functionality described in Section 4.2.

## D SINGLE-APP SUCCESS DETECTION

### D.1 COARSE DETECTION: KEY COMPONENT MATCHING

Given a single screenshot, PaddleOCR is used to extract text, which is then lowercased and concatenated to minimise inaccuracies. This text is matched against key components of the final state (defined by human annotators in Section 3.2). Matching starts from the last screenshot and moves

---

Auto-UI has been renamed to Auto-GUI, but in this paper, we use Auto-UI as it is more commonly referenced in previous works.

```
https://github.com/openatx/uiautomator2
https://developer.android.com/studio/run/emulator
https://github.com/PaddlePaddle/PaddleOCR
```

Table 9: The proportion of reduction in MLLM evaluation times through key component matching, and the F1 score performance of our MLLM evaluator (without key component matching) across reasoning and action modes. Bold values indicate the best performance for each task and language pair.

| Task | Language | Reduction Rate | No Action | | Text Action | | Image Action | |
|---|---|---|---|---|---|---|---|---|
| | | | Result-only | Reason-and-Result | Result-only | Reason-and-Result | Result-only | Reason-and-Result |
| Single-app | English | 0.313 | 0.911 (-0.003) | 0.922 (-0.033) | 0.919 (-0.016) | 0.903 (-0.040) | **0.926 (-0.006)** | 0.915 (-0.050) |
| | Chinese | 0.670 | 0.879 (-0.076) | 0.857 (-0.102) | 0.883 (-0.092) | **0.884 (-0.113)** | 0.872 (-0.093) | 0.864 (-0.129) |

backward until a match is found or the first screenshot is reached. If no match is found, the task is marked as failed, skipping fine detection.

## D.2 FINE DETECTION: MLLM EVALUATION

If coarse detection is successful, fine detection is performed using a MLLM evaluator (based on GPT-4o). The evaluator receives task descriptions, screenshots, and executed actions to assess task success. Action information can be presented as either text or concatenated screenshots. Prompts used for the MLLM evaluator are detailed in Appendix D.4.

## D.3 APPROACH EVALUATION AND RESULTS

To validate the single-app success detection pipeline, we compared its detection against human evaluations for AppAgent and M3A (English tasks), and CogAgent and MobileAgentV2 (Chinese tasks). Two reasoning and three action modes were tested to prompt the MLLM, and an ablation study was conducted to assess the impact of coarse detection.

Table 9 presents the proportion of fine detection time reduced before and after applying coarse detection, along with the F1 scores for each reasoning and action mode across English and Chinese tasks, both with and without coarse detection. The results demonstrate that coarse detection effectively enhances performance by reducing the frequency of fine detection calls and improving the success detection F1 score, particularly in Chinese tasks where MLLM evaluation struggles. While no significant differences were found between reasoning modes, incorporating action data improved decision-making but also increased token length, which sometimes led to hallucinations.

Overall, in the best-performing evaluation modes, our pipeline achieved F1 scores of 0.926 for English tasks and 0.884 for Chinese tasks, demonstrating its effectiveness in aligning with human evaluations. For further task evaluations, we use these modes to detect success: result-only reasoning with image action for English tasks, and reason-and-result with text action for Chinese tasks.

## D.4 PROMPTING TEMPLATES

### D.4.1 SYSTEM PROMPT

```
You are an expert in evaluating smartphone operation tasks. Your primary role is
to determine whether a task has been successfully completed based on a series of
screenshots (provided in order of execution) and the corresponding task description.
### Guidelines:
1. **No Assumptions**: Evaluate solely based on the provided screenshots. Do not
infer or assume details that aren't explicitly shown.
2. **Subtask Completion**: A task is successful only when all its subtasks are
successfully completed. For example, for the task "Go to the website github.com. Add
this website to the reading list,", it is successful only if the screenshots show
github.com has been navigated to and then added to the reading list.
3. **Common Reasons for Subtask Failure**:
- **Incomplete**: A subtask is not successful if it is not performed or achieved. Same
task example above, visiting the website but not adding it to the reading list results
in task failure.
- **Incorrect Execution**: A subtask fails if the screenshots do not align with any
part of the instruction.
```

```
– **Wrong Noun/Entity**:  If the subtask is "Go to the website github.com." but the
screenshots show google.com, the subtask fails.  Similar entities (e.g., 'iPhone 11' vs.
'iPhone 12' or 'driving directions' vs. 'walking directions') are considered different,
leading to task failure if not correctly executed.
– **Wrong Verb/Action**:  If the subtask is "Like a post," but the screenshots show the
post was reposted instead, the subtask fails due to incorrect action.
4.  **Additional Actions**:  If intermediate screenshots show all subtasks are
successful, consider the task a success, even if additional actions are shown afterward.
This applies as long as these actions do not impact task completion or cause the
original task to fail.
5.  **Filtering Subtask**:  If a subtask involves filtering based on specific criteria,
ensure the filter has been applied (i.e., a specific app feature).  If the filter is
treated as an additional search condition, the subtask fails.
6.  **Order of Subtasks**:  Subtasks can be completed in any order unless they are
explicitly dependent on each other.
7.  **Subtasks Completed Midway**:  Subtasks completed in the middle of the process may
not be reflected in the final screenshot; these should still be considered successful
if they align with the task requirements.
8.  **Corrective Actions**:  Subtasks that initially appear to fail but are corrected
by subsequent actions should be considered successful only when the correction fully
aligns with the original task.
9.  **Intermediate Steps**:  It's acceptable if a subtask isn't completed in one go, as
long as the final result meets the task requirements; consider this a success.
10.  **Focus on Overview**:  Pay attention to the overall objective and avoid letting
minor, irrelevant details distract from the main evaluation.
11.  **UI Differences**:  Be mindful of subtle UI differences (e.g., different font
styles or colors indicating selected tabs).
action_sys_prompt_template(action_mode)
**These guidelines serve as a general framework.  Apply them thoughtfully and avoid
overfitting to edge cases not covered.  Be strict and cautious when determining whether
a task has been successfully completed or not.  Use 1 to indicate success and 0 to
indicate failure.**
```

### D.4.2  SYSTEM PROMPT WITH ACTION

```
12.  **Use of Action Information**:  Some quick pop-ups may not be captured by
screenshots provided.  If needed, consider the action information when evaluating
the task.
13.  **Single Action for Multiple Subtasks**:  Some subtasks can be completed with a
single action, such as clicking an icon that shuffles a playlist.
### Common Actions:  – Click/Tap:  The user selects or activates a specific point on
the screen, triggering an event or interaction.
– Long Press:  The user presses and holds a point to trigger a secondary action or menu.
– Swipe/Scroll:  The user drags their finger across the screen to scroll or navigate;
the content or screen position changes according to the direction.
– Type/Input Text:  The user types or inputs text into a field.
– Back:  The user presses the back button to return to the previous screen.
```

### D.4.3  BASE PROMPT

```
Now, here is a smartphone operation task description:
**task_description** history_info
Please carefully determine whether the task has been correctly and completely executed
according to the provided screenshots.  Use 1 to indicate success and 0 to indicate
failure.
action_prompt[0]
reasoning_prompt
Remember:
– Do not make assumptions based on information not presented in the screenshots.  Only
evaluate what is explicitly shown.
– Ensure that every entity and action in the task description is precisely matched and
fulfilled.
– Consider additional actions taken after a task is successfully completed as part
of the success, as long as those actions don't impact the task's completion or cause
failure.
– A filtering subtask is only correct when a specific filter is applied as a feature of
the app.  Using the criteria as a keyword search will cause the subtask to fail.
– Subtasks can be completed in any order unless they are explicitly dependent on each
other.
– Subtasks completed correctly mid-process, even if not reflected in the final
screenshot, should be considered successful.
```

```
- Subtasks that initially appear to fail but are corrected by subsequent actions should
be considered successful.
- A task can be considered successful even if some subtasks are not completed in one go,
as long as the final result meets the task requirements.
- Focus on the overall objective of the task without being distracted by minor,
irrelevant details.
- Pay attention to subtle UI differences that might indicate task completion or failure,
such as highlighted tabs or changes in font.
action_prompt[1]
```

### D.4.4 BASE PROMPT WITH TEXT ACTION

```
To assist you in determining whether the task was successful, action information
is provided.  Use this information only when you cannot determine success purely
based on the screenshots.  The i-th screenshot may contain details that change the
screenshot from the i-th to the i+1-th, while the last screenshot contains no action
information as the task ends afterward.  In some screenshots, a red dot may indicate
where a specific action occurred (e.g., clicked or long-pressed), triggering an event
or interaction.  If there isn't a red dot, the action is more complex than a single
position operation (e.g., a swipe or text input).  You can find the details of these
actions below, if applicable.
extra_action
```

```
- Consider the action information only when necessary.
- Pop-ups that appear immediately after an action may not be captured in the
screenshots; do not consider this a failure.
- Some subtasks can be completed with a single action, such as clicking an icon that
shuffles a playlist.
```

### D.4.5 BASE PROMPT WITH IMAGE ACTION

```
To assist you in determining whether the task was successful, action information is
provided.  Use this information only when you cannot determine success purely based on
the screenshots.  The action information on the i-th screenshot describes the changes
from the i-th screenshot to the i+1-th screenshot, while the last screenshot contains
no action information as the task ends afterward.  This information is presented as
a white strip attached to the original screenshot, separated by a blue line.  In some
screenshots, a red dot may indicate where a specific action occurred (e.g., clicked or
long-pressed), triggering an event or interaction.
```

```
- Consider the action information only when necessary.
- Pop-ups that appear immediately after an action may not be captured in the
screenshots; do not consider this a failure.
- Some subtasks can be completed with a single action, such as clicking an icon that
shuffles a playlist.
```

### D.4.6 RESULT-ONLY PROMPT

```
Please provide your decision using the following template without any reasoning:
Result:  <1 OR 0>
```

### D.4.7 REASON-AND-RESULT PROMPT

```
Use the following format for your response:
Reason:  <Brief description of why you believe the task was successful or failed,
including the alignment or misalignment between the task description and screenshots,
starting with "I believe this task is successful/failed">
Result:  <1 OR 0>
```

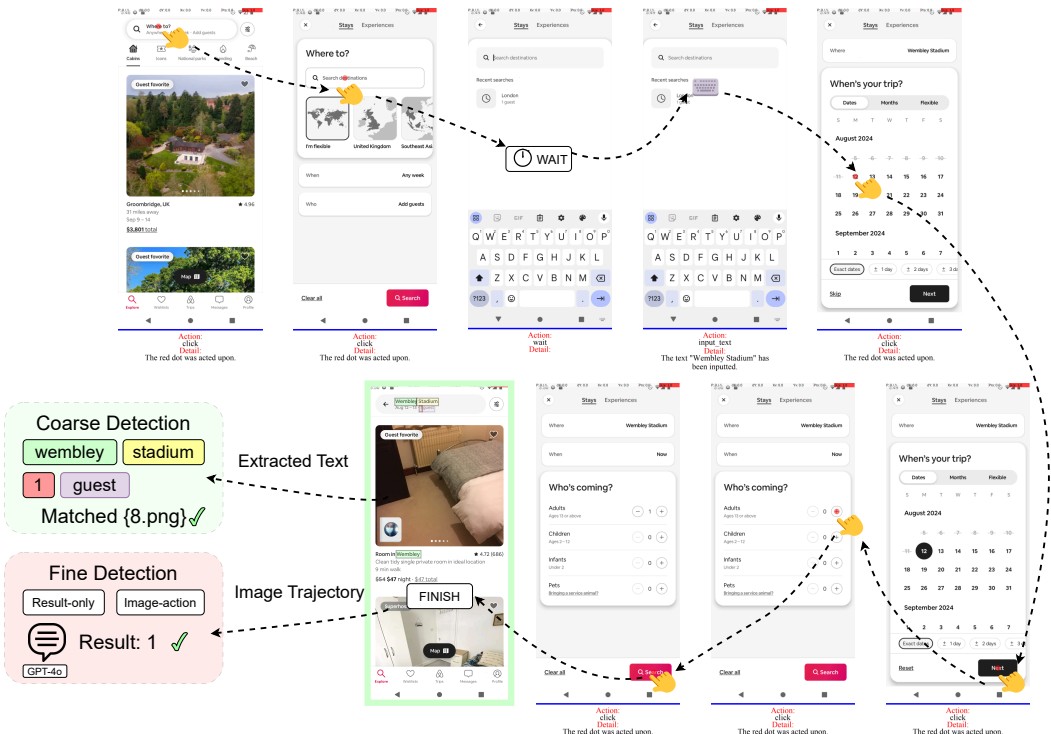

Figure 10: Evaluation of the "airbnb_1" task executed by M3A. All four annotated key components were successfully matched in the OCR-extracted text from the final screenshot, allowing the task to pass both coarse and fine detection.

## D.5 EXAMPLE OF SUCCESS DETECTION

Figure 10 illustrates a coarse-to-fine evaluation of the "airbnb_1" task executed by M3A, which corresponds to the Airbnb Level 2 task listed in Table 4).

## E CROSS-APP SUCCESS DETECTION

### E.1 SUBTASK GENERATION

For a cross-app task, each subtask is tied to a single app, and any adjacent subtasks must use different apps. However, the same app can appear multiple times as long as there is at least one different app between occurrences. Beyond "app" and "task description", each subtask also includes the fields "history" and "memory". The "history" field is a boolean value indicating whether the subtask requires information from previous tasks, highlighted as phrases in the task description. This information, referred to as "memory", consists of phrases that will be matched with the highlighted "history" phrases. Such subtasks are generated by a MLLM and then reviewed by humans to ensure quality. Examples of subtasks are provided below, and detailed prompts can be found in the Appendix E.5.

### E.2 STAGE 1: TRAJECTORY SPLIT

Stage 1 splits the entire trajectory into segments based solely on app transitions as preparation for detecting subtask success. The previous subtask generation step provides an ordered list of apps for each task, indicating the sequence in which they should be operated for successful completion. A MLLM processes this app list along with the complete series of execution screenshots, segmenting the trajectory so that each part includes only screenshots related to the corresponding app's operations. If the segmentation is invalid, such as when an app is missing or the sequence is incorrect, the task is marked as unsuccessful due to errors in one or more apps.

### E.3 STAGE 2: SEQUENTIAL SUBTASK SUCCESS DETECTION

Stage 2 is activated when the segmentation is valid, meaning each app in the ordered list has a unique series of screenshots. Subtasks are checked sequentially, with each subtask evaluated only if its predecessor is marked as successful. If a subtask is marked as successful, the phrases in its "memory" field (unless the field is empty), will be required as historical references for subsequent subtasks. This memory is generated by another MLLM, which summarises the current screenshots based on the required phrases and appends the relevant information to the memory set for future use. If a subsequent subtask's "history" field is marked as true, the necessary phrases are then extracted and matched with the stored information to assist in evaluating success. Such historical data, combined with partial task screenshots and action details, is used to determine the subtask's success. Since each subtask involves only a single app, it uses the same MLLM evaluation method applied in single-app success detection. The entire task is considered successful only if all subtasks pass. Otherwise, it fails as soon as any subtask is marked unsuccessful.

### E.4 APPROACH EVALUATION AND RESULTS

To validate the cross-app success detection pipeline, we compared its results against human evaluations using four different agents per language. For English tasks, the agents were M3A, T3A, Auto-UI, and OdysseyAgent, while for Chinese tasks, we used AppAgent, MobileAgent, MobileAgentV2, and CogAgent. Table 10 presents the F1 scores of our cross-app success detection pipeline for both English and Chinese tasks. The performance is lower compared to single-app success detection due to the increased complexity of cross-app tasks. With over 90% of tasks being true negatives, even a small number of errors significantly impacts the overall performance. Additionally, we observed that for each agent, false positives and false negatives occurred at a similar rate. Thus, despite a relatively modest F1 score, the pipeline's success detection still reflects each agent's performance.

Table 10: The F1 score performance of our cross-app success detection pipeline.

|  | Cross-app | |
|---|---|---|
|  | English | Chinese |
| F1 Score | 0.833 | 0.857 |

### E.5 PROMPTING TEMPLATES

#### E.5.1 SYSTEM PROMPT OF STAGE 1

```
You are provided with a sequence of screenshots representing an agent performing tasks
across multiple apps on a smartphone.  Each screenshot corresponds to a specific action.
You are also given a list of apps that should be used in the task.
**Your task is to:**
1.  Split the screenshots into segments based on transitions between apps in the given
list.  Do not change the order of apps, even if they do not match the screenshot order.
Output the results based on the provided app list order.
2.  For each app, identify where the agent opens and operates within the app.  Each
app interaction requires at least two screenshots:  one for opening the app and one
for quitting or switching to another, except for the final app, which may not require a
quit action.
3.  **Ensure that the start and end indices you provide are within the range of
screenshots sent to you.** You will receive a certain number of screenshots, and you
must repeat how many screenshots you received before processing.  Any indices provided
should not exceed the total number of screenshots.
4.  If an app from the list is missing in the screenshots, return `-1` for both the
start and end screenshot indices for that app.
5.  Ignore screenshots that show irrelevant actions (e.g., the home screen or unrelated
apps).  You may mention them in the analysis but do not include them in the final
result.
6.  An app may appear more than once in the list (e.g., `["AppA", "AppB", "AppA"]`),
but there must be another app between repeated instances of the same app.
7.  There might be distractors (e.g., advertisements and popups) in the screenshots;
you should not interpret them as transitions between apps.
### Example Input:
**App list:** `["AppA", "AppB", "AppA"]`
**Screenshots:** A sequence of numbered screenshots.
### Example Reasoning:  1.  **Screenshots 1-3:** The agent opens AppA, and operates
within it.  2.  **Screenshots 4-5:** The agent opens AppB and operates within it.  3.
**Screenshot 6:** The agent interacts with the home screen, which is irrelevant.  4.
**Screenshots 7-9:** The agent opens AppA again and operates within it.
```

```
### Final Output:  { "AppA_1": { "start screen": 1, "end screen": 3 }, "AppB": {
"start screen": 4, "end screen": 5 }, "AppA_2": { "start screen": 7, "end screen":
9 } }
**task_description**
```

## E.5.2   USER PROMPT OF STAGE 1

```
Here is the app list:  task_app Ensure the order of apps in your final output is
exactly the same as the order provided in my app list.
```

## E.5.3   SYSTEM PROMPT OF STAGE 2 MEMORY

```
You are an MLLM tasked with analyzing screenshots and summarizing the relevant
information based on a description provided by the user.  Only summarize information
from screenshots that relate to the description, ignoring any that are unrelated.  If
the screenshots show a list of results (e.g., a search page), summarize or list all
the relevant results.  The summary should be clear and concise, without bullet points,
step-by-step details, or line breaks.
```

## E.5.4   USER PROMPT OF STAGE 2 MEMORY

```
Here is the description:  memory_text
```

## E.5.5   SUBTASK GENERATION

```
You are tasked with splitting a smartphone control instruction into a series of
subtasks, each corresponding to specific app interactions.  For each subtask, you
should define:
1.  **app**:  The name of the app being used in the subtask.
2.  **task**:  A string describing the action to be performed.  Do not include the
app name in the task description unless necessary (e.g., if the task is to only open
the app).  Use '{PREVIOUS MEMORY}' if the task depends on information from a previous
subtask.  This should be exactly the same phrase as the previous subtask's memory (i.e.,
if history is True).
3.  **history**:  A boolean value ('True' or 'False') indicating whether this subtask
relies on data from a previous subtask.
4.  **memory**:  If applicable, specify a piece of information that the current subtask
generates or retrieves, which will be passed to the next subtask.  If no memory is
needed, set this to 'None'.
**Guidelines**:
- Use the same language for the split task as the task description.
- If there are several consecutive subtasks for the same app, combine them into a single
 subtask (i.e., adjacent subtasks should not have the same app).  Subtasks for the same
 app are acceptable if there is at least one subtask for a different app in between.
- By default, each subtask should be independent unless explicitly needing data from a
 prior subtask (in which case, set '"history": True').
- Flexibly determine whether any information should be stored as **memory** and passed
 to subsequent tasks, based on the task's natural requirements.
- Output the subtasks in a structured format like the following:
{ "subtask_1":{ "app":"APP", "task":"TASK", "history":"BOOL", "memory":"MEMORY" },
"subtask_2":{ "app":"APP", "task":"TASK", "history":"BOOL", "memory":"MEMORY" }, ...  }
###Example 1
**Task**:  Adjust the notification settings for the YouTube app on your phone using
Settings, then proceed to open YouTube.
**Result**:
{ "subtask_1":{ "app":"Settings", "task":"Adjust the notification settings for
the YouTube app on your phone", "history":false, "memory":"None" }, "subtask_2":{
"app":"YouTube", "task":"Open YouTube", "history":false, "memory":"None" } }
### Example 2
**Task**:  Utilize the X app to research and identify a highly recommended robotic
vacuum cleaner, and then go to Amazon to purchase one.
```

Table 11: Task performance on single-app Chinese tasks. SRC and MSR refer to Self-Reported Completion and Maximum Steps Reached, respectively. The token costs of four agents are omitted because they use locally hosted open-source models.

| Agent | Success Rate | Mean Step Ratio on Success | Termination Reason | | | Termination Inaccuracy | | Mean Exec Time per Step (sec) | Mean Token Cost per Step (USD) |
|---|---|---|---|---|---|---|---|---|---|
| | | | SRC Rate | MSR Rate | Error Rate | Premature Rate | Overdue Rate | | |
| *Agentic Workflow (GPT-4o)* | | | | | | | | | |
| AppAgent | 0.247 | 1.66 | 0.100 | 0.393 | 0.507 | 0.600 | 0.407 | 25.6 | 0.013 |
| AutoDroid | 0.187 | 1.25 | 0.567 | 0.360 | 0.073 | 0.729 | 0.111 | 48.8 | **0.011** |
| MobileAgent | 0.240 | 1.39 | 0.273 | 0.653 | 0.074 | 0.439 | 0.133 | 35.6 | 0.037 |
| MobileAgentV2 | 0.440 | 1.28 | 0.460 | 0.487 | 0.053 | 0.333 | 0.274 | 104.5 | 0.075 |
| M3A | **0.447** | 1.08 | 0.640 | 0.360 | **0** | 0.323 | 0.037 | 20.8 | 0.097 |
| T3A | 0.380 | 1.31 | 0.507 | 0.493 | **0** | 0.408 | 0.162 | **12.6** | 0.128 |
| SeeAct | 0.327 | 1.91 | 0.067 | 0.927 | 0.006 | **0.300** | 0.302 | 23.0 | 0.050 |
| *Agent-as-a-Model* | | | | | | | | | |
| Auto-UI | 0.007 | **0.50** | **0.893** | **0.107** | **0** | 0.993 | **0** | - | - |
| CogAgent | 0.027 | 1.79 | 0.060 | 0.893 | 0.047 | 1.000 | 0.030 | - | - |
| DigiRL | 0 | - | 0.387 | 0.520 | 0.093 | 1.000 | **0** | - | - |
| OdysseyAgent | 0.007 | 2.00 | 0 | 1.000 | **0** | - | 0.007 | - | - |

```
**Result**:
{ "subtask_1":{ "app":"X", "task":"Research and identify a highly recommended
robotic vacuum cleaner", "history":false, "memory":"robotic vacuum cleaner" },
"subtask_2":{ "app":"Amazon", "task":"Go to Amazon to purchase {robotic vacuum
cleaner}", "history":true, "memory":"None" } }
Now, for any smartphone control instruction, decompose the task into subtasks using the
format above.
```

# F EXPERIMENT DETAILS

## F.1 AGENT CONFIGURATION

The agents in this benchmark include variations in core models and optional modules. Of the 11 agents, 7 originally used off-the-shelf (M)LLMs such as GPT-4V and Qwen-VL-Max. For consistency, these agents were upgraded to GPT-4o, including replacing MobileAgentV2's Qwen-VL-Chat with GPT-4o-mini for icon recognition. For Auto-UI and DigiRL (fine-tuned), the Auto-UI-Base core model was selected.

Agent-specific configurations include:

- **AppAgent**, **SeeAct**, **M3A**, and **T3A**: Added AdbKeyboard for Chinese character input, following the MobileAgent setup.

- **Auto-UI**: Enabled "action history" and "chain of actions" features.

- **OdysseyAgent**: Enabled action and screenshot history.

- **AppAgent** and **AutoDroid**: No additional knowledge or exploration was allowed before experiments.

For all other settings, the default configurations provided by the developers were used. Agents were allowed to execute up to twice the number of "golden steps" for a task, after which execution was halted.

## F.2 EXPERIMENTAL RESULTS

See Tables 11, 12, 13 for the detailed experiment results of single-app Chinese, cross-app English, and cross-app Chinese tasks respectively.

---

https://github.com/senzhk/ADBKeyBoard

Table 12: Task performance on cross-app English tasks. SRC and MSR refer to Self-Reported Completion and Maximum Steps Reached, respectively. The token costs of four agents are omitted because they use locally hosted open-source models.

| Agent | Success Rate | Mean Step Ratio on Success | Termination Reason | | | Termination Inaccuracy | | Mean Exec Time per Step (sec) | Mean Token Cost per Step (USD) |
|---|---|---|---|---|---|---|---|---|---|
| | | | SRC Rate | MSR Rate | Error Rate | Premature Rate | Overdue Rate | | |
| *Agentic Workflow (GPT-4o)* | | | | | | | | | |
| AppAgent | 0 | - | 0.200 | 0.550 | 0.250 | 1.000 | **0** | 22.9 | **0.014** |
| MobileAgent | 0.050 | 2.00 | 0.100 | 0.900 | **0** | 1.000 | 0.056 | 25.3 | 0.089 |
| MobileAgentV2 | 0.100 | 2.00 | 0.250 | 0.750 | **0** | 1.000 | 0.133 | 58.8 | 0.071 |
| M3A | **0.200** | **1.16** | **0.700** | **0.300** | **0** | 0.714 | **0** | 17.3 | 0.082 |
| T3A | 0.100 | 1.43 | 0.600 | 0.400 | **0** | 0.833 | **0** | **12.1** | 0.091 |
| SeeAct | 0.100 | 1.52 | 0.150 | 0.850 | **0** | **0.333** | **0** | 19.9 | 0.043 |
| *Agent-as-a-Model* | | | | | | | | | |
| Auto-UI | 0 | - | 0.100 | 0.800 | 0.100 | 1.000 | **0** | - | - |
| CogAgent | 0 | - | 0.050 | 0.950 | **0** | 1.000 | **0** | - | - |
| DigiRL | 0 | - | 0.050 | 0.550 | 0.400 | 1.000 | **0** | - | - |
| OdysseyAgent | 0 | - | 0 | 0.650 | 0.350 | - | 0.007 | - | - |

Table 13: Task performance on cross-app Chinese tasks. SRC and MSR refer to Self-Reported Completion and Maximum Steps Reached, respectively. The token costs of four agents are omitted because they use locally hosted open-source models.

| Agent | Success Rate | Mean Step Ratio on Success | Termination Reason | | | Termination Inaccuracy | | Mean Exec Time per Step (sec) | Mean Token Cost per Step (USD) |
|---|---|---|---|---|---|---|---|---|---|
| | | | SRC Rate | MSR Rate | Error Rate | Premature Rate | Overdue Rate | | |
| *Agentic Workflow (GPT-4o)* | | | | | | | | | |
| AppAgent | 0 | - | 0 | 0.550 | 0.450 | - | **0** | 23.5 | **0.014** |
| MobileAgent | **0.100** | 1.62 | 0.150 | 0.750 | 0.100 | **0.667** | 0.067 | 53.4 | 0.064 |
| MobileAgentV2 | **0.100** | 1.89 | 0.200 | 0.750 | 0.050 | 1.000 | 0.133 | 104.1 | 0.075 |
| M3A | **0.100** | 1.32 | 0.500 | 0.500 | **0** | 0.800 | **0** | 17.8 | 0.091 |
| T3A | **0.100** | **1.08** | 0.750 | 0.250 | **0** | 0.867 | **0** | **13.4** | 0.110 |
| SeeAct | 0.050 | 2.00 | 0.100 | 0.900 | **0** | 1.000 | 0.056 | 17.3 | 0.045 |
| *Agent-as-a-Model* | | | | | | | | | |
| AutoUI | 0 | - | **1.00** | 0 | **0** | 1.000 | **0** | - | - |
| CogAgent | 0 | - | 0.050 | 0.850 | 0.100 | 1.000 | **0** | - | - |
| DigirlAgent | 0 | - | 0.800 | 0.050 | 0.150 | 1.000 | **0** | - | - |
| GUI_Odyssey | 0 | - | 0 | 0.500 | 0.500 | - | **0** | - | - |

## F.3 PERFORMANCE ACROSS TASK DIFFICULTY LEVELS

Table 14 shows agent performance across different difficulty levels. As expected, agents perform better on easier tasks, confirming that our tasks are designed with increasing difficulty, where lower-level tasks serve as subtasks for higher-level ones. The overall trend in performance across difficulty levels aligns with each agent's general success rate discussed in Section 6.1.

## G EXPERIMENTS ON OPEN-ENDED SINGLE-APP ENGLISH TASKS

To further explore the scalability of our success detection approaches, we designed an initial set of ten "open-ended" single-app English tasks across distinct apps, as detailed in Table 15.

Table 15: Open-ended single-app English tasks.

| App | Task Description |
|---|---|
| Airbnb | I'm traveling to London with three friends and need accommodation. I'll manage the checkout process myself. |
| Amazon | I'd like to buy wedding gifts for my friend and their partner. I'll take care of the checkout myself. |
| Calculator | I want to show my friend the multiplication of two negative numbers is indeed a positive number. |
| Chrome | I'm planning a trip to ski and would like to save a blog to read later. |
| Clock | Please set two alarms, one for weekdays and another for weekends. I prefer waking up later on weekends. |
| Merriam-Webster | I'd like to expand my vocabulary in political ideologies. I aim to learn two new terms today. |
| Google Maps | My car is low on gas, and I'm also feeling hungry. |
| Settings | Sometimes I have trouble reading the screen clearly. |
| Spotify | Create a music playlist for me in a recommended genre. Just two songs will do. |
| YouTube | I'm interested in watching tech tutorial videos recently. |

Table 14: Success rates on single-app English, single-app Chinese, cross-app English and cross-app Chinese tasks, categorised by difficulty level. AutoDroid was tested only on single-app tasks as its agent framework, Droidbot (Li et al., 2017), supports only these tasks.

| Agent | Single-app English Tasks | | | Single-app Chinese Tasks | | | Cross-app English Tasks | | Cross-app Chinese Tasks | |
|---|---|---|---|---|---|---|---|---|---|---|
| | Level 1 | Level 2 | Level 3 | Level 1 | Level 2 | Level 3 | Level 1 | Level 2 | Level 1 | Level 2 |
| *Agentic Workflow (GPT-4o)* | | | | | | | | | | |
| AppAgent | 0.540 | 0.340 | 0.140 | 0.400 | 0.180 | 0.160 | 0 | 0 | 0 | 0 |
| AutoDroid | 0.560 | 0.300 | 0.120 | 0.360 | 0.120 | 0.080 | - | - | - | - |
| MobileAgent | 0.620 | 0.380 | 0.160 | 0.300 | 0.240 | 0.180 | 0.067 | 0 | 0.067 | 0.200 |
| MobileAgentV2 | 0.700 | 0.400 | 0.200 | **0.580** | 0.420 | **0.320** | 0.133 | 0 | **0.133** | 0 |
| M3A | **0.800** | **0.700** | **0.420** | 0.500 | **0.520** | 0.320 | **0.267** | 0 | **0.133** | 0 |
| T3A | 0.720 | 0.480 | 0.260 | 0.480 | 0.460 | 0.200 | 0.133 | 0 | **0.133** | 0 |
| SeeAct | 0.600 | 0.460 | 0.120 | 0.500 | 0.340 | 0.140 | 0.133 | 0 | 0.067 | 0 |
| *Agent-as-a-Model* | | | | | | | | | | |
| Auto-UI | 0.040 | 0 | 0 | 0.020 | 0 | 0 | 0 | 0 | 0 | 0 |
| CogAgent | 0.060 | 0 | 0 | 0.040 | 0.040 | 0 | 0 | 0 | 0 | 0 |
| DigiRL | 0.020 | 0.040 | 0 | 0 | 0 | 0 | 0 | 0 | 0 | 0 |
| OdysseyAgent | 0.140 | 0.020 | 0 | 0.004 | 0.020 | 0 | 0 | 0 | 0 | 0 |

As discussed in Section 3.2, when a task description is clearly defined with a specific goal, its executions typically converge to the same final state. Such tasks can be treated as "closed-ended" tasks, which form the basis for human-annotated key components. In contrast, for a more vague task description, the task is considered "open-ended". The final state may result in multiple possible outcomes, making it challenging to define key components explicitly. While the coarse detection phase may be limited in such cases, we hypothesised that our fine detection approach, relying on the MLLM evaluator, remains effective and can still be applied to "open-ended" tasks.

In this initial experiment, we tested the seven agents that follow the agentic workflow on the ten "open-ended" tasks. Given the open-ended nature of these tasks and the absence of predefined golden steps, agents were allowed a maximum of 20 steps to complete each task. We compared the alignment of success detection results between human evaluations and our MLLM evaluator. Using the same MLLM evaluator introduced in Section 5.2, we identified 22 true positives, 2 false positives, 2 false negatives, and 44 true negatives. This resulted in an F1 score of 0.917, consistent with the corresponding results for "closed-ended" tasks reported in Table 9. These findings demonstrate the potential of applying our MLLM evaluator to a broader range of tasks, both "open-ended" and "close-ended", highlighting its scalability to tasks beyond our benchmark.

Table 16: Success rates on open-ended single-app English tasks.

| Agent | Success Rate |
|---|---|
| *Agentic Workflow (GPT-4o)* | |
| AppAgent | 0.200 |
| AutoDroid | 0.300 |
| MobileAgent | 0.200 |
| MobileAgentV2 | 0.200 |
| M3A | **0.700** |
| T3A | 0.400 |
| SeeAct | 0.400 |

Table 16 presents the success rates of the seven agents on these ten tasks. M3A consistently outperformed the other agents. However, compared to the success rates reported in Table 3, MobileAgentV2 exhibited the largest performance gap, suggesting its limitations in handling "open-ended" tasks.

In future work, we aim to expand this initial experiment with a more comprehensive task collection to further improve and assess the feasibility of our MLLM evaluator for a wider range of tasks, and to investigate agent performance on "open-ended" tasks.

## H    CASE STUDY

Three case studies are presented to illustrate representative scenarios of task execution by agents. These include: (1) an invalid action taken by AppAgent due to misinterpretation of the UI structure in the XML file, (2) a dynamically changing screen without any action execution, repetitive actions due to the lack of reflection, and unrelated behaviours to the task description in MobileAgent, and (3) the combined actions employed by M3A.

### H.1    APPAGENT ON CONTACT_2 TASK

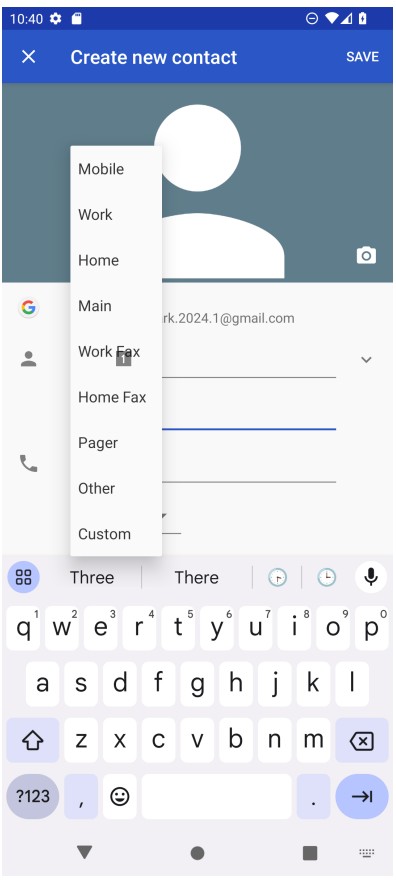

(a) Annotated screenshot

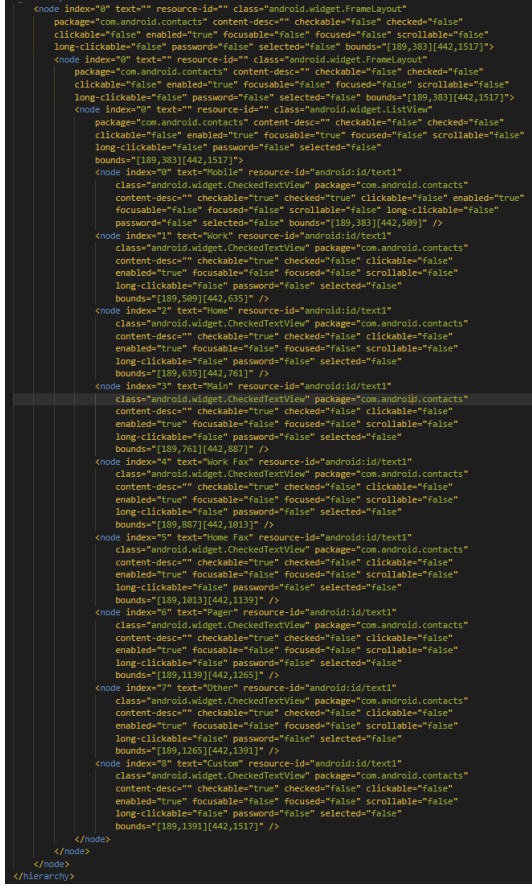

(b) Parsed XML file

Figure 11: The screenshot and XML file before the last action for AppAgent executing task *contact_2*. The model generated invalid action **tap(2)**. **Task description**: "Modify the last name of one of the contacts to 'Three'. Update the label for the contact's phone number to Work. Set the company to 'Huawei'. Add an email agent.benchmark.2024@gmail.com. Label the email as Work".

As shown in Figure 11, in the final step of task *contact_2*, AppAgent encountered a critical error due to a misinterpretation of the UI structure. The model incorrectly parsed the XML, treating the entire pop-up menu as a single element instead of recognizing each individual operable component, which reduced the number of widgets the agent could interact with. In addition, the agent executed an invalid action, **tap(2)**, targeting a non-clickable element. This issue highlights that an imperfect operable action detection mechanism may limit the agent's ability to navigate complex UI hierarchies and execute fine-grained interactions.

## H.2   MOBILEAGENT ON EXPEDIA_3 TASK

As shown in Figure 12 and Figure 13, MobileAgent's execution of task *expedia_3* reveals several noteworthy points: (1) Although the transition between the second and third screenshots (highlighted with a red border) lacks valid actions, the interface still changes, indicating that content is loading during a waiting period (i.e., a dynamically changing screen). (2) The agent generates repetitive actions despite no changes in the interface, but after several iterations, a correction occurs (highlighted with a blue border). (3) Interestingly, at the beginning of task execution, the agent initially attempted to chat with ChatGPT, which was unrelated to the task description. By the time the agent attempted to execute something relevant, several steps had already been wasted, leaving insufficient opportunities to complete the task properly.

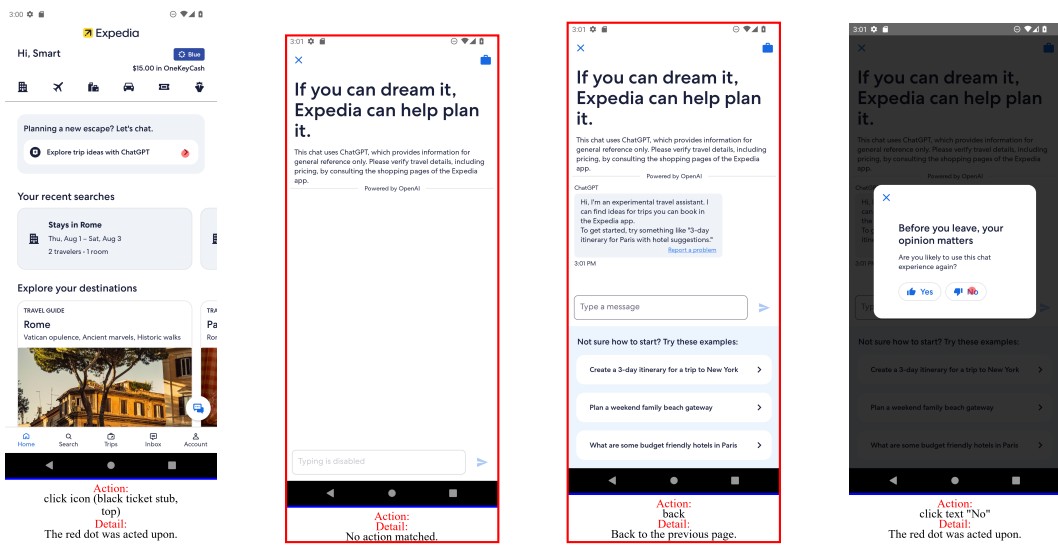

Figure 12: Trajectory of MobileAgent on expedia_3 (Part 1). **Task description**: "Check things to do in Paris. Get the search results for 25th to 28th of any month."

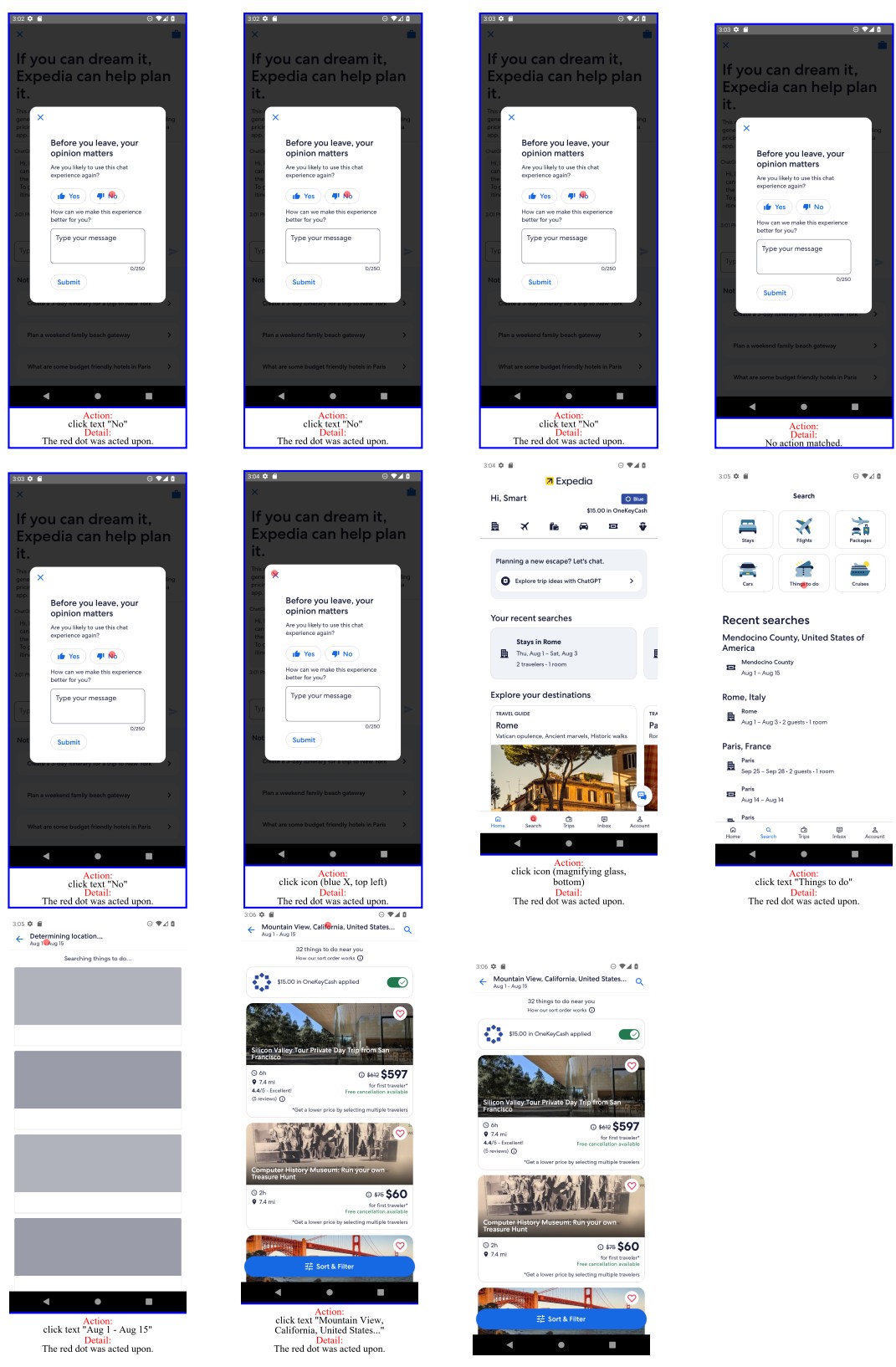

Figure 13: Trajectory of MobileAgent on expedia_3 (Part 2). **Task description**: "Check things to do in Paris. Get the search results for 25th to 28th of any month."

## H.3 M3A vs Human on google_tasks_0 task

(a) Trajectory of M3A on google_tasks_0

(b) Trajectory of Human on google_tasks_0

Figure 14: Trajectory of M3A vs human on google_tasks_0. **Task description**: "Create a new list 'Work'."

By comparing Figure 14a and Figure 14b, it is evident that M3A employed a combined action strategy, encapsulating text input and pressing the "enter" key within a single-step operation. This approach led to a more concise execution, requiring one fewer step compared to the human trajectory.

