# OpenReview forum: "SPA-BENCH: A COMPREHENSIVE BENCHMARK FOR SMARTPHONE AGENT EVALUATION"
_ICLR.cc/2025/Conference — ICLR 2025 Spotlight_

### Official Review · Reviewer_VmmX · 2024-11-04

**Soundness:** 3
**Presentation:** 3
**Contribution:** 2
**Rating:** 6
**Confidence:** 3

**Summary:**

This paper presents a benchmark for mobile device control agents, featuring the incorporation of Chinese applications in addition to English apps and newly proposed evaluation metrics regarding task completion. The benchmark includes a set of tasks that involve single and multi-app usage. The authors assess the performance of state-of-the-art agents developed through prompting or fine-tuning (e.g., agent fine-tuned with reinforcement learning). The valuable analyses in this work reveal the limitations of existing algorithms, particularly when handling tasks that span multiple applications or incorporate Chinese-language interfaces.

The key contributions of these works are benchmark task suites with both English and Chinese apps, additional evaluation metrics, and experiments with in-depth analyses of various agents.

**Strengths:**

S1 - This benchmark offers a notable addition to the community by employing Chinese applications. This expands the scope and applicability of benchmarking in multilingual contexts.

S2 - The benchmark includes diverse tasks, levels, and metrics. It also introduces several new evaluation metrics. Furthermore, a novel coarse-to-fine evaluation approach has been proposed.

S3 - The authors provided comprehensive benchmark references and included many of the existing baselines. Hence, this study offers a comprehensive comparison, enabling valuable insights. For example, it reveals the necessity of improved memory capabilities and lack of understanding in task termination.

**Weaknesses:**

W1 - This benchmark’s unique challenge is unclear. The authors should highlight the specific interesting challenge or novel difficulties it poses compared to existing benchmarks.

W2 - While the newly introduced termination metrics are intriguing and valuable, they seem to require more careful design. Could the authors propose a strategy for applying these metrics to agents that do not produce language-based rationales for termination? (Minor) Additionally, it would be helpful if the authors could underscore the significance of these metrics (e.g., why the ability to understand the termination of the tasks would be important when developing agents) while introducing the metric (i.e., Section 5.1).

W3 - The criteria for levels 1, 2, and 3 appear somewhat ambiguous. If the levels are based solely on the number of steps required, the division doesn't sound necessary. Could the authors clarify this distinction if these levels also account for the complexity of cross-app usage?

W4 - Several claims in the paper require further elaboration. For instance, the qualitative complexity of the a11y tree could be clarified by providing additional metrics, such as the number of tokens, as referenced in lines 367-371. Additionally, could the authors further elaborate on what they mean by "adequately grounding them in real-world context", as mentioned in lines 386-397?

W5 - It would be helpful to clarify the source of parsing errors. Are these issues arising from the agent's operations, or are they related to the environment? Additionally, there is a minor typographical error with a misplaced comma in line 449.

W6 - The paper lacks details on the specific prompts used for agents, and supplementary materials would facilitate the reproducibility of the experiments.

**Questions:**

Q1 - Could the authors elaborate on the unique challenge introduced by this benchmark? While it is clear that this work provides valuable references and novel attributes (e.g., evaluation metrics), a more explicit articulation of its key contributions would be helpful.

Q2 - Could the authors explain the rationale for the coarse-to-fine evaluation design? While this sounds helpful, I couldn't find a rationale for this design choice. Can the authors provide the reason for choosing this method? Also, what happens if the agent meets the second fine-grained part but misses the first coarse-grained part (or is it ensured that these cases don't usually occur)? A deeper insight into the reasoning and clarification behind these would offer a better understanding of the effectiveness of this approach.

Q3 - (Minor) For some experiments, English and Chinese languages are treated separately (e.g., human annotation in line 318). Could the authors clarify the motivation for this separation?

Q4 - In line 430, the term "shortcut" used by the M3A agent is somewhat ambiguous. Does this mean the agent generates a new action outside the predefined action space? If so, how is this interface managed within the environment? If not, clearer phrasing could avoid potential misunderstandings, such as stating, "The predefined action space includes combinations of actions that humans typically perform independently".

Q5 - Could the authors confirm if the fine-tuned agents were trained specifically on the dataset proposed in this benchmark? If they were fine-tuned, what factors might explain their low success rates? If they were not, wouldn't it be fair to fine-tune them when comparing the algorithmic performance? Although it might not be necessary to fine-tune them in this benchmark specifically, noting how they are trained (e.g., whether they are trained in this benchmark or not) seems necessary to be noted to avoid misleading the readers.


I note that I am highly willing to raise the overall scores.

---

> ### Author Response · Authors · 2024-11-21
>
> We sincerely thank the reviewer for their thoughtful feedback and provide our responses below:
>
> **W1 & Q1:**
>
> > unique challenges and key contributions
>
> We appreciate the reviewer’s feedback and agree that clarifying the unique challenges posed by our benchmark is critical. Below, we highlight the distinctive aspects of SPA-Bench across diverse and realistic task design, scalable agent framework, and automated evaluation pipeline, which set it apart from existing benchmarks (Table 1).
>
> **[Diverse and Realistic Task Design]**: The primary challenge addressed by SPA-Bench is creating tasks that reflect real-world smartphone usage, particularly those aligned with users’ daily routines. While existing benchmarks often focus on simplified or constrained scenarios, SPA-Bench evaluates agents across 66 apps, including 52 third-party apps, 7 Google Suite apps, and 7 system apps in English and Chinese contexts. This diversity introduces a wide variety of task complexities not seen in other benchmarks like AndroidArena [1], B-MoCA [2], or MobileAgentBench [3], which typically evaluate agents using system-level or Google Suite apps.
>
> Furthermore, AndroidWorld [4] covers fewer than 15 open-source apps, which are less realistic than and less representative of commonly used third-party apps. LlamaTouch [5], which includes nearly 30 third-party apps, features tasks that are mostly completed in fewer than 10 steps, and about one-third of its tasks involve simply opening an app and logging in. SPA-Bench, on the other hand, goes beyond these constraints to include realistic task flows, such as shopping on Amazon or posting on social media, which require agents to engage with app-specific features and UI layouts. This variety ensures a closer alignment with real-world smartphone usage patterns.
>
> **[Scalable Agent Framework]**: Adapting agents to new benchmarks often presents a bottleneck in evaluation. SPA-Bench addresses this challenge with a plug-and-play agent framework that enables seamless integration of agents with minimal adaptation. This capability significantly lowers the barrier to benchmarking new and emerging agents. Currently supporting 11 distinct agents, SPA-Bench represents a notable improvement over existing benchmarks. Additionally, our framework leverages multi-worker processes and emulator setups, ensuring both scalability and consistency in testing.
>
> **[Automated Evaluation Pipeline]**: Evaluation scalability is another key challenge in benchmarking agents across diverse tasks and apps. Unlike existing benchmarks, which often rely on manual evaluation or hand-crafted validation logic for specific tasks, SPA-Bench introduces an automated evaluation pipeline. This pipeline leverages screenshot-based success detection, requiring no additional human intervention for individual tasks when scaling up. It enables large-scale evaluations while significantly reducing resource and labor costs.
>
> In addition, the pipeline assesses agent performance across multiple dimensions, such as success rate, resource consumption, and termination accuracies. This multidimensional evaluation provides a comprehensive understanding of agents’ capabilities and limitations, guiding future research in agent design and benchmarking methodologies.

---

> ### Author Response · Authors · 2024-11-21
>
> **W2:**
>
> > propose a strategy for applying these metrics to agents that do not produce language-based rationales for termination
>
> We apologise if there was any misunderstanding, and we would appreciate clarification regarding the reviewer’s comment about “agents that do not produce language-based rationales”. As we understand it, the termination signal itself is not inherently tied to whether the agent or model is language-based. In our setup, all agents provide a language-based action to indicate termination. For example, AppAgent outputs an “Exit” action, while MobileAgent uses a “Stop” action. These actions explicitly indicate the agent’s belief that the task has been completed successfully, and we categorise them as self-reported completion. If an agent were not language-based, as long as it uses an alternative method to signal task completion (e.g., a distinct non-verbal signal or action), our termination metrics could still be applied. We welcome further clarification on this point to ensure we address the concern comprehensively.
>
> > underscore the significance of these metrics
>
> We thank the reviewer for this insightful suggestion. We agree that elaborating on the significance of these metrics in Section 5.1 enhances the paper, and we have incorporated these updates in the revised version. Currently, our termination metrics provide valuable insights into the reasons behind task completion and allow us to better evaluate discrepancies between an agent’s success rate and self-completion rate. Additionally, premature and overdue termination signals provide nuanced insights into agent behaviour.
>
> *Premature termination* impacts success rates by causing agents to incorrectly assume success before completing the task. For instance, as shown in Table 3, AutoDroid exhibits a nearly 50% premature termination rate in single-app English tasks. While this contributes to a lower mean step ratio on successful tasks, it also indicates a significant limitation: the agent prematurely halts tasks that are not yet truly successful. A more conservative self-evaluator, requiring stronger evidence of task completion, could mitigate this issue and improve overall success rates.
>
> *Overdue termination,* on the other hand, affects efficiency by causing agents to take unnecessary steps before concluding a task. For example, SeeAct has a high overdue termination rate, leading to an increased mean step ratio on successful tasks. This indicates the self-evaluator could be adjusted to be slightly more lenient towards classifying tasks as successful, reducing redundant steps while maintaining success rates.
>
> These metrics help diagnose the limitations of current self-evaluators and offer clear directions for improving agent design. By analysing the balance between premature and overdue terminations, we can refine self-evaluators to achieve both higher accuracy and greater efficiency.

---

> ### Author Response · Authors · 2024-11-21
>
> **W3:**
> > criteria for levels 1, 2, and 3 appear somewhat ambiguous; clarify this distinction if these levels also account for the complexity of cross-app
>
> We appreciate the opportunity to clarify how the levels are defined for both single-app and cross-app tasks, and they account for different considerations.
>
> *Single-app tasks:* The difficulty levels (Level 1, Level 2, and Level 3) are not determined solely by the number of steps required. Instead, they are designed as sets with progressively increasing complexity. Level 1 tasks serve as foundational, straightforward activities requiring the least number of actions. Level 2 builds on Level 1 by adding additional requirements, making the task more complex. Similarly, Level 3 introduces further requirements on top of Level 2, significantly increasing complexity. While higher-level tasks typically involve more steps for human execution due to these additional requirements, the levels are primarily defined by task complexity rather than step count alone.
>
> Compared to other benchmarks, AndroidWorld [4] (which utilises human annotators’ subjective ratings), MobileAgentBench [3] and LlamaTouch [5] (which categorises tasks solely based on step count), SPA-Bench adopts a structured, set-based approach to task design. Nevertheless, defining task difficulty remains an open question. A more ideal approach might involve multidimensional difficulty metrics that assess challenges posed to agents' specific abilities, such as memory, UI understanding, or handling animated interface. Exploring such multidimensional metrics represents a promising direction for future work.
>
> *Cross-app tasks:* Difficulty levels for cross-app tasks are defined differently, as these tasks are not organised into sets like single-app tasks. Instead, levels are determined by the number of apps involved in the task. Level 1 cross-app tasks require interactions between two apps, while Level 2 tasks involve switching among three apps. As the number of apps increases, tasks naturally become more complex due to the added inter-app dependencies, app-switching, and coordination, which also leads to a higher number of steps. The rationale for testing cross-app tasks is to provide a more realistic evaluation of app navigation and multitasking, as prior agents and benchmarks have primarily focused on single-app tasks.
>
> Additionally, Figure 9 in our paper presents the step distribution for both single-app and cross-app tasks by difficulty level. Interestingly, Level 1 cross-app tasks can sometimes require a similar number of steps as Level 3 single-app tasks. This highlights the distinct nature of cross-app tasks, where complexity arises from app-switching and inter-app dependencies rather than simply from additional linear requirements. We hope this explanation clarifies the criteria for task difficulty and its implications for evaluating agent performance.

---

> ### Author Response · Authors · 2024-11-22
>
> **W4:**
> > the qualitative complexity of the a11y tree could be clarified by providing additional metrics, such as the number of tokens
>
> To illustrate the complexity of the accessibility (a11y) tree, we compare three agents as an example: MobileAgentV2 (using raw screenshots only), M3A (using raw screenshots and a simplified a11y tree), and T3A (using the full a11y tree only). Our experiments shows that MobileAgentV2 used an average of 12,400 prompt tokens per step in Chinese single-app tasks, compared to 16,300 and 22,000 for M3A and T3A, respectively. In English single-app tasks, the token sizes were smaller for all three agents, with MobileAgentV2 using 11,200 tokens, M3A using 15,300 tokens, and T3A using 19,700 tokens. These differences highlight the greater complexity of the a11y tree especially for Chinese apps, which likely contributes to the degraded performance of the agents. We include these metrics in the table below to better support this claim.
>
>
> | Agent | Mean Prompt Tokens per Step (English) | Mean Prompt Tokens per Step (Chinese) | Mean Completion Tokens per Step (English) | Mean Completion Tokens per Step (Chinese) |
> |:-------------:|---------------------------------------:|---------------------------------------:|-------------------------------------------:|-------------------------------------------:|
> | AppAgent | 2064 | 1889 | 125 | 122 |
> | AutoDroid | 867 | 1290 | 161 | 162 |
> | MobileAgent | 9570 | 6403 | 98 | 120 |
> | MobileAgentV2 | 11230 | 12353 | 168 | 191 |
> | M3A | 15285 | 16346 | 66 | 75 |
> | T3A | 19736 | 21953 | 81 | 91 |
> | SeeAct | 5529 | 5812 | 927 | 1053 |
>
> > further elaborate on what authors mean by "adequately grounding them in real-world context"
>
> It refers to limitations observed in agents based on fine-tuned models. Specifically, these agents are often optimised for generating textual actions based on a fixed set of UI scenarios that are accurately demonstrated by humans. For instance, a tap action is typically deemed successful if the generated coordinates fall within 14% of the screen distance to the ground truth. While this may suffice in controlled offline environments, such errors in distance can prevent agents from correctly clicking on actionable elements in real-world settings, causing task failures. Moreover, reliance on fixed scenarios limits the agents' ability to adapt to new, unseen UI contexts or to recover from detoured states caused by mistaken actions.
>
> We acknowledge that this explanation could have been clearer in the original text, and we have updated the relevant section to better elaborate on these challenges and provide additional clarity in the revised version.
>
> **W5:**
> > It would be helpful to clarify the source of parsing errors. Are these issues arising from the agent's operations, or are they related to the environment?
>
> Parsing errors are issues arising from the agent's operations. For example, parsing errors occurred when the text parsing modules (e.g., using regex) failed to recognise the format of the core models' textual output, resulting in errors during the translation into valid actions. One specific instance is with AppAgent's parsing module, which requires the text output to include the exact phrase "Thought: ". If this phrase is missing from the model output, the parsing module cannot proceed, leading to an error.
>
> For further clarification, in Appendix C.3, we provided a detailed explanation of the two types of errors encountered during experiments: 1) Expected errors: These are agent-related issues, such as parsing errors or selecting actions outside the predefined action space; 2) Unexpected errors: These arise from external factors, such as environment malfunctions or network disruptions. For unexpected errors, we rerun the experiment to ensure fairness, while expected errors are recorded as part of the agent's performance.
>
> In addition, we acknowledge the typographical error and have addressd it in the revised version.

---

> ### Author Response · Authors · 2024-11-22
>
> **W6:**
> > The paper lacks details on the specific prompts used for agents, and supplementary materials would facilitate the reproducibility of the experiments
>
> To ensure transparency and reproducibility, we plan to open-source the entire benchmark codebase and data soon, including task sets, emulator snapshots, agent frameworks, agent modifications, and the evaluation pipeline.
>
> We understand the importance of providing details to facilitate reproducibility. In this benchmark, we brought together 11 agents, faithfully adhering to their original implementations, including model prompts available in their respective open-source codebases. We wish to clarify no agents or models were fine-tuned specifically for our task sets or Android environments (such as device size and system version), except for a few general and agent-specific configurations listed in Appendix F.1. These configurations include:
> 1. Core Model Adjustments: Proprietary models were switched to GPT-4o (or GPT-4o-mini for icon recognition in MobileAgentV2).
> 2. Auto-UI and DigiRL: Auto-UI-Base selected as core model.
> 3. Chinese Character Input: AdbKeyboard added to AppAgent, SeeAct, M3A, and T3A.
> 4. Auto-UI: Enabled “action history” and “chain of actions” features.
> 5. OdysseyAgent: Enabled action and screenshot history.
> 6. AppAgent and AutoDroid: No pre-experiment knowledge or exploration was permitted.
> 7. Execution Limits: All agents were limited to executing up to twice the “golden steps” per task.
>
> We made some additional adaptations to integrate agents into the experiment workflow within our benchmark:
> 1. Action Grounding: Applied action grounding modules to enable interaction with Android devices (Appendix C.2).
> 2. Data Logging: Logged executed actions, performance metrics, and errors during each agent’s execution loop (Appendix C.3).
> 3. Error Handling: Automatically retry and re-ran tasks in cases of unforeseeable issues (e.g., network failures, Android malfunctions) to ensure fair and accurate assessments (Appendix C.3).
> 4. Physical Device Compatibility: Modified AndroidWorld and AndroidEnv codebases to support physical smartphones for part of the Chinese app experiments.
> 5. Consistent Testing Environment: Emulators were configured with a Pixel 8 device frame and API Level 32 Android system image; physical devices used were Google Pixel 8.
>
> We hope the information provided above addresses the reviewer's concerns regarding the reproducibility of the experiments.

---

> ### Author Response · Authors · 2024-11-22
>
> **Q2:**
>
> We appreciate the reviewer's insightful question regarding the rationale and implementation of the coarse-to-fine evaluation design. We are pleased to provide a detailed explanation and address the concerns.
>
> > the rationale for the coarse-to-fine evaluation design
>
> This design is specifically used for single-app task evaluation, and its primary motivation is scalability and efficiency. Human evaluation is considered the gold standard but is prohibitively expensive in terms of time and monetary cost, as seen in approaches used by AppAgent and MobileAgent, which only evaluate a limited set of tasks (fewer than 50). Similarly, hand-crafted validation logic, while effective, requires significant human effort when scaling to larger task sets. To address this, we propose a scalable approach where GPT-4o is used for fine-grained evaluation, aiming to align closely with human evaluation in accuracy.
>
> However, GPT-4o-based evaluation itself also incurs monetary costs. Some agents may perform poorly on certain tasks, such as failing to produce any relevant actions or only partially completing the task. In such cases, applying GPT-4o evaluation would be excessive and resource-intensive. To address this, we introduced a filtering mechanism—coarse detection—that screens out trajectories far from being successful, thereby improving the overall efficiency of the evaluation process.
>
> The design of the coarse filter is inspired by our observations and insights from MobileAgentBench [3]. We noticed that while multiple paths may exist to successfully complete a task (i.e., different action sequences), the final state or screenshot typically converges to a similar outcome. This observation led us to develop the concept of “key components”, which are essential textual elements that must appear in the final state of a successful task. For example, in the task "Set an alarm for 8am", the key component would be “8” rather than “8” and “am”, since UI variations may present “8:00” instead of “8:00 am”. These key components are annotated to ensure universality across different devices, Android versions, and UI designs.
>
> While key component annotation requires initial human effort, it is reusable and can reduce the cost of fine-grained detection in large-scale evaluations, improving efficiency. Additionally, while the coarse-to-fine design involves human involvement in coarse-detection preparation, the process can be skipped if necessary, relying solely on fine-grained detection. This approach, though more expensive, ensures scalability without human intervention, albeit with slightly reduced efficiency.
>
> > What happens if the agent meets the second fine-grained part but misses the first coarse-grained part
>
> If an agent fails the coarse-grained detection, the task is considered unsuccessful, and fine-grained detection is not applied. We have taken extensive measures to ensure the robustness of key components through cross-validation across different devices, Android versions, and annotators, minimising the risk of false negatives in coarse detection. This ensures that the coarse-to-fine design does not decrease the performance compared to using fine detection alone.
>
> Interestingly, when evaluating our coarse-to-fine detection, we observed rare cases where trajectories passed the fine-grained detection but failed the coarse-grained detection. These cases were attributable to false positives in the fine-grained detection. In such situations, the coarse detection helps prevent incorrect classifications by filtering out these false positives, contributing to a more reliable evaluation process. This also explains why, in some cases, the coarse-to-fine method achieves slightly higher accuracy compared to fine-grained detection alone.
>
> Thank the reviewer again for highlighting this important aspect, we hope this explanation clarifies the reasoning and benefits of the coarse-to-fine evaluation approach, as well as how it ensures both scalability and reliability.

---

> ### Author Response · Authors · 2024-11-22
>
> **Q3:**
>
> We thank the reviewer for the thoughtful question. In the revised version, we have also added the overall performance in Table 2 to provide a clearer and more comprehensive picture.
>
> > English and Chinese languages are treated separately (e.g., human annotation in line 318). Could the authors clarify the motivation for this separation?
>
> We treated English and Chinese apps separately because Chinese apps present unique challenges that significantly impact agent performance, and we wanted to verify these differences through our experiments. These challenges include: 1) Language Barriers: Most models and agents are primarily optimised for English, and their performance in Chinese is generally less robust due to differences in language structure and the availability of training data; 2) UI Design Differences: Chinese apps often feature more intricate layouts, animations, pop-ups, and advertisements compared to English apps, which makes UI understanding and action grounding more complex. Our experiments confirmed that agents performed noticeably worse on Chinese apps, as shown in Table 2. Similarly, we evaluated our success detection pipeline separately for English and Chinese apps. This was motivated by the same factors, as the unique challenges of Chinese apps also affect success detection, requiring separate evaluations to ensure accuracy and fairness.
>
> **Q4:**
>
> > the term "shortcut" used by the M3A agent is somewhat ambiguous; clearer phrasing could avoid potential misunderstandings
>
> We acknowledge that the term "shortcut" conflates two distinct concepts: (1) "combined actions", where agents like M3A execute a single predefined action that encompasses multiple operations (e.g., typing text and pressing "enter" as one action), and (2) "strategic shortcuts," where agents intentionally select an efficient alternative action (e.g., clicking on a recommended item instead of using the search bar) based on the context of the task (e.g., line 431).
>
> To avoid misunderstandings, we agree that the term "combined action" is more precise for the first case and have updated the terminology in the revised version of the paper. To clarify, the "combined actions" executed by M3A are predefined within the original action space of AndroidWorld and M3A’s implementation. These are not "new actions" generated dynamically by the agent, and were designed to mirror typical human behaviours for efficiency.

---

> ### Author Response · Authors · 2024-11-22
>
> **Q5:**
>
> > Were the fine-tuned agents were trained specifically on the dataset proposed in this benchmark?
>
> Thank the reviewer for raising this important question. We would like to confirm that none of the fine-tuned agents were specifically trained on the dataset proposed in this benchmark. We apologise for any ambiguity in our terminology. When we referred to "fine-tuned model-based agents," we meant agents whose original implementations were built using fine-tuned models.
>
> We believe our comparison is fair, as all agents were tested on unseen tasks, aligning with the rationale of this benchmark to evaluate agents' generalisation abilities. That said, we acknowledge that agents based on fine-tuned models inherently face certain disadvantages compared to those using off-the-shelf models (GPT-4o), which are designed with broader generalisation capabilities. Fine-tuned agents often excel in tasks closely resembling their training data, but this benchmark aims to assess whether they can also perform well in real-world, end-to-end interactions with Android devices. This evaluation complements existing research by exploring their performance beyond the datasets for which they were specifically fine-tuned on. Therefore, we included four fine-tuned model-based agents alongside seven off-the-shelf model-based agents, aiming to provide a more comprehensive comparison of their capabilities in handling real-device tasks.
>
> > noting how they are trained seems necessary to be noted to avoid misleading the readers
>
> Regarding the training datasets for these agents:
>
> - Auto-UI, CogAgent, and DigiRL were fine-tuned using the Android-In-The-Wild (AITW) dataset.
> - CogAgent was additionally fine-tuned on manually collected data, Mind2Web, and multiple publicly available visual question-answering datasets.
> - OdysseyAgent was fine-tuned on its self-proposed dataset, which focuses on cross-app smartphone control tasks.
>
> Both AITW and OdysseyAgent datasets share some similarities with our benchmark in terms of task description styles. However, AITW sometimes includes simpler questions (e.g., “What is the capital of Japan?”) or tasks requiring fewer steps (e.g., “Open a new Chrome tab”) compared to our more complex tasks. Additionally, our benchmark increases the difficulty by requiring real-device interactions instead of offline simulations and by integrating tasks across diverse contexts, including Chinese apps. These aspects introduce additional complexity compared to datasets like AITW and GUI-Odyssey.
>
> We appreciate the suggestion to ensure clarity regarding the context and challenges faced by these agents in our benchmark. In Section 6.1 (“Impact of Core Models and Input Modalities”), we provided a more detailed analysis of the limitations of fine-tuned agents in real-world applications. Additionally, in our response to W4, we elaborated on the potential reasons for their performance gaps and the challenges of adapting to our benchmark tasks.
>
> [1] Xing, Mingzhe, et al. "Understanding the weakness of large language model agents within a complex android environment." Proceedings of the 30th ACM SIGKDD Conference on Knowledge Discovery and Data Mining. 2024.
> [2] Lee, Juyong, et al. "Benchmarking Mobile Device Control Agents across Diverse Configurations." arXiv preprint arXiv:2404.16660 (2024).
> [3] Wang, Luyuan, et al. "MobileAgentBench: An Efficient and User-Friendly Benchmark for Mobile LLM Agents." arXiv preprint arXiv:2406.08184 (2024).
> [4] Rawles, Christopher, et al. "AndroidWorld: A dynamic benchmarking environment for autonomous agents." arXiv preprint arXiv:2405.14573 (2024).
> [5] Zhang, Li, et al. "LlamaTouch: A Faithful and Scalable Testbed for Mobile UI Automation Task Evaluation." arXiv preprint arXiv:2404.16054 (2024).
>
> We hope our responses address the reviewer's concerns and remain open to any further discussion.

---

> > ### Comment · Reviewer_VmmX · 2024-11-25
> >
> > Thanks for sincerely caring and clearly responding to many of my questions! Many of my concerns have been addressed, and I have raised my rating from 3 to 5. While several clarifications have not yet been included in the revised version, I assume that the authors will include these in future revisions. Regarding my remaining concerns, I have left some further comments below. I hope my suggestions are helpful in making your work more solid.
> >
> > **W1 & Q1** - I highly value your benchmark and contributions, but I still do think the benchmark lacks a unique challenge that other benchmarks do not address. One cautious suggestion might be to create 'open-ended tasks' since the benchmark exploits evaluators are free of handcrafted.
> >
> > **W2** - My question was not about the baselines you experimented with. My concern is for other (probably developed in the future) algorithms that might not incorporate task completion in their action space. Would there be any method to consider task completion for those (yet explored) baselines? Or do you think that signaling task completion from the agent is a ‘must-included’ component when developing an agent?
> >
> > **W3** - I first note that I am addressed on this point. On top of your clarification, would incorporating a handful of human annotators (e.g., 5-10 numbers of people) to create more solid criteria be meaningful, in your opinion?
> >
> > **Q5** - I agree that your justification of the experiment setup is fair to a certain degree, but I am still not fully addressed on this setup. The main concern is that we need to be careful to term “unseen tasks” while handling proprietary LLMs (as they might have seen them). I still do not see high value in the results of evaluating open-sourced agents that are not fine-tuned in the proposed benchmark. Considering the difference in the data and parameter scale used for training, both groups do not seem comparable. My suggestion is either (1) not to compare the results from open-source models with those from foundation models (regarding a paragraph starting from line 475) or (2) fine-tune the open-source models, at least in a representative subset of tasks.

---

> > > ### Author Response · Authors · 2024-11-25
> > >
> > > We sincerely appreciate the reviewer’s feedback on our previous response. Regarding the paper revisions, we will soon update a new version with additional clarifications and inform the reviewers once it is ready.
> > >
> > > **Q1 & W1**
> > > > One cautious suggestion might be to create 'open-ended tasks' since the benchmark exploits evaluators are free of handcrafted.
> > >
> > > Thank the reviewer for the constructive feedback and for valuing our benchmark and contributions. We have expanded our tasks sets by a small number of single-app English open-ended tasks, and are currently experimenting them with GPT-4o-based agents to explore whether our evaluators can demonstrate scalability to such tasks. We hope these initial experiments will further illustrate the ability of our evaluation framework to effectively evaluate open-ended tasks, an area that poses a unique challenge and may not be adequately addressed by other existing works. Once these results are ready, we will update with the reviewer. In the future, we plan to extend this work to include a broader and more diverse set of tasks, including additional open-ended scenarios.
> > >
> > > **W2**
> > > > Would there be any method to consider task completion for those (yet explored) baselines? Or do you think that signaling task completion from the agent is a ‘must-included’ component when developing an agent?
> > >
> > > We thank the reviewer for the thoughtful question. We believe signaling task completion is essential for an agent. Without it, the agent would continue running indefinitely, even if the task is completed, unless externally stopped (e.g., by a maximum step limit). This is why we expect agents to have a built-in task completion signal.
> > >
> > > However, if an agent does not include a built-in task completion signal, our evaluator could act as an external mechanism for determining task completion. In such case, the evaluator could check after each step whether the task is complete and stop the agent if necessary. Nevertheless, this would also introduce additional costs and reduce the significance of premature and overdue termination metrics, as they are designed to assess the alignment between the agent's self-reported completion with the evaluator’s success signal.
> > >
> > > **W3**
> > > > On top of your clarification, would incorporating a handful of human annotators (e.g., 5-10 numbers of people) to create more solid criteria be meaningful, in your opinion?
> > >
> > > We appreciate the reviewer’s question. We believe that incorporating more human annotators could indeed be helpful and lead to more robust criteria under certain conditions. Engaging a small group of annotators involved in the feedback process during the pre-annotation stage could refine both the criteria and annotation guidelines. Additionally, as the complexity of difficulty levels increases, having more annotators will be essential to ensure the scalability and consistency of large-scale annotation. In future work, we plan to explore more multi-dimensional difficulty levels, and we anticipate that involving additional annotators will further enhance the criteria and improve the overall quality of the annotations.

---

> > > ### Author Response · Authors · 2024-11-25
> > >
> > > **Q5**
> > > > The main concern is that we need to be careful to term “unseen tasks” while handling proprietary LLMs (as they might have seen them).
> > >
> > > We apologise for any ambiguity and agree that we need to be more careful with terminology. By "unseen tasks", we meant that our benchmark was not specifically used to fine-tune the models using human annotations (e.g., task screenshots). However, we agree that there may still be a chance that the models, whether proprietary or open-source, have encountered these tasks in some form.
> > >
> > > > I still do not see high value in the results of evaluating open-sourced agents that are not fine-tuned in the proposed benchmark. Considering the difference in the data and parameter scale used for training, both groups do not seem comparable. My suggestion is either (1) not to compare the results from open-source models with those from foundation models (regarding a paragraph starting from line 475) or (2) fine-tune the open-source models, at least in a representative subset of tasks.
> > >
> > > We acknowledge the inherent disadvantages open-source models face compared to proprietary models in our experimental setup. Our original motivation was to explore the performance gap between these two groups in the context of smartphone agent control, where the models act as the core controllers. Our results, therefore, reflect the performance differences between GPT-4o and its open-source counterparts in this specific setting, rather than offering a broader comparison.
> > >
> > > We also recognise that this part of the experiment lack comprehensiveness. In light of the reviewer’s feedback, we have decided to adopt the first suggestion and remove the comparison from our paper for now. In future work, we plan to explore the second suggestion by fine-tuning the open-source models, enabling more meaningful comparisons.
> > >
> > > We are grateful once again for the reviewer’s feedback and hope our clarification addresses more of the concerns. As mentioned earlier, once the revision is updated, we will notify the reviewer. We aim to better demonstrate the quality and clarity of our work and always welcome any further discussions.

---

> > > ### Author Response · Authors · 2024-12-01
> > > **Follow-Up on Reviewer VmmX Comments**
> > >
> > > Dear Reviewer VmmX, as the discussion period approaches its conclusion on 2nd December, we would like to kindly ask if you have had an opportunity to review our response and revised paper. If you have any remaining concerns or suggestions that you would like us to clarify, we would be grateful for your feedback. Your insights are highly valuable to us, and we hope that our revisions have addressed your concerns and questions, making our work more solid and beneficial for the research community.
> > >
> > > Regarding the novelty and unique challenges of our work, we would like to provide a brief summary. Our aim is to offer a comprehensive benchmark for evaluating smartphone control agents, focusing on how they perform real-world tasks that reflect the daily routines of actual human users. Compared to existing works, our benchmark includes:
> > > - **Broader Coverage of Tasks**: Tasks derived from a wider range of third-party apps, including both English and Chinese apps, spanning single-app and cross-app scenarios.
> > > - **Extensive Evaluation of Agents**: Evaluation of the largest number of agents among similar works, using their original implementations, to highlight the plug-and-play feature of our framework and its compatibility with future agents.
> > > - **Comprehensive Metrics**: Seven evaluation metrics designed to provide more detailed and informative insights compared to existing benchmarks.
> > > - **Automation and Scalability**: Automated evaluation methods that are free of human intervention, unlike other works relying on handcrafted validation logic or manual evaluation.
> > >
> > > Initially, our work focused on “closed-ended” tasks, as these are the main focus of most existing benchmarks. “Open-ended” tasks, typically reliant on manual evaluation, were planned as a future extension. Your suggestion to test our evaluator’s ability on “open-ended” tasks was particularly helpful and aligns with our efforts to improve SPA-Bench. Due to time constraints, in the revised paper, we conducted an initial experiment with 10 newly designed “open-ended” tasks. The results, detailed in Appendix G, demonstrate our evaluator’s potential to handle such tasks in a fully automated manner. This capability further distinguishes our benchmark from others.
> > >
> > > We would like to thank you once again for your time and constructive feedback, which has helped us to revise and strengthen our work. If there are any remaining concerns or points you feel we should address, we would be happy to respond further. We look forward to hearing from you.

---

> > > > ### Comment · Reviewer_VmmX · 2024-12-02
> > > >
> > > > Thanks for the clarification and efforts to address the raised concerns. Appending extra open-ended tasks in Appendix G can be meaningful for emphasizing the strength of your evaluators. Appreciating your hard work, I raised my score from 5 to 6. I would like to see your (maybe another) future work incorporating directions you have considered (like, open-ended tasks)!

---

> ### Author Response · Authors · 2024-11-28
>
> We would like to express our gratitude once again for the reviewer’s constructive feedback. We have uploaded the latest paper revision and continue to improve our work.
>
> Regarding the previous **W1 & Q1**, we wanted to share that we have conducted an initial experiment, detailed in Appendix G, to further demonstrate the unique scalability of our evaluator, including for "open-ended" tasks. We acknowledge that this initial experiment is small in scale, and we aim to expand and refine it in our future work. We hope this can be recognised as a unique challenge that distinguishes our benchmark from others, and better showcases what we address: a scalable benchmark with diverse task scenarios that closely reflect real-world smartphone control tasks encountered in daily user routines.
>
> As the paper revision deadline approaches, we would greatly appreciate any further comments from the reviewer and will do our best to address them. Additionally, we would also be delighted to engage in further discussion after the revision deadline to ensure the quality and clarity of our work.

---

### Official Review · Reviewer_UgAV · 2024-11-05

**Soundness:** 3
**Presentation:** 3
**Contribution:** 3
**Rating:** 8
**Confidence:** 4

**Summary:**

The present work introduces a new smartphone agent benchmark called “SPA-BENCH.” The motivation for this research stems from the rapid development and increasing adoption of (multimodal) LLM-powered smartphone agents. SPA-BENCH seeks to address the challenges of evaluating these agents by providing a structured evaluation environment that simulates real-world conditions. SPA-BENCH includes a diverse set of tasks that cover both single-app and cross-app tasks on system and third-party applications, catering to both English and Chinese users. The authors also conduct extensive experiments with 7 off-the-shelf LLM agents and 4 finetuned LLM agents, highlighting the key insights derived from these experiments.

**Strengths:**

1.This paper studies an important and timely problem. Developing a comprehensive benchmark is important to advance LLM agents in smartphone applications.

2.The present work includes comprehensive experiments with 11 LLM agents on diverse tasks. The findings derived from these experiments could provide insights for future LLM agent design.

3.This paper introduces a plug-and-play framework, which could facilitate real-time interaction with Android devices. It is important to support diverse tasks and minimize the gap to real-world usage.

4.The design of the proposed benchmark is adequately motivated. In general, the paper is well-written.

**Weaknesses:**

1.The evaluation tasks are constructed by human annotators. It is fine, but I think the present work will benefit a lot if the authors could discuss more about the process of recruiting human annotators and protocols for quality control. It will also be great if the authors can discuss the possibility of synthesizing behavior trajectory, which might extend the impact of present work from solely evaluation to fine-tuning or even pre-training.

2.In addition to off-the-shelf LLM agents and fine-tuned agents, it is also important to evaluate locally deployable agents and cloud-based agents [1], which is particularly relevant for mobile agent setting. I suggest the authors to re-organize the taxonomy of agents as two main categories, i.e., agent-as-a-model (fine-tuned or pre-trained models customized for agentic tasks) [2] and agentic workflow (usually based on off-the-shelf model, possibly with modular design) [3].

3.Can SPA-bench be applied to a broader range of smart devices? Such as tablets with Android systems or IOS devices. I think it is important to discuss the potential and challenges here.

4.The authors should also elaborate on the main differences of smartphone agents compared to PC agents or cloud-based agents. It provides important insights for context of application.

[1] Gunter, Tom, et al. "Apple intelligence foundation language models." arXiv preprint arXiv:2407.21075 (2024).

[2] Lai, Hanyu, et al. "AutoWebGLM: Bootstrap And Reinforce A Large Language Model-based Web Navigating Agent." arXiv preprint arXiv:2404.03648 (2024).

[3] Shang, Yu, et al. "AgentSquare: Automatic LLM Agent Search in Modular Design Space." arXiv preprint arXiv:2410.06153 (2024).

**Questions:**

Please refer to the Weaknesses above.

---

> ### Author Response · Authors · 2024-11-21
>
> We sincerely thank the reviewer for their thoughtful feedback and provide our responses below:
>
> **Weakness 1:**
>
> >the process of recruiting human annotators and protocols for quality control
>
> In our study, the main annotation was conducted by the joint first authors, with assistant annotation from coauthors who are also regular users of the relevant mobile apps and are well-acquainted with their core features. To ensure consistency and clarity, the main annotators developed detailed guidelines for the assistants, referencing similar works to establish standards on key considerations such as feature selection, task difficulty levels, and trajectory collection processes, accompanied by illustrative examples. Quality control was further reinforced through cross-checking trajectories among annotators, verifying both the appropriateness of tasks and the accuracy of trajectories. Additionally, we validated each task’s feasibility across different Android devices to ensure reproducibility of results with the same steps.
>
> >the possibility of synthesizing behavior trajectory
>
> We agree that this is a promising avenue to enhance the impact of our benchmark. While direct trajectory synthesis remains challenging due to its similarity to the smartphone agent’s task itself, we are exploring some potential extensions to support trajectory collection, with the goal of achieving these advancements and addressing the associated challenges in future work.
>
> *Task Decomposition and Composition:* Starting with an initial set of human-annotated tasks as a foundation, automated methods (e.g., MLLMs) could try to split larger tasks into smaller sub-tasks or combine small tasks to form more complex ones. However, a potential challenge is the limited UI understanding capabilities, as observed in some of the agent experiments in this benchmark, can significantly impact the quality of these operations.
>
> *Automated Task Generation:* Task descriptions could be generated automatically, and agent-execution trajectories selected based on performance metrics. This would allow us to identify and retain successful trajectories while minimising manual intervention. However, a potential limitation is the bias introduced during trajectory selection. Trajectories that are easier and more suitable for existing agents to accomplish are more likely to be retained, potentially narrowing the diversity of the dataset.

---

> ### Author Response · Authors · 2024-11-21
>
> **Weakness 2:**
> > evaluate locally deployable agents and cloud-based agents
>
> Thank the reviewer for highlighting this relevant consideration for mobile agent systems. We agree that evaluating locally deployable and cloud-based agents would enrich the scope of our work. While our framework could support these agents with some adaptation, there are several challenges that currently limit full integration.
>
> The primary issue lies in the fact that many of these agents are not open-sourced or do not provide accessible APIs, making their inclusion challenging. If these agents become open-sourced or provide APIs in the future, we are eager to integrate them into our framework. For instance, locally deployable agents could be supported by adding a bridging module to facilitate communication. This module would handle task descriptions, generate trajectories, and store the results, which could then be automatically evaluated using our existing pipeline.
>
> Additionally, industrial smartphone agents, such as Apple Intelligence and HONOR YOYO Assistant, present unique challenges for benchmarking. Many of these agents are still in beta versions and are often deployed on specific devices or operating systems with varying sets of supported applications. These differences raise concerns about fairness and scalability in performance comparisons. Furthermore, without system simulation or control mechanisms like Android Emulators and Snapshots, it becomes more challenging to ensure reproducibility and scalability of experiments.
>
> We acknowledge these limitations and agree that, with more support and resources in the future, extending our experiments to include locally deployable and cloud-based agents would be a valuable direction to pursue.
>
> > re-organize the taxonomy of agents as two main categories, i.e., agent-as-a-model and agentic workflow
>
> We sincerely appreciate the reviewer for the thoughtful suggestion, as it offers a valuable perspective for refining the taxonomy and enhancing the clarity of our presentation.  We agree that off-the-shelf agents naturally align with the "agentic workflow" category, while fine-tuned agents can be categorised under "agent-as-a-model". For example, fine-tuned agents often rely no separated visual perception modules compared to off-the-shelf agents, with their models specifically customised for smartphone control tasks. To better reflect this distinction, we have re-organised the taxonomy in the revised paper.

---

> ### Author Response · Authors · 2024-11-21
>
> **Weakness 3:**
>
> > Can SPA-bench be applied to a broader range of smart devices?
>
> We thank the reviewer for this insightful question. While SPA-Bench is currently designed for Android smartphones, its modular architecture offers significant potential for adaptation to other platforms, such as Android tablets and iOS devices. The evaluation pipeline is generalisable, requiring task descriptions, agent-executed screenshots, and relevant log data for performance metrics. With appropriate adaptations to collect these data points on other platforms, SPA-Bench could evaluate agents designed for a broader range of smart devices.
>
> For example, adapting SPA-Bench to iOS would require the development of equivalent modules for action execution and interface information extraction, similar to the Android Debug Bridge (ADB) and Android Emulators used in our current setup. Additionally, while we have not explicitly tested on larger Android devices such as tablets, the Android tools utilised in SPA-Bench are inherently compatible with these devices. This assumption is supported by peer work like GUI-Odyssey [1] and B-MoCA [2], which also tested agents on tablet devices. However, a key challenge in broadening SPA-Bench's applicability lies in the availability of agents for platforms like iOS. Most current open-source agents are Android-specific, leveraging its accessibility features and open ecosystem. As agents for other platforms become available, SPA-Bench can be extended to support them effectively, aligning with our goal of creating an inclusive and scalable framework.
>
> **Weakness 4:**
>
> > the main differences of smartphone agents compared to PC agents or cloud-based agents
>
> *Smartphone control agents vs. PC control agents:* The key differences lie in the action space, the complexity of UI navigation and the task scope. Smartphone control agents are typically limited to touch-based actions, such as tapping and swiping. In contrast, PC control agents interact with both a mouse and a keyboard, allowing for more diverse input methods, including drag-and-drop, right-click, and keyboard shortcuts. Furthermore, the larger screen size and higher density of actionable UI elements on PCs add significant complexity to navigation and task execution compared to smartphones, where UI layouts are often simplified for smaller screens. Regarding the task scope, smartphone agents primarily focus on app-based daily tasks, such as messaging, media consumption, and online shopping, whereas PC agents are often designed for more professional and productivity-focused tasks. For example, in OSWorld [3], 31.7% of tasks involve office applications, including document processors and slide editors, as well as terminal-based and image tools. These differences in task scope highlight how smartphone agents cater to everyday routines, while PC agents often address workplace and business-related applications. Despite these differences, both types of agents share the challenge of generalising across diverse environments and handling tasks involving multi-step workflows.
>
> *Locally deployable smartphone agents vs. cloud-based smartphone agents:* The primary differences here concern the size of the agent model, latency and privacy consideration. Locally deployable agents typically use smaller models optimsed for on-device deployment, which allows them to operate with minimal latency in regions with unreliable internet connectivity. Cloud-based agents, on the other hand, often utilise larger models with extensive computational resources, enabling them to achieve superior performance. Additionally, locally deployable agents process all data on the device, ensuring greater privacy, as sensitive information is not transmitted to external servers. Despite these differences, both types of agents share common goals of efficient task execution, real-time decision-making, and adaptability to diverse app environments.
>
> [1] Lu, Quanfeng, et al. "GUI Odyssey: A Comprehensive Dataset for Cross-App GUI Navigation on Mobile Devices." arXiv preprint arXiv:2406.08451 (2024).
> [2] Lee, Juyong, et al. "Benchmarking Mobile Device Control Agents across Diverse Configurations." arXiv preprint arXiv:2404.16660 (2024)
> [3] Xie, Tianbao, et al. "Osworld: Benchmarking multimodal agents for open-ended tasks in real computer environments." arXiv preprint arXiv:2404.07972 (2024)
>
> We hope our responses address the reviewer's concerns and remain open to any further discussion.

---

> > ### Comment · Reviewer_UgAV · 2024-11-26
> >
> > Thank you for your response. It addresses most of my concerns. I am happy to keep an 8 score.

---

### Official Review · Reviewer_K4f8 · 2024-11-06

**Soundness:** 3
**Presentation:** 3
**Contribution:** 3
**Rating:** 8
**Confidence:** 3

**Summary:**

The paper introduces Spa-Bench, a comprehensive benchmark designed for evaluating smartphone agents across a variety of tasks in both English and Chinese languages. This benchmark supports over 340 tasks that span single-app and cross-app scenarios, covering different app types and real-world complexities such as third-party app interactions and varying difficulty levels. SPA-BENCH also integrates a plug-and-play framework that supports 11 agents and features a scalable, automated evaluation pipeline. This pipeline assesses agent performance based on success rates, efficiency, resource consumption, and reliability, eliminating the need for handcrafted evaluation logic. Experiments reveal that while proprietary (M)LLM-based agents outperform open-source counterparts, they still face challenges in real-world application due to time and cost inefficiencies. The paper emphasizes the need for advancements in UI understanding, memory retention, and error handling to improve the practicality of autonomous smartphone agents. I raise an acceptance of this paper.

**Strengths:**

1. Diverse task collection of 340 tasks

2. Automated evaluation pipeline consists of multiple evaluation metrics

**Weaknesses:**

Paper can be better organized.

**Questions:**

1. Paper can be better organized. For example, there is a very large blank space between line 188 and 191. These large blank spaces need to be fixed.

2. In the paper, authors mention the human-executed trajectory to avoid shortcuts. Why do we need to avoid the shortcuts? I think it is not a bad choice if the agent is able to find a shortcut solving the problem.

---

> ### Author Response · Authors · 2024-11-21
>
> We sincerely thank the reviewer for their thoughtful feedback and provide our responses below:
>
> **Question 1:**
> > Paper can be better organized.
>
> We thank the reviewer for the suggestion. We have uploaded a revised version of our paper, which we hope offers improved organisation.
>
> **Question 2:**
>
> > In the paper, authors mention the human-executed trajectory to avoid shortcuts. Why do we need to avoid the shortcuts? I think it is not a bad choice if the agent is able to find a shortcut solving the problem.
>
> We appreciate the reviewer for raising this important point about shortcuts. We fully agree that an agent's ability to identify and utilise shortcuts is a valuable strength, and we would like to clarify the treatment of shortcuts in both human- and agent-executed trajectories within our benchmark.
>
> *Human-executed trajectories:* Shortcuts are inherently dynamic and influenced by factors such as recommendation algorithms, or user-specific history. Therefore, to ensure a reproducible and unbiased baseline, we ask human annotators to avoid shortcuts when creating the "golden steps". Allowing shortcuts at this stage could result in human trajectories appearing shorter than the typically necessary steps required to complete the task.
>
> *Agent-executed trajectories:* During task execution, agents are permitted to take up to twice the number of steps in the human-defined golden trajectory. By excluding shortcuts from the "golden steps", we ensure that agents are evaluated fairly against a consistent baseline, regardless of whether shortcuts are available in specific scenarios. At the same time, we fully support and encourage agents using shortcuts during execution, as this reflects their ability to optimise performance when a more efficient option is available.
>
> We hope our responses address the reviewer's concerns and remain open to any further discussion.

---

> > ### Comment · Reviewer_K4f8 · 2024-11-26
> > **Acknowledge the changes and keep score unchanged**
> >
> > I acknowledge the changes and will keep my score as 8

---

### Author Response · Authors · 2024-11-22
**Summary of Our Responses & Paper Revisions**

**[Edit on 03 Dec]**

We would like to express our sincere gratitude once again to all the reviewers for their constructive feedback, which has greatly helped us improve the quality and clarity of our work. The final revised version of the paper incorporates the reviewers' suggestions to enhance both clarification and readability. We hope our work will contribute to the smartphone control agent research community and help in the development of better agents. We also look forward to continuing our work in this area in the future.

---

We sincerely thank all reviewers for their thoughtful feedback and constructive suggestions, which have greatly contributed to improving our paper. In our rebuttal, we have striven to address each question and concern with clarity, providing additional explanations, experimental data, and updates to the paper where necessary. Below, we summarise the key points of our responses and revisions:

1.  **Paper Organisation and Clarity**
    In response to feedback regarding the organisation of the paper, we have revised its structure to enhance clarity and logical flow. Specific sections, such as the taxonomy of agents and the limitations of fine-tuned agents in real-world settings, have been updated for improved readability and precision.

2.  **Shortcuts in Human-Executed and Agent-Executed Trajectories**
    We clarified the role of shortcuts, explaining that human annotators avoid shortcuts to establish a reproducible and unbiased baseline. Agents, however, are encouraged to use shortcuts during execution to reflect their optimisation capabilities. This ensures fair evaluation while promoting agents’ efficiency.

3.  **Unique Challenges and Key Contributions**
    We emphasised SPA-Bench’s unique contributions, including diverse and realistic task design, a scalable agent framework, and an automated evaluation pipeline. We compared SPA-Bench to existing benchmarks, showcasing its distinct advantages in handling third-party apps, multilingual contexts, and realistic task flows.

4.  **Task Difficulty Levels**
    We clarified the criteria for defining task difficulty levels, emphasising their basis in task complexity rather than step count alone. Future directions for multidimensional difficulty metrics were suggested to further refine this classification.

5.  **Evaluation Metrics**
    Additional explanations were provided for key metrics, such as termination accuracies, including their significance in diagnosing agent behaviour and guiding improvements in self-evaluator design. We also discussed the number of prompt tokens utilised by agents, which reflects the complexity of the layouts that models in different agents need to handle.

6.  **Fine-Tuned vs. Off-the-Shelf Agents**
    We confirmed that fine-tuned agents were not specifically trained on this benchmark and discussed the generalisation challenges they face. Training datasets for fine-tuned agents were detailed, and performance gaps were analysed in the context of real-world interactions and task complexities.

In the revised paper, the main changes and updates have been highlighted in blue to ensure transparency and easy identification of key revisions. Please note that some minor adjustments and refinements, such as formatting corrections and phrasing improvements, are not highlighted but have also been incorporated to enhance the overall quality and readability of the paper.

We are grateful for all the reviewers’ insights, which have been instrumental in refining our work. We believe these revisions strengthen the paper and improve its clarity. We welcome further discussions and remain committed to addressing any remaining concerns.

---

> ### Author Response · Authors · 2024-11-24
> **Follow-Up on Reviewer Comments**
>
> Dear reviewers, since the end of the discussion period is approaching, we would like to kindly inquire if you have had a chance to review our responses to your comments. We sincerely appreciate the valuable feedback you have provided and have carefully incorporated your suggestions to strengthen our work. Please let us know if there are any remaining concerns we can address.

---

### Meta-Review · Area_Chair_f12N · 2024-12-23

**Metareview:**

This paper presents SmartPhoneAgent (SPA-) Benchmark, roughly consisting of a set of common mobile tasks, a method for integrating with a slew of current Android-based LLM-based agents, and an evaluation method that takes outputs of those agents on tasks and judges them.  All reviewers lauded the comprehensiveness of the tasks and ability to integrate with current LLM-based agents, along with experimental results on those real agents (e.g., current LLM-based agents are still thwarted by inconsistent Android UI design elements, etc).

**Additional Comments On Reviewer Discussion:**

Reviewers were broadly responsive to frequent author rebuttal updates, and many concerns were addressed.  We encourage the authors to consider those concerns that were not addressed, some of reviewer VmmX's specifically.

---

### Decision · Program_Chairs · 2025-01-22

Accept (Spotlight)